# The impact of cloudiness and cloud type on the atmospheric heating rate of black and brown carbon in the Po Valley

Luca Ferrero[1,*], Asta Gregorič[2,3], Grisa Močnik[3,4], Martin Rigler[2], Sergio Cogliati[1,5], Francesca Barnaba[6], Luca Di Liberto[6], Gian Paolo Gobbi[6], Niccolò Losi[1] and Ezio Bolzacchini[1]

[1]GEMMA and POLARIS Research Centers, Department of Earth and Environmental Sciences, University of Milano-Bicocca, Piazza della Scienza 1, 20126, Milan, Italy.

[2]Aerosol d.o.o., Kamniška 39A, SI-1000 Ljubljana, Slovenia.

[3]Center for Atmospheric Research, University of Nova Gorica, Vipavska 11c, SI-5270 Ajdovščina, Slovenia.

[4]Department of Condensed Matter Physics, Jozef Stefan Institute, SI-1000 Ljubljana, Slovenia.

[5]Remote Sensing of Environmental Dynamics Lab., DISAT, University of Milano-Bicocca, P.zza della Scienza 1, 20126, Milano, Italy

[6]ISAC -CNR, Roma - Tor Vergata, Via Fosso Del Cavaliere 100, 00133, Roma, Italy.

*Correspondence to*: Luca Ferrero (luca.ferrero@unimib.it)

**Abstract**. We experimentally quantified the impact of cloud fraction and cloud type on the heating rates (HRs) of black and brown carbon ($HR_{BC}$ and $HR_{BrC}$). In particular, we examined in more detail the cloud effect on HRs detected in a previous study (Ferrero et al., 2018). High time-resolution measurements of the aerosol absorption coefficient at multiple-wavelengths were coupled with spectral measurements of the direct, diffuse and surface reflected irradiance, and with lidar-ceilometer data during a field campaign in Milan, Po Valley (Italy). The experimental set-up allowed a direct determination of the total HR (and its speciation: $HR_{BC}$ and $HR_{BrC}$) in all sky condition (from clear-sky to cloudy). The highest total HR values were found in the middle of winter (1.43±0.05 K day$^{-1}$) and the lowest in spring (0.54±0.02 K day$^{-1}$). Overall, the $HR_{BrC}$ accounted for 13.7±0.2% of the total HR, the BrC being characterized by an absorption Angstrom exponent (AAE) of 3.49±0.01. To investigate the role of clouds, sky conditions were classified in terms of cloudiness (fraction of sky covered by clouds: oktas) and cloud types: stratus (St), cumulus (Cu), stratocumulus (Sc), altostratus (As), altocumulus (Ac), cirrus (Ci) and cirrocumulus-cirrostratus (Cc-Cs). During the campaign, clear sky conditions were present 23% of the time, the remaining time (77%) being characterized by cloudy conditions. Average cloudiness was 3.58±0.04 oktas (highest in February: 4.56±0.07 oktas, lowest in November: 2.91±0.06 oktas). St were mostly responsible of overcast situations (oktas=7-8, frequency: 87 and 96%), Sc dominated the intermediate cloudiness conditions (oktas=5-6, frequency: 47 and 66%) and the transition from Cc-Cs to Sc determined moderate cloudiness (oktas=3-4); finally, low cloudiness (oktas=1-2) were mostly dominated by Ci and Cu (frequency: 59 and 40%, respectively).

HR measurements showed a constant decrease with increasing cloudiness of the atmosphere enabling us to quantify for the first time the bias (in %) in the aerosol HR introduced by the simplified assumption of clear-sky conditions from radiative transfer model calculations. Our results showed that the HR of light absorbing aerosol was ~20-30% lower in low cloudiness (oktas=1-2) and over 80% lower in complete overcast conditions (i.e.,

oktas=7-8), compared to clear sky ones. This means that, in the simplified assumption of clear-sky conditions, the HR of light absorbing aerosol can be largely overestimated (by 50% in low cloudiness, oktas=1-2 and up to 500% in complete overcast conditions, i.e., oktas=7-8).

The impact of different cloud types on the HR was also investigated. Cirrus were found to have a modest impact, decreasing the $HR_{BC}$ and $HR_{BrC}$ by -5% at most. Cumulus decreased the $HR_{BC}$ and $HR_{BrC}$ by -31±12 and -26±7%, respectively, while cirrocumulus-cirrostratus by -60±8 and -54±4%, which was comparable to the impact of altocumulus (-60±6 and -46±4%). A higher impact on $HR_{BC}$ and $HR_{BrC}$ was found for stratocumulus (-63±6 and -58±4%, respectively) and altostratus (-78±5 and -73±4%, respectively). The highest impact was associated to stratus suppressing the $HR_{BC}$ and $HR_{BrC}$ by -85±5 and -83±3%, respectively. The presence of clouds caused a decrease of both $HR_{BC}$ and $HR_{BrC}$ (normalized to the absorption coefficient of the respective species) by a factor of -11.8±1.2% and -12.6±1.4% per okta. This study highlights the need to take into account the role of both cloudiness and different cloud types when estimating the HR caused by both BC and BrC, and in turn decrease the uncertainties associated with the quantification of their impact on the climate.

## 1    Introduction

The impact of aerosols on the climate is traditionally investigated focusing on their direct, indirect and semi-direct effects (Bond et al., 2013; IPCC, 2013; Ferrero et al., 2018, 2014; Ramanathan and Feng, 2009; Koren et al. 2008, 2004; Kaufman et al., 2002). Direct effects are related to the sunlight interaction with aerosols trough absorption and scattering; indirect effects are related to the ability of aerosol to act as cloud condensation nuclei affecting the clouds' formation and properties; semi-direct effects are those related to a feedback on cloud evolution affecting other atmospheric parameters (e.g. the thermal structure of the atmosphere) (IPCC, 2013; Ramanathan and Feng, 2009; Koren et al. 2008, 2004; Kaufman et al., 2002). Both direct and indirect radiative effects of anthropogenic and natural aerosols on climate are still the major sources of uncertainties (IPCC, 2013). Recent studies show, for example, that the aerosol direct radiative effect, on a global scale, may switch from positive to negative forcing on short (e.g. daily) time-scales (Lolli et al., 2018; Tosca et al., 2017; Campbell et al., 2016). This is due to the fact that aerosols are a heterogeneous complex mixture of particles characterized by different size, chemistry, and shape (e.g., Costabile et al., 2013), greatly varying in time and space both in the horizontal and vertical dimension (e.g., Ferrero et al., 2012). On a global scale, most of the values reported for the aerosol direct radiative effect were derived from models (Bond et al., 2013; Koch and Del Genio, 2010). This has the advantage of providing continuous direct radiative effect fields in space and time. However, inaccuracies related to simplified model assumptions on chemistry, shape, and the mixing state of particles can affect the results (Nordmann et al., 2014; Koch et al., 2009), amplifying the uncertainties on the estimated global and regional aerosol effects on the climate (Andreae and Ramanathan, 2013). The aerosol direct radiative effect has been usually determined in clear-sky conditions both in model simulations and measurements. Although the clear sky approximation is useful when comparing measurements to radiative transfer modelling outcomes during experimental campaigns performed in fair weather conditions (e.g., Ferrero et al., 2014; Ramana et al., 2007), in general this simplification cannot capture the complexity of the phenomenon in the majority of weather conditions (Myhre et al., 2013). In fact, clouds are one of the most important factors influencing the solar radiation reaching the ground. By scattering and absorbing the radiation, clouds can affect the radiation magnitude and modify its spectrum especially in the ultraviolet (UV)

region (López et al., 2009; Thiel et al., 2008; Calbó et al., 2005). During cloudy conditions the global irradiance is usually reduced, however, the presence of clouds sometimes results in short-term enhancement of global irradiance (Duchon and O'Malley, 1999). In some specific cases, scattering of radiation from the sides of the cloud may enhance global irradiance in the UV to the levels higher than those in clear sky conditions (Mims and Frederick, 1994; Feister et al., 2015). Mims and Frederick (1994) determined the that scattering from the sides of cumulus clouds can enhance the total (global) UV-B solar irradiance by 20% or more over the maximum solar noon value when cumulus clouds were close to (but not when blocking) the solar disk. In a similar way, Feister et al. (2015) concluded that the scattering of solar radiation by clouds can enhance UV irradiance at the surface – for example, cumulonimbus clouds, with top heights close to the tropical tropopause layer, have the potential to significantly enhance diffuse UV-B radiance over its clear sky value. UV radiation also interacts with aerosols, and particularly with those featuring significant absorption values in this spectral region. UV represents an important region for brown carbon (BrC) absorption with respect to other light absorbing aerosol (LAA) components (e.g. black carbon, BC). Thus, the presence of clouds could influence the impact of different LAA species on the climate in a different way.

Up to now, the role of cloudiness and cloud type on the aerosol direct radiative effect was poorly investigated. Matus et al. (2015) recently used a complex combination of the CLOUDSAT's satellite multi-sensor radiative flux and heating rate (HR) products to infer both the direct radiative effect at the top-of-atmosphere and HR profiles of aerosols that lie above the clouds. The study showed how results were affected by the cloudiness (e.g. cloud fraction) and, for example for the south eastern Atlantic, reported a direct radiative effect ranging from -3.1 to -0.6 W m$^{-2}$ going from clear sky to cloudy conditions.

A further investigation by Myhre et al. (2013) reported results of modelling simulations during the AeroCom Project (Phase II): in all sky conditions (thus including the effect of clouds) they estimated an all-sky direct radiative effect for total anthropogenic aerosols of -0.27 W m$^{-2}$ (range: −0.58 to −0.02 W m$^{-2}$), this being about half of the clear sky one. The most important factors responsible for the observed difference were the amount of aerosol absorption and the location of aerosol layers in relation to clouds (above or below). In fact, the presence of LAA (mainly BC, BrC, and mineral dust) might have important effects on the radiative balance. It is estimated that, due to its absorption of sunlight, BC is the second most important positive anthropogenic climate-forcing agent after $CO_2$ (Bond et al., 2013; Ramanathan and Carmichael, 2008), while BrC contributes ~10-30% to the total absorption on a global scale (Ferrero et al., 2018; Kumar et al., 2018; Shamjad et al., 2015; Chung et al., 2012). As a main difference compared to $CO_2$, absorbing aerosols are short-lived climate forcers, thus representing a potential global warming mitigation target. However, the real potential benefit of any mitigation strategy should also be based on observational measurements, possibly carried out in all sky conditions.

It also noteworthy that the HR induced by absorbing aerosol can trigger different atmospheric feedbacks. BC and mineral dust can alter the atmospheric thermal structure, thus affecting atmospheric stability, cloud distribution and even synoptic winds such as the monsoons (IPCC, 2013; Bond et al., 2013; Ramanathan and Feng, 2009; Koch et al., 2009; Ramanathan and Carmichael, 2008; Koren et al. 2008, 2004; Kaufman et al., 2002). These feedbacks should be quantified on the basis of HR measurements in all sky conditions. In agreement with this, both Andreae and Ramanathan (2013) and Chung et al. (2012) called for model-independent, observation-based determination of the absorptive direct radiative effect of aerosols. Since, similarly to aerosols, cloudiness and cloud type change

on short time scales, long-term, highly time-resolved measurements covering different conditions, are necessary to unravel the impact of LAA on the HR.

Satellite-based studies investigated the role of cloudiness and cloud type on the HR of aerosol layers above clouds (Matus et al., 2015). To our knowledge, there has been no experimental investigation of cloudiness and cloud type impact on the HR of aerosol layers below clouds, where most of the aerosol pollution typically resides. Cloud-aerosol feedbacks can strongly depend on the HR magnitude in cloudy conditions. As a matter of fact, the atmospheric heating induced by absorbing aerosol is traditionally related to a decrease of atmospheric relative humidity and less cloud cover (semi-direct effect). This effect can further increase the amount of the incoming solar radiation that reaches Earth's surface (and any close-to-surface LAA layers), leading to a positive feedback characterized by additional warming and a further decrease in the cloud amount (e.g. Koren et al., 2004). However, Perlwitz and Miller (2010) reported a counterintuitive feedback linking the atmospheric heating induced by tropospheric absorbing aerosol to a cloud cover increase (especially low-level clouds) due to a delicate interplay between relative humidity and temperature. The study concluded that high absorption by aerosols was responsible for two counter-acting processes: a large diabatic heating of the atmospheric column (thus decreasing relative humidity), and a corresponding increase in the specific humidity able to exceed the temperature effect on relative humidity, with the net result of increasing low cloud cover with increasing aerosol absorption. This is an important result that underlines the importance of measuring the atmospheric HR in cloudy conditions as a constraint and/or input for more comprehensive climate models to shed light on the sign and magnitude of the related feedbacks on cloud dynamics.

This study attempts to experimentally measure for the first time the impact of different cloudiness and cloud types on the HR exerted by near-surface LAA species. The study was performed in Milan (Italy), located in the middle of the Po Valley (section 2), which is an air pollution hot-spot in Europe with meteorological characteristics similar to those of a multitude of basin valleys surrounded by hills or mountains in which low wind speeds and stable atmospheric conditions promote the accumulation of aerosol (Zotter et al., 2017; Moroni et al., 2013, 2012; Ferrero et al., 2013, 2011a; Barnaba et al., 2010; Carbone et al., 2010; Rodriguez et al., 2007). Cloud presence cannot be neglected over the investigated area considering that in the last 50 years annual mean cloudiness, expressed in oktas, is estimated to be ~5.5 over Europe (Stjern et al., 2009) and ~4 over Italy (Maugeri et al., 2001). This feature is similar with 80 years of data of cloud cover in the United States (Crocke et al., 1999). To determine the HR, we used a methodology previously developed in Ferrero et al. (2018), and further extended it here to explore the effects of cloudiness and different cloud types on HR of BC and BrC. More specifically, this work introduces the following novelties: 1) it describes the interaction between cloudiness and light-absorbing aerosol, presenting the aerosol HR as a function of cloudiness, and in turn estimates the systematic bias introduced by incorrectly assuming clear-sky conditions in radiative transfer models; 2) it introduces a cloud type classification and investigates the impact of both cloudiness and cloud types on the total HR; 3) it separates BC and BrC contributions and investigates their relative impact on the total HR in function of sky conditions. The results presented in this study add an important piece of information in the general context of cloud-aerosol interactions and their influence on HR.

## 2 Methods

Aerosol, cloud and spectral irradiance measurements were carried in an experimental measurement station located in Milan (Italy) on the rooftop (10 m above the ground level) of the U9-building of the University of Milano-Bicocca (45°30'38"N, 9°12'42"E, Italy; Figure 1). The site is located in the midst of the Po Valley, one of the most industrialized and heavily populated area in Europe. In the Po Valley, stable atmospheric conditions often occur causing a marked seasonal variation of aerosol concentrations within the mixing layer (Barnaba et al., 2010), well visible even from satellites (Ferrero et al., 2019; Di Nicolantonio et al., 2007, 2009; Barnaba and Gobbi 2004). A full description of the aerosol behavior in Milan at the University of Milano-Bicocca and the related aerosol properties (vertical profiles, chemistry, hygroscopicity, sources, and toxicity) are reported in previous studies (Diemoz et al., 2019a; Lorelei et al., 2019; D'Angelo et al., 2016; Curci et al., 2015; Ferrero et al., 2015, 2010; Sangiorgi et al., 2011, 2014; Sandrini et al., 2014). In the framework of the present work it is important to underline that the U9 experimental site is particularly well suited for atmospheric radiation measurements: it is characterized by a full hemispherical sky-view equipped with the instruments described in Section 2.1. The measurement setup allowed the experimental determination of the instantaneous aerosol HR (K day$^{-1}$) induced by absorbing aerosol as detailed in Section 2.2. The methodological approach used to quantify the cloud fraction and to classify the cloud type is reported in Section 2.3.

### 2.1 Instruments

The aerosol, cloud and radiation instrumentations has been installed at the U9 sampling site in Milan since 2015. Site location is shown in Figure 1. The complete instrumental set up (Figure S1) is described hereafter.

#### 2.1.1 Light absorbing aerosol measurements and apportionment

Measurements of the wavelength dependent aerosol absorption coefficient $b_{abs(\lambda)}$ were obtained using the Magee Scientific Aethalometer AE-31. This allowed multi spectral measurements (7-$\lambda$: 370, 470, 520, 590, 660, 880 and 950 nm) in the wide UV-VIS-NIR region, not available from other instruments (e.g. MAAP, PSAP, photoacoustic) (Virkkula et al., 2010; Petzold et al., 2005). This spectral range is needed for the HR determination (section 2.2). The use of Aethalometers also presents the advantage of global long-term data series (Ferrero et al., 2016; Eleftheriadis et al., 2009; Collaud-Coen et al., 2010; Junker et al., 2006) that could allow to derive historical data of the HR in the future.

To account for both the multiple scattering enhancement (the elongation of the optical path induced by the filter fibers) and the loading effects (the non-linear optical path reduction induced by absorbing particles accumulating in the filter), the AE-31 data were corrected applying the Weingartner et al. (2003) procedure (Ferrero et al., 2018, 2014, 2011b; Collaud-Coen et al., 2010). As detailed by Collaud Coen et al. (2010), the Weingartner et al. (2003) procedure compensates for all the Aethalometer artifacts (the backscattering is indirectly included within the multiple scattering correction), showing a good robustness (negative values are not generated and results in good agreement with other filter photometers), and, most importantly, it does not affect the derived aerosol Absorption Angstrom Exponent (AAE) (fundamental for HR determination, section 2.2). Overall, the multiple scattering parameter $C$ was 3.24±0.03 as obtained by comparing the AE-31 data at 660 nm with a MAAP at the very similar wavelength (637 nm, Müller et al., 2011) (regression between AE-31 and MAAP in Figure S2). This value lies very close to that suggested by the Global Atmospheric Watch (GAW) program (2016), i.e. C=3.5. The physical

meaning of the similarity between the obtained C value (3.24±0.03) and the GAW one implies that Milan (in the middle of the Po Valley) is characterized by continental type aerosols (e.g. Carbone et al., 2010) and consistent with the global average. To verify the reliability of the obtained C value, it was also computed following Collaud Coen et al. (2010) procedure. They defined the reference value of C ($C_{ref}$ = 2.81±0.11) for the AE-31 tape based on data from pristine environments (Jungfraujoch and Hohenpeissenberg sites where aerosol has a single scattering albedo of ~1); at the same time, Collaud Coen et al. (2010) defined C for any type of aerosol as follows:

$$C = C_{ref} + \alpha \frac{\omega_0}{1-\omega_0} \tag{1}$$

where α is the parameter for the Arnott (2005) scattering correction (0.0713 at 660 nm) and $\omega_0$ the single scattering albedo. In wintertime in Milan, within the mixing layer, the single scattering albedo was found to be 0.846±0.011 at 675 nm by Ferrero et al. (2014). From eq. 1 it follows that the expected C in Milan is 3.20±0.35; within its range the experimental 3.24±0.03 value lies. Details concerning wavelength differences are discussed in the Supplement (section Measured and computed C factor). The loading effects were dynamically determined following the Sandradewi et al. (2008b) approach while the final equivalent BC concentrations (eBC) were obtained applying the AE-31 apparent mass attenuation cross-section (16.6 $m^2$ $g^{-1}$ at 880 nm).

The above mentioned compensation procedures introduce an uncertainty in the absorption coefficient measurements. Collaud-Coen et al. (2010) tested these procedures in different sites and estimated the global accuracy of the Weingartner et al. (2003) correction applied in the present work to be ~23%. Moreover, Drinovec et al. (2015) showed a good agreement between Aethalometer AE-31 data (corrected using Weingartner et al., 2003) and that of the new version AE-33 with a slope close to unity and $R^2$>0.90. Thus, the Collaud-Coen et al. (2010) accuracy estimation is considered as the worst scenario.

As the spectral signature of the aerosol absorption coefficient $b_{abs(\lambda)}$ reflects the different nature of absorbing aerosol (BC and BrC), once $b_{abs(\lambda)}$ is obtained, it can be apportioned to determine the contributions of BC and BrC, respectively. This result can be achieved considering that BC aerosol absorption is characterized by an Absorption Angstrom Exponent, AAE ≈1 (Massabò et al., 2015; Sandradewi et al., 2008a; Bond and Bengstrom, 2006). Conversely, BrC absorption is spectrally more variable, with an AAE from 3 to 10 (Ferrero et al., 2018; Shamjad et al., 2015; Massabò et al., 2015; Bikkina et al., 2013; Yang et al., 2009; Kirchstetter et al., 2004). The wavelength dependence of the absorption coefficient of BrC can be described by the simple harmonic oscillator reported in Moosmüller et al. (2011) explaining that the much lower absorption in the IR region (compared to UV) is a consequence of the resonances in the UV from which the IR region is far removed. This calculation also yields a decreasing AAE values with increasing wavelengths. This is equivalent to the band-gap model with the Urbach tail as detailed in Sun et al. (2007) and references in Moosmüller et al. (2011), where the key factor is the difference between the highest occupied and lowest unoccupied energy state of the molecules included in the BrC ensemble. In this study we determined $AAE_{BrC}$ following the innovative apportionment method proposed by Massabò et al. (2015). This allows to apportion $b_{abs(\lambda)}$ to BC and BrC and to determine, at the same time, the $AAE_{BrC}$ assuming that all of BrC results from biomass burning. The method by Massabò et al. (2015) was previously applied to the Milan U9 measurements leading to an annual average $AAE_{BrC}$ = 3.66±0.03.

### 2.1.2 Radiative, meteorological and lidar measurements

Spectral irradiance measurements were collected using a Multiplexer-Radiometer-Irradiometer (Figure S1; details in Cogliati et al., 2015) which resolves the UV-VIS-NIR spectrum (350 - 1000 nm) in 3648 spectral bands (3648-element linear CCD-array detector (Toshiba TCD1304AP, Japan) for both the downwelling and the upwelling radiation fluxes. The instrument was developed at the University of Milano-Bicocca using an optical switch (MPM-2000-2x8-VIS, Ocean Optics Inc., USA) to sequentially select between different input fibers fixed to up- and down-facing entrance fore-optics. The configuration used in the present work connects each spectrometer to 3 input ports: 1) The CC-3 cosine-corrected irradiance probes to collect the down-welling irradiance; 2) the bare fiber optics with a 25° Field-of-View to measure the up-welling radiance from the terrestrial surface; 3) the blind port that is used to record the instrument dark-current. A 5 m long optical fiber with a bundle core with a diameter of 1 mm is used to connect the entrance fore-optics to the multiplexer input, while the connection between the multiplexer output ports and the spectrometers is obtained with 0.3 m long optical fibers. The set-up allows to sequentially measure dark-current and both up- and down-welling spectra simultaneously with the two spectrometers. The two spectrometers used are High Resolution HR4000 holographic grating spectrometers (Ocean Optics Inc., USA). Finally, the Multiplexer-Radiometer-Irradiometer was equipped with a rotating shadow-band to measure separately the spectra of the direct, diffuse and reflected irradiance ($F_{dir}(\lambda)$, $F_{dif}(\lambda)$, $F_{ref}(\lambda)$). The reflected irradiance originated from a Lambertian concrete surface (due to its flat and homogeneous characteristics which well represents the average spectral reflectance of the Milano urban area; Ferrero et al, 2018). Broadband (300-3000 nm) downwelling (global and diffuse) and upwelling (reflected) irradiance measurements were also collected using LSI-Lastem radiometers (DPA154 and C201R, class 1, ISO-9060, 3% accuracy). Diffuse broadband irradiance was measured using the DPA154 global radiometer equipped with a shadow band whose effect was corrected (Ferrero et al., 2018) to determine the true amount of both diffuse and direct (obtained after subtraction from the global) irradiance. Next, MRI spectra were normalized and completed with normalized literature spectra (Ferrero et al., 2018) to cover the broadband range (300−3000 nm) and irradiance intensity measured by standard LSI-Lastem pyranometers allowing the HR to be evaluated over the whole shortwave range ($b_{abs}(\lambda)$ was estimated outside the AE-31 range using its AAE). The approach was previously validated (Ferrero et al., 2018): the HR in the strict UV−Vis-NIR range (350−950 nm of the AE31 and the MRI), accounted on average for 86.4 ± 0.4% of the total broadband values.

In addition to radiation measurements, temperature, relative humidity, pressure and wind parameters were measured using the following LSI-Lastem sensors: DMA580 and DMA570 for thermo-hygrometric measurements (for T and RH: range -30 - +70 °C and 10% - 98%, accuracy of ± 0.1 °C and ± 2.5% sensibility of 0.025°C and 0.2%), the CX110P barometer model for pressure (range 800-1100 hPa, accuracy of 1 hPa) and the combiSD anemometer (range of 0 - 60 m/s and 0-360°) for wind.

The experimental station U9 is also equipped with an Automatic Lidar-Ceilometer operated by ISAC-CNR in the framework of the Italian Automated LIdar-CEilometers NETwork (ALICENET, www.alice-net.eu) and contributing to the EUMETNET E-Profile Network (https://www.eumetnet.eu/). This is a Jenoptik Nimbus 15k biaxial lidar-ceilometer operating 24 hours per day, 7 days per week. It is equipped with a Nd:YAG laser that emits light pulses at 1064 nm with an energy of 8 μJ per pulse and a repetition rate of 5 kHz. The backscattered light is detected by an avalanche photodiode in photon counting mode (Wiegner & Geiß, 2012; Madonna et al.,

2015). The vertical and temporal resolution of the raw signals are 15 m and 30 seconds, respectively. In order to
270 improve the signal-to-noise ratio of the backscatter signal, the signal is processed with temporal averages of 2
minutes. The full overlap is obtained at altitude of some hundred meters above the observation site and overlap
correction functions are applied in the first layers. The Nimbus 15k lidar-ceilometer is able to determine cloud
base height (CBH), penetration depth and, with specific processing, mixing layer height, vertical profiles of aerosol
optical and physical properties (e.g., Diemoz et al., 2019a, 2019b; Dionisi et al., 2018). We used the U9 ceilometer
data for cloud layering and relevant cloud base height as the system can reliably detect multiple cloud layers and
cirrus clouds (Wiegner et al., 2014; Boers et al., 2010; Martucci et al., 2010) within its operating vertical range
(up to 15 km). Given the vertical resolution of the instrument, expected uncertainty of the cloud base height derived
by the lidar-ceilometer is less than ±30 m.

Global and diffuse irradiance measurements, coupled with the ceilometer data were used to determine the sky
cloud fraction and to classify the cloud types by following the methodology presented in the Section 2.3.

**2.2    Heating rate measurements**

The instantaneous aerosol HR (K day$^{-1}$) induced by absorbing aerosol is experimentally obtained using the
methodology reported and validated in Ferrero et al. (2018), where the reader is referred to for the details of the
285 approach. Here we briefly summarize the method.

The heating rate is determined from the air density ($\rho$, kg m$^{-3}$), the isobaric specific heat of dry air ($C_p$, 1005 J kg$^{-1}$ K$^{-1}$) and the  radiative power absorbed by aerosol per unit volume of air (W m$^{-3}$) describing the interaction
between the radiation (either direct from the sun, diffuse by atmosphere and clouds, and reflected from the ground)
and the LAA (BC and BrC in Milan). The HR is determined as follows (Ferrero et al., 2018):

$$HR = \frac{1}{\rho C_p} \cdot \sum_{dir,dif,ref} \int_{\theta=0}^{\theta=\pi/2} \int_{\lambda=300}^{\lambda=3000} \frac{F_{dir,dif,ref}(\lambda,\theta)}{\cos(\theta)} b_{abs}(\lambda) d\lambda d\theta \qquad (2)$$

where the subscripts *dir*, *dif* and *ref* refer to the direct, diffuse and reflected components of the spectral irradiance
*F* of wavelength $\lambda$ impinging on the LAA with a zenith angle $\theta$ (from any azimuth).

Under the isotropic and Lambertian assumptions (as used in Ferrero et al., 2018) equation 2 can be solved
becoming:

$$HR = HR_{dir} + HR_{dif} + HR_{ref} =$$

$$= \frac{1}{\rho C_p} \cdot \left[ \frac{1}{\cos(\theta_z)} \int_\lambda F_{dir}(\lambda) \, b_{abs}(\lambda) \, d\lambda + 2 \int_\lambda F_{dif}(\lambda) \, b_{abs}(\lambda) \, d\lambda + 2 \int_\lambda F_{ref}(\lambda) \, b_{abs}(\lambda) \, d\lambda \right] \qquad (3)$$

where $\theta_z$ refers to the solar zenith angle, while $F_{dir}(\lambda)$, $F_{dif}(\lambda)$ and $F_{ref}(\lambda)$ are the spectral direct, diffuse and reflected
irradiances. Eqs. 2 and 3 are related to the concept of actinic flux (Tian et al., 2020; Gao et al., 2008; Liou, 2007);
an extended description, as well as its demonstration is detailed in the Supplement.

As the intensity of the irradiance components is a function of cloudiness and cloud type (section 2.3), eq. 3 enables
to assess the impact of the latter components on the aerosol absorption of shortwave radiation and thus on the
corresponding HR (sections 3.2 and 3.3).

The most important advantages and limitations of this measurement-based approach to derive HR are as follows.
Advantages:

-    no radiative transfer assumptions needed (i.e. no assumption of clear sky conditions), as the parameters
input to equations 3 are all measured quantities,

- possibility to follow the rapid HR dynamic to investigate the HR temporal evolution, as measurements of spectral irradiance and absorption coefficient are carried out with high temporal resolution,

- possibility to derive HR in all sky conditions, as measurements of spectral irradiance and the absorption coefficient are independent from atmospheric conditions enabling us to investigate the impact induced by the clouds.

Limitation:

- the HR is independent of the thickness of the investigated atmospheric layer and refers to the vertical location of the atmospheric layer in which it is experimentally determined. In the present work the HR was determined to the near-surface atmospheric layer.

With respect to this limitation, it should be mentioned that BC and HR vertical profiles data previously collected at the same site and in other valley basins revealed that the HR was constant inside the mixing layer (Ferrero et al., 2014). In fact, above our observational site, vertical profile measurements with a tethered balloon and a lidar-ceilometer were performed since 2005, mostly showing homogeneous concentrations of aerosol (and related extinction coefficient) within the mixing layer, particularly in daytime (Ferrero et al., 2019). The same condition was verified by the lidar-ceilometer data collected during the present campaign (Figure S3, supplemental material). The methodology is therefore believed to be also representative for the whole mixing layer if the aerosol vertical dispersion is homogeneous within this layer. This might not be the case for other regions, where the upper troposphere is impacted by high levels of BrC from biomass burning (Zhang et al, 2019), but Ferrero et al. (2019) showed that in Milan 87.0% of aerosol optical depth signal was built up within the mixing layer, 8.2% in the residual layer and 4.9% in the free troposphere.

## 2.3    Cloud fraction, cloud classification and average photon energy

### 2.3.1    Cloud fraction

The cloudiness was determined following the approach reported in Ehnberg and Bollen (2005) that enables to calculate the fraction of the sky covered by cloud in terms of oktas ($N$), overall leading to 9 classes, corresponding to values of $N$ ranging from 0 (clear sky) to 8 (complete overcast situation). As reported by Ehnberg and Bollen (2005), the amount of global irradiance ($F_{glo}$) is related to the solar elevation angle ($\pi/2-\theta$) and to the cloudiness following the Nielsen et al. (1981) equation:

$$F_{glo}(N) = \left[ \frac{a_0(N)+ a_1(N)\sin\left(\frac{\pi}{2}-\theta\right)+a_3(N)\sin^3\left(\frac{\pi}{2}-\theta\right)-L(N)}{a(N)} \right] \tag{4}$$

where $N$ represents one of the possible 9 classes of sky conditions expressed in oktas (from 0, i.e. clear sky, to 8, i.e. complete overcast) and a, $a_0$, $a_1$, $a_3$ and $L$ are empirical coefficients that enable to compute the expected global irradiance for each oktas class ($F_{glo}(N)$), at a fixed solar elevation angle ($\pi/2-\theta$). Their values, extracted from the original work of Ehnberg and Bollen (2005), are summarized in Table S1. Overall, eq. 4 allows to determine the unique oktas value $N$ by comparing the measured global irradiance ($F_{glo}$) with $F_{glo}(N)$ at any given time.

With this approach, the cloudiness can be used to evaluate the interaction between incoming radiation and LAA in cloudy conditions, but does not provide the opportunity to discriminate between cloud type. The following section describes the methods applied to overcome this limitation by implementing a cloud classification scheme.

### 2.3.2 Cloud classification

Identification of cloud classes is by common practice still largely performed by human observations based on the reference standard defined by the World Meteorological Organization (WMO; https://cloudatlas.wmo.int/en/home.html). However, these observations lack the required time resolution which was needed in the present work to couple highly time resolved HR data with cloud type. Cloud classification literature reports a huge quantity of papers and reviews aimed at classify clouds by means of different techniques and their integration to avoid the limits of a simple human inspection. Most of these rely on different ensemble of instruments: 1) ground based, 2) remote sensing/satellite based or 3) installed on meteorological balloons (Tapakis and Charalambides, 2013). Some examples are reported in Singh and Glennen (2005), Ricciardelli et al. (2008), Calbó and Sabburg (2008), Tapakis and Charalambides (2013).

To exploit the full potential of our measurements, we needed a cloud type classification method able to follow the high temporal resolution of the observations including the high spatial and temporal variability of clouds.

Among the above-mentioned instrumental ensembles, ground based instruments provide measurement of the incident solar irradiance to detect the effect of clouds (Calbò et al., 2001). The concept of using irradiance measurements to estimate cloud types was first introduced in the work of Duchon & O'Malley (1999) which is based on the fact that clouds with different velocities and optical depth cross the slowly changing path of the solar beam over different time durations. Given the available irradiance data (section 2.1), in the present work, the cloud classification starts from the Duchon & O'Malley (1999) method which was successfully applied in the geographical context of the Po Valley (Galli et al., 2004). In particular, we used irradiance measurements ($F_{glo}$) to compute two parameters $R_t$ and $SD_t$ as follows:

$$R_t = \frac{1}{20}\sum_{i=t-10}^{i=t+10} \frac{F_{glo(i)}}{F_{glo\_CS(i)}} \tag{5}$$

$$SD_t = \sigma_{t\pm10}(F_{glo(t\pm10)} \cdot Sf_{t\pm10}) \tag{6}$$

$R_t$ is the 20-minutes running average ratio between the observed global irradiance ($F_{glo}$) and the modelled clear sky irradiance (Robledo and Soler, 2000) expected at the same place ($F_{glo\_CS}$) at the time t; $R_t$ describes the time-dependent cloud efficiency in reducing the incoming solar radiation ($R_t$=1 in perfect clear sky while $R_t$~0 in complete overcast conditions). $SD_t$ represents the 20-minutes standard deviation (SD) of the scaled global irradiance ($F_{glo}\cdot Sf$) centered at the time t, and describes the temporal stability of clouds in the atmosphere (e.g. persistent stratus clouds are characterized by a $SD_t$~0 while cumulus in good weather by higher values of $SD_t$). The scaling factor $Sf_t$ (Duchon & O'Malley, 1999) is given by:

$$Sf_t = \frac{1400\ W\ m^{-2}}{F_{glo\_CS(t)}} \tag{7}$$

Visualization of the SD vs. R (SD-R plot) results thus represents a first tool to distinguish different cloud categories as a function of their efficiency in reducing the incoming solar radiation (R) and their persistency (SD). The potential of the SD-R plot is presented in Figure 2a-h; it shows four examples of the temporal evolution of the observed $F_{glo}$, $F_{glo\_CS}$ and $F_{dif}$ (left column) and the corresponding SD-R diagrams (right column). More in detail:

1- the first case (Figure 2a) shows $F_{glo}$ that follows $F_{glo\_CS}$ without any significant temporal deviation, thus leading to a cluster of data in the SD-R diagram (Figure 2b) characterized by R~1 and SD~0 W m$^{-2}$. These conditions are those associated to clear sky (CS) by Duchon & O'Malley (1999),

2- the second case (Figure 2c) shows $F_{glo}$ completely dominated by the diffuse irradiance ($F_{dif}$) throughout the day (note that in Figure 2c $F_{dif}$ is superimposed on $F_{glo}$); this condition differs completely from the CS case as both R and SD approach 0 (Figure 2d). Duchon & O'Malley (1999) associate these conditions to the presence of persistent stratiform clouds,

3- the third case (Figure 2e) reports $F_{glo}$ approaching $F_{glo\_CS}$ being at the same time characterized by small amplitude oscillations. In this case R ranges between 0.75 and 1 and SD from 0 to ~100 W m$^{-2}$ (Figure 2f). The cluster of data is thus more dispersed than that of the CS case featuring a larger variation in R and SD. Duchon & O'Malley (1999) attributed this situation to the presence of Cirrus (Ci) underlining that in some borderline cases a misclassification between CS and Ci (just based on SD-R plot) could be possible,

4- the last case (Figure 2g) represents a transition from a CS situation (before noon) to cloudy conditions (after midday) characterized by a significant scatter of $F_{glo}$. Figure 2h clearly shows that the sky condition evolves from the CS toward cloudy sky, shifting the R data from ~1 down to ~0.25 and increasing SD from ~100 to ~500 W m$^{-2}$. According to Duchon & O'Malley (1999), the arrival of Cumulus during a "good weather" day could be the reason for such behavior (Cu clouds movement in the sky results in fast sun/shadows transitions). Also, in this case, the SD-R plot alone cannot exclude the presence of other cloud types responsible for a similar behavior (e.g. Altocumulus, Ac; Cirrocumulus, Cc; Cirrostratus, Cs). Note that in order to show the variation of data in the SD-R diagram (Figure 2h) as a function of time, an hourly-resolvedcolor code was assigned to the data points; the corresponding regions in Figure 2g were delimited by dashed lines with the same color code.

Overall, Figure 2a-h shows the potential (and limits) of the SD-R plots for a preliminary broad sky/cloud classification. As mentioned, the SD-R diagram alone leaves margins of misclassification, especially because it is impossible to retrieve the required information when different cloud types at different levels are present simultaneously.

In the present work, we attempted a further refinement of cloud classification including the information of the cloud base height (CBH) and the number of cloud layers obtained from the automated Lidar-Ceilometer measurements. The cloud base height is a key parameter in the characterization of clouds (Hirsch et al., 2011), since its estimation limits the number of potential cloud classes (that the SD-R classifier has to discriminate between) thus maximizing the efficiency of the Duchon & O'Malley (1999) classification algorithm. In fact, ceilometer instruments were developed and are commonly used in airports to operationally detect cloud layers, and their use for aerosol–related studies is more recent. Furthermore, the use of ceilometer data for cloud classification and cloud study purposes does not represent an absolute novelty in the scientific literature as demonstrated by recent works by Huertas-Tato et al. (2017) and Costa-Surós et al. (2013). The availability of CBH information allows to divide cloud type in three fundamental categories (Tapakis and Charalambides, 2013): low level clouds (<2 km), mid-altitude clouds (2-7 km) and high-altitude clouds (>7 km). From a general perspective high level cloud category includes Cirrus (Ci), Cirrocumulus (Cc) and Cirrostratus (Cs); mid-level clouds include Altocumulus (Ac), Altostratus (As), and Nimbostratus (Ns); low level clouds include Cumulus (Cu), Stratocumulus (Sc), Stratus (St), and Cumulonimbus (Cb) (Tapakis and Charalambides, 2013; Ahrens, 2009; Cotton et al., 2011).

We color-coded the SD-R diagram in Figure 3 using the ceilometer-based information on cloud altitude. The plot shows that, on average, low level clouds are located on the left side of the SD-R diagram (stratiform clouds), high-altitude clouds are conversely on the opposite side (Ci and Cu clouds); finally, mid-altitudes clouds mostly cover the central part describing all the possible transitions/combinations from St to Cu and Ci, e.g. altostratus (As), altocumulus (Ac).

Overall, adding the CBH information to the SD-R plot enabled us to identify eight cloud types: St (stratus), Cu (cumulus) and Sc (stratocumulus) as low level clouds; As (altostratus) and Ac (altocumulus) as mid-altitude clouds; Ci (cirrus) and Cc-Cs (cirrocumulus and cirrostratus merged in one single class) as high-altitude clouds.

A summary of the threshold values of R, SD, and cloud level used here to the final cloud classification is given in Table 1, the R and SD limits being based on the works of Duchon & O'Malley (1999) and Harrison et al. (2008) and those of CBH being derived considering the cloud properties at midlatitudes.

Finally, to avoid misclassification due to the presence of multiple cloud layers, the analysis was limited to those cases where only one cloud layer was detected by the ceilometer (8405 single layer cases, representing 61% of all measurements). Another reason for limiting the analyses to one cloud layer is due to the main aim of this work: to quantify the effects of different cloudiness and cloud types on LAA HR. We wanted to avoid conditions with multiple-layer clouds as this would result in confounding information for the purpose of the present study.

Figure 4 shows the SD-R diagram of all data (grey) with superimposed R and SD mean value and 99% confidence interval of each of the eight identified cloud classes, plus clear sky (CS). The final cloud classification was obtained for the period from November 2015 - March 2016, during which all necessary parameters were available (section 3).

Since this methodology is applied for the first time in the Po Valley, a complete validation of the aforementioned approach is reported in Appendix B ("Cloud type validation"). It includes two validation exercises: the first was carried out comparing the present automatized cloud classification with a visual cloud classification based on sky images collected during 1 month of wintertime field campaign; the second was carried out comparing the present automatized cloud classification with the one discussed by Ylivinkka et al., (2020). In fact, simultaneously to the submission of our work, Ylivinkka et al. (2020) proposed a classification based on the coupling of irradiance and CBH measurements. Overall, based on these comparisons, agreement of our classification is of 80% with the visual approach and of 90% with the Ylivinkka et al., (2020) methodology, these results further demonstrating the reliability of the cloud classification algorithm used in our study.

## 3    Results and Discussion

Data measured over Milan from November 2015 to March 2016 are presented in Section 3.1, this period covering the simultaneous presence of radiation, lidar-ceilometer and absorption information necessary for the analysis. The role of cloudiness and cloud type on the total HR is discussed in section 3.2, and in section 3.3, the impact of clouds on the HR is discussed with respect to the light absorbing aerosol species: BC and BrC. All data are reported as mean±95% confidence interval.

### 3.1    eBC, irradiance, HR and cloud data presentation

Highly time resolved data (5 minutes) of eBC, $F_{glo}$, CBH, cloudiness (oktas) and resulting HR are shown in Figure 5; their monthly average values are presented in Figure 6a and summarized in Table 2.

The lowest eBC and $b_{abs}$(880 nm) values (monthly averages: 1.54±0.04 µg m$^{-3}$ and 7.6±0.2 Mm$^{-1}$) were recorded in March while, their highest values were found in December (6.29±0.09 µg m$^{-3}$ and 31.1±0.5 Mm$^{-1}$, respectively) with a maximum value of 27.44 µg m$^{-3}$ (135.7 Mm$^{-1}$). In December, the average PM$_{10}$ and PM$_{2.5}$ were also at their maximum, with 73.1±0.6 and 69.3±0.6 µg m$^{-3}$, respectively (source: Milan Environmental Protection Agency, ARPA Lombardia) and the eBC accounted for ~10% of PM mass concentration. These high values of eBC and PM$_{10}$ and PM$_{2.5}$ agree with those observed previously in wintertime in the Po Valley, when strong emissions in the Po Valley are released into a stable boundary layer (Sandrini et al., 2014; Ferrero et al., 2011b, 2014, 2018; Barnaba et al., 2010). During the investigated period, the lowest monthly irradiance value was observed in December ($F_{glo}$: 141±4 W m$^{-2}$; Table 2) while the highest in March ($F_{glo}$: 310±7 W m$^{-2}$). The highest monthly average HR was recorded in December (1.43±0.05 K day$^{-1}$) while the lowest one in March (0.54±0.02 K day$^{-1}$; see Figure 6a, and Table 2). Even though the HR monthly behavior correlated with eBC (Table 2; R$^2$=0.82, not shown), it is also useful to compare the maximum to minimum ratio of eBC monthly mean (December to March, eBC ratio: 4.10±0.12) to the same for the HR (2.65±0.16). This ratio is higher for eBC because the incoming irradiance was lower in December ($F_{glo}$: 141±4 W m$^{-2}$; Figures 6b) with respect to March ($F_{glo}$: 310±7 W m$^{-2}$, ratio of 0.45±0.02), partially compensating the marked wintertime increase of eBC. This is due to the interaction of LAA with $F_{dir}$. In fact, once $F_{dir}$ is scaled by $cos(\theta_z)$ (eq. 3, section 2.2, Figure S4) it is quite constant along the year (and perfectly constant only in clear sky conditions). Conversely, the diffuse and reflected irradiance, under the isotropic and Lambertian assumptions (eq. 3), remain seasonally modulated (Figure S4).

These observations illustrate the importance of both the amount and the type (direct, diffuse and reflected) of radiation that interacts with LAA. In brief, any process able to influence the total amount and the type of impinging irradiance (e.g. presence/absence of clouds, cloudiness and cloud type) will result in a different HR, even at constant LAA concentrations (and their absorption). The investigation of this aspect is the main focus and added value of this study. High resolution data (Figure 5 and Figure S4) provided a first hint on the importance of cloud presence on HR; a sharp global irradiance decrease was observed in cloudy conditions, especially in the presence of low level clouds (low CBH) and high cloud cover (7-8 oktas).

Thus, both cloudiness and cloud type were carefully determined as detailed in sections 2.3.1 and 2.3.2. Overall, during the whole campaign, the average cloudiness was 3.58±0.04 oktas with the highest monthly value in February (4.56±0.07 oktas) and the lowest in November (2.91±0.06 oktas). These data are in line with the mean cloudiness over Europe (~5.5 oktas; Stjern et al., 2009) and over Italy (~4 oktas; Maugeri et al., 2001). Moreover, during the campaign, clear sky (CS) conditions were only present 23% of the time, the remaining time (77%) being characterized by partially cloudy (35%, 1-6 oktas) to totally cloudy (42%, 7-8 oktas) conditions.

Cloudy conditions are therefore frequent. The frequency of specific cloud-type occurrence is given in Figure 7a. The dominating cloud type was St (42%), followed by Sc (13%) Ci, Cc-Cs (7% and 5%, respectively). The contribution of each cloud type to the cloudiness is reported in Figure 7b. While St were mostly responsible of overcast situations (oktas=7-8, frequency: 87 and 96%), Sc dominated the intermediate cloudiness conditions (oktas=5-6, frequency: 47 and 66%) and the transition from Cc-Cs to Sc determined moderate cloudiness (oktas=3-4). Finally, low cloudiness (oktas=1-2) were mostly dominated by Ci and Cu (frequency: 59 and 40%,

respectively). As mentioned (section 2.3.2 and Figure 4), low level clouds (<2 km) include Stratus (St), Cumulus (Cu) and Stratocumulus (Sc), mid-altitude clouds (2-7 km) include Altostratus (As), and Altocumulus (Ac) and high-altitude clouds (>7 km) include Cirrus (Ci), Cirrocumulus and Cirrostratus (Cc-Cs). Thus, it is clear that the higher cloud cover (higher oktas) is due to a higher frequency of low-mid altitude clouds. This is evident in Figure 7b which reports the average CBH for each okta. CBH was related with oktas (Figure S5a, Supplemental material) underling the linkage (together with Figure 7b) between the fraction of sky covered by clouds and the cloud type responsible for it, at least at the measuring site. Indeed, the cloudiness is a non-linear function of the cloud type, as cloud type are related to the meteorological patterns: e.g. highly persistent stratiform clouds generate cloudy weather in conditions with lower wind (see Supplemental material for further details). Figure 8 summarizes the average cloudiness associated with different cloud type showing an oktas rise from cirrus clouds (0.51±0.05 oktas) till stratus clouds (7.20±0.04 oktas) dominated conditions. This is in agreement with a recent work of Bartoszek et al. (2020) who associated higher cloudiness level with the presence of stratiform clouds. The possible role of wind on cloud type is explored in Figure S6 and in the supplement (section: Wind speed, cloudiness and clouds).

## 3.2 Cloud impact on the heating rate

### 3.2.1 The role of cloudiness

Figure 6a already provided the first indication of the important influence of clouds on the total HR. In fact, it shows the magnitude of the absolute (and relative) contribution of the diffuse component ($HR_{dif}$) with respect to the total HR revealing that, on a monthly basis, the diffuse contribution accounted on average 40±1% (of the total HR). In most cases this was comparable or even higher than $HR_{dir}$. The only exception was in November 2015 when the lowest $HR_{dif}$ (Figure 6a) and $F_{dif}$ (Figure 6b) fractions in total HR and $F_{glo}$ were measured (30.4±1.4% and 34.3±2.6% of the total, respectively), this also being the month with the lowest average cloudiness (2.91±0.06 oktas). The aforementioned data demonstrate the importance of the diffuse component of radiation. Therefore, the absolute values of the HR and its components were firstly investigated as a function of cloudiness (clear sky and complete overcast situations, seasonal averages, Figure 9a). In the wintertime clear sky, the direct component of the HR ($HR_{dir}$) was higher than $HR_{dif}$ and $HR_{ref}$ accounting for 1.35±0.04 K day$^{-1}$ and explaining on average 60±5% of the total HR. Similarly, in the springtime clear sky $HR_{dir}$ was 0.47±0.01 K day$^{-1}$ again higher than $HR_{dif}$ and $HR_{ref}$. Conversely, in complete overcast conditions (Oktas=7-8), $HR_{dif}$ dominated (84±1% of total HR) and accounted for 0.33±0.01 and for 0.19±0.01 K day$^{-1}$ during winter and spring, respectively.

In order to further investigate the role of cloudiness, we decoupled the variability of the HR induced by radiation from that due to LAA concentrations. Thus, the HR values and that of its components ($HR_{dir}$, $HR_{dif}$ and $HR_{ref}$) were normalized to the unit mass of eBC (K m$^3$ day$^{-1}$ μg$^{-1}$) and reported as a function of cloudiness in Figure 9b together with the measured irradiance ($F_{glo}$, $F_{dir}$, $F_{dif}$ and $F_{ref}$) – this parameter reports the efficiency of warming per mass concentration of eBC in different cloudiness. Overall, Figure 9b shows the general decease of HR/eBC for increasing cloud cover, a pattern also observed for both $HR_{dir}$/eBC and $HR_{ref}$/eBC which follow the respective decrease of direct and reflected irradiance. Note that at oktas values of 7-8, $HR_{dir}$/eBC reached values close to 0 (due to the suppression of $F_{dir}$ by clouds) while $HR_{ref}$/eBC was 0.03±3*10$^{-4}$ K m$^3$ day$^{-1}$ μg$^{-1}$ due to the presence of surficial albedo effect on the diffuse irradiance ($F_{dif}$). $HR_{dif}$/eBC increased with increasing cloudiness up to intermediate cloudiness conditions (5-6 oktas), reaching a maximum (0.16±0.01 K m$^3$ day$^{-1}$ μg$^{-1}$). This is in line with the behavior of the diffuse irradiance: maximum of 147±6 W m$^{-2}$ (at 5-6 oktas) doubling the value in overcast

conditions (74±3 W m$^{-2}$; 7-8 oktas) and exceeding 150% of that in clear sky (91±2 W m$^{-2}$). In overcast situations (oktas=7-8) both HR$_{dif}$/eBC and the diffuse irradiance reached their minimum due to the capability of clouds to effectively attenuate the incoming radiation. However, in these conditions, HR$_{dif}$/eBC was still not null (0.08±0.01 K m$^3$ day$^{-1}$ μg$^{-1}$) dominating the total atmospheric HR, with a contribution of 84±1%.

HR/eBC and cloudiness data were linearly related showing a high level of correlation (R$^2$=0.935, Figure S5b). Cloudiness could thus be used as good predictor (in modelling activity) for the HR/eBC.

As from Figure S5a (section 3.1), the CBH appeared related to the cloudiness, an additional linear correlation was tested between HR/eBC and CBH (Figure S5c; R$^2$=0.857); this relationship is weaker than that between HR/eBC and cloudiness. Cloudiness, describing the fraction of sky covered by clouds, is a better predictor of the capability to suppress the incoming radiation (and thus the HR promoted by LAA). The relationship between CBH and cloudiness should be also investigated in other monitoring site around the world to explore the possibility to use CBH (together with cloudiness) as a promising prognostic variable for the HR of LAA in future studies.

Overall, our experimental HR data enabled us to estimate the degree of error introduced by improperly assuming clear-sky conditions in radiative transfer calculations. Particularly, we found that the simplified assumption of clear-sky conditions leads to an overestimation of the LAA-induced HR by a factor ranging from 50 to 470% (50% in low cloudiness, oktas=1-2, 109% in moderate cloudiness, oktas=3-4, 148% in intermediate cloudiness, oktas=5-6, and 470% in cloudy conditions oktas=7-8). These results clearly highlight that clouds are responsible for an important feedback on the aerosol HR that needs to be carefully quantified, pointing to the need to correctly include and model cloudy conditions in radiative transfer calculations aimed at evaluating the real contribution of aerosol forcing on the atmospheric HR on a global scale.

### 3.2.2 Cloudiness and diurnal pattern of HR

The presence of clouds can also alter the HR diurnal pattern. Figure 10a-d shows the mean diurnal pattern of eBC, wind speed, $F_{glo}$, and HR in both clear sky (oktas=0) and cloudy conditions (oktas=7-8). In clear sky, the eBC peaked at 8:00 LST (6.41±0.31 μg m$^{-3}$) during the rush hour (Figure 10a); then eBC decreased until its minimum in the early afternoon (1.07±0.10 μg m$^{-3}$) when the wind speed reached its maximum (1.5±0.1 m s$^{-1}$, Figure 10b). The incoming $F_{glo}$ in clear sky peaked as expected at midday with 497±10 W m$^{-2}$ (Figure 10c). This caused an asymmetric HR diurnal pattern, being characterized by a fast increase to the maximum at 10:00 LST (3.60±0.18 K day$^{-1}$) and a subsequent slower decrease till sunset (Figure 10d). This pattern was not present in cloudy conditions (Figure 10d). First, eBC showed a moderate peak at 10:00 LST (4.09±0.20 μg m$^{-3}$) being quite stable during afternoon – remaining above 3 μg m$^{-3}$ until 16:00 LST (Figure 10a). The eBC behavior was consistent with that of wind speed which only slightly rose during the day, however being always below 1 m s$^{-1}$ (on average 0.64±0.03 m s$^{-1}$, Figure 10b). The incoming $F_{glo}$ in cloudy conditions peaked again as expected at midday with 103±4 W m$^{-2}$ with a much slower increase during the day (Figure 10c). The supplemental material (section: Wind speed, cloudiness and clouds) and Figure 7b show that cloudy conditions were mostly associated to stratus and very low windy conditions (0.64±0.02 m s$^{-1}$), explaining the flat diurnal behavior of eBC differing from the clear sky case. Moreover, the absence of any direct irradiance in cloudy conditions (Figure 9b; section 3.1) determines that $F_{glo}$ was essentially due to the diffuse irradiance whose symmetrical bell shape curve drove the HR behavior (Figure 10d), peaking at midday with a value of 0.74±0.01 K day$^{-1}$ (much lower than in CS).

As a conclusion, in different cloudiness conditions, not only the absolute magnitude of the HR is different, but also its diurnal pattern. This also changes the related atmospheric feedbacks, such as the influence on the liquid water content (Jacobson et al., 2002), planetary boundary layer dynamics (Wang et al., 2018; Ferrero et al., 2014), regional circulation systems (Ramanathan and Feng, 2009; Ramanathan and Carmichael, 2008), and finally on the cloud dynamic and evolution itself (Koren et al., 2008; Bond et al., 2013). Thus, an inappropriate use of clear sky assumption in models will also reflect on the modelled HR-triggered feedbacks. These results also acquire relevance in the context of the counterintuitive semi-direct effect proposed by Perlwitz and Miller (2010) and referred to in Section 1: the atmospheric heating induced by tropospheric absorbing aerosol could lead to a cloud cover increase (especially low-level clouds). Such a feedback stresses the need for a proper inclusion of sky conditions into radiative transfer calculations.

### 3.2.3    The role of cloud type

The previous sections showed the effect of cloudiness on the total LAA HR. The impact of each cloud type on the HR is addressed here as not all clouds have the same effect on irradiance (Tapakis and Charalambides, 2013). As previously done, we refer to HR values normalized to eBC unit mass (HR/eBC) to decouple radiation and aerosol effects. Figure 11a-d shows the total HR/eBC and $F_{glo}$ together with the corresponding components (HR$_{dir}$/eBC and $F_{dir}$;  HR$_{dif}$/eBC and $F_{dif}$ ; HR$_{ref}$/eBC and $F_{ref}$; Figure 11b-d). The figure shows a prefect agreement between cloud type, irradiance and the corresponding HR/eBC component ($R^2 > 0.93$; not shown). It also highlights how critical it is, for radiative transfer calculations and HR determination, to take into account the role of each cloud type. Only the cloud influence on the HR$_{dif}$/eBC is markedly different from the other components.

In terms of absolute values (not normalized for eBC), Figure 12 reveals that the HR$_{dir}$ was only dominant during periods of CS and Ci clouds (HR$_{dir}$: 1.11±0.04 and 0.92±0.05 K day$^{-1}$, respectively), explaining 66±3 and 57±4% of the total atmospheric HR. In the cases of other clouds (St, As and Sc) HR$_{dif}$ dominates, reaching the highest absolute contribution of 84.4±3.8, 83.0±10.7 and 76±4% (HR$_{dif}$: 0.25±0.01, 0.34±0.03 and 0.66±0.02 K day$^{-1}$), respectively.

Given this impact of cloud type, the ability of cloudiness to be a good predictor for the HR (as detailed in section 3.2.1) and the relationship (over the investigated site) between cloudiness and cloud type (section 3.1, Figure 7b), the synergic impact of cloudiness and cloud type on HR was investigated and presented in Figure 13. In the figure, we summarize the HR results in terms of percent difference from the clear sky (CS) case by averaging the cloudiness (in oktas) for each cloud type (as detected in section 3.3). Overall, the derived linear regression indicates a HR decrease of -11.9±1.2% per okta. The regression $R^2$ (0.963) was slightly higher than that reported in Figure S5b ($R^2$=0.935; relationship with the cloudiness only) suggesting the need (for precise calculations) to account for the cloud types responsible for any sky coverage in agreement with a recent work of Bartoszek et al. (2020). Figure 13 also allowed us to associate the HR decrease to each specific cloud type over Milan. Particularly, Ci were found to produce a modest impact on cloudiness (0.50±0.05 oktas) decreasing the HR by ~3%, while Cu (1.76±0.09 oktas) decrease the HR by -26±8%. Cc-Cs (oktas of 3.56±0.14) were responsible for a -49±6 decrease of the HR. Their impact was comparable to that of Sc (4.68±0.10 oktas, -48±4% of HR). Ac (4.11±0.18 oktas) had a higher impact, decreasing the HR by -59±6%. The highest impact was due to As (6.57±0.15 oktas; -76±4% of HR) and by St (oktas: 7.19±0.04) that suppressed the HR by a factor of -83±4%.

### 3.3 The impact of clouds on the BC and BrC heating rates

In this last part of the work we focus on the HR of the two main absorbing aerosol species: BC and BrC (obtained as detailed in section 2.1.1). The monthly averaged values of HR of BC and BrC ($HR_{BC}$ and $HR_{BrC}$) are reported in Figure 14. The highest $HR_{BC}$ and $HR_{BrC}$ values were recorded in December ($1.24\pm0.03$ K day$^{-1}$ and $0.19\pm0.01$ K day$^{-1}$) while the lowest were recorded in March ($0.46\pm0.01$ K day$^{-1}$ and $0.07\pm0.01$ K day$^{-1}$). Overall, $HR_{BrC}$ accounted for $13.7\pm0.2\%$ of the total HR.

The variability of total $HR_{BC}$ and $HR_{BrC}$ as a function of cloudiness is reported in Figure 15a, with panels b-d showing their direct ($HR_{BC,dir}$ and $HR_{BrC,dir}$), diffuse ($HR_{BC,dif}$ and $HR_{BrC,dif}$) and reflected ($HR_{BC,ref}$ and $HR_{BrC,ref}$) components. Figure 15a shows that both $HR_{BC}$ and $HR_{BrC}$ decreased with increasing cloudiness, going from the CS maxima ($HR_{BC}$ and $HR_{BrC}$: $1.14\pm0.03$ and $0.20\pm0.01$ K day$^{-1}$) to the completely overcast conditions (oktas=8) minima of $0.16\pm0.01$ and $0.02\pm10^{-3}$ K day$^{-1}$ (mainly due to St and As clouds; see Figure 7b). As shown in Figure 9b, the change of irradiance magnitude with cloudiness was different for direct, diffuse and reflected components affecting the corresponding direct, diffuse and reflected components of $HR_{BC}$ and of $HR_{BrC}$ (Figure 15b-d). $HR_{BC,dir}$ and $HR_{BrC,dir}$ (Figure 15b) decreased as a function of cloudiness from $0.74\pm0.03$ and $0.11\pm0.01$ K day$^{-1}$ (oktas=0) to negligible levels (HR<$10^{-4}$ K day$^{-1}$) in completely overcast conditions. $HR_{BC,dif}$ and $HR_{BrC,dif}$ (Figure 15c) increased with cloudiness, reaching their maximum in partially cloudy conditions (at oktas=6, $0.51\pm0.01$ and $0.09\pm0.01$ K day$^{-1}$). Further increasing cloudiness reduced their values to minimum values ($0.13\pm0.01$ and $0.02\pm0.01$ K day$^{-1}$). $HR_{BC,ref}$ and $HR_{BrC,ref}$ (Figure 15d) behave similarly to the total $HR_{BC}$ and $HR_{BrC}$, since the reflected irradiance is dominated by the global irradiance impinging on the ground (see Figure 9b for a comparison); $HR_{BC,ref}$ and $HR_{BrC,ref}$ decreased with increasing oktas from maximum values in clear sky ($HR_{BC,ref}$ and $HR_{BrC,ref}$: $0.17\pm4*10^{-3}$ and $0.03\pm1*10^{-3}$ K day$^{-1}$) down to overcast minimum ($HR_{BC,ref}$ and $HR_{BrC,ref}$ $0.02\pm10^{-3}$ and $3*10^{-3}\pm10^{-3}$ K day$^{-1}$). Figure 15a-d also shows that $HR_{BC}$ was always greater (in absolute values) than $HR_{BrC}$, as expected. The relative decrease of $HR_{BrC}$ from CS to complete overcast conditions was $12\pm6\%$ larger with respect to that of $HR_{BC}$. At a first glance, Figure 15a-d could give the impression that BrC is more efficient in heating the surrounding atmosphere (with respect to BC) in CS conditions. However, any change of both BC and BrC $b_{abs}(\lambda)$ in different sky conditions has to be taken into account to avoid any misinterpretation of the results. While the variability of BC $b_{abs}(\lambda)$ with cloudiness was limited (with the exception of oktas=1, Figure S7a), this was not the case for BrC. In fact, $b_{abs(\lambda)}$ BrC values in high cloudiness were statistically lower than the ones in CS (at oktas=8, $b_{abs(\lambda)}$ of BrC was -23±3% lower than in CS, Figure S7b). The relative decrease of the $HR_{BrC}$ with cloudiness was therefore higher compared to that of $HR_{BC}$. Understanding of the reason behind the observation of higher $b_{abs}(\lambda)$ values for BrC in CS is beyond the aim of the present paper (we can speculate it could be related to the formation of secondary BrC at high radiation levels, e.g., Kumar et al., 2018).

Here we focus on the fact that the magnitude of $b_{abs(\lambda)}$ of BC and BrC changed differently with cloudiness. Thus, in order to decouple the variability of the HR induced by the varying incoming irradiance from that due to changes in $b_{abs}(\lambda)$, both $HR_{BC}$ and $HR_{BrC}$ were normalized to the dimensionless integral of the $b_{abs}(\lambda)$ over the whole aethalometer spectrum. In this way, the magnitude of $b_{abs}(\lambda)$ is accounted for along the whole spectrum avoiding the choice of an arbitrary wavelength as a reference for the normalization. Similarly to section 3.2.2 for the total of LAA HR, the variability of the normalized $HR_{BC}$ and $HR_{BrC}$ was investigated with respect to cloudiness and cloud type; in this respect, both $HR_{BC}$ and $HR_{BrC}$ were normalized to the dimensionless integral of the $b_{abs}(\lambda)$ for each cloud type. Figure 16a shows the decrease of normalized $HR_{BC}$ and $HR_{BrC}$ as a function of average cloudiness

for each cloud type. We found a strong linear relationship between the decrease of both normalized $HR_{BC}$ and $HR_{BrC}$ (relative to CS) and the mean cloudiness (in okta) for each cloud type. Focusing on the cloud type, Ci were found to produce a statistically negligible impact on cloudiness (0.50±0.05 oktas) decreasing the $HR_{BC}$ and $HR_{BrC}$ by ~1-6%, respectively. Cu (1.76±0.09 oktas) decreased the $HR_{BC}$ and $HR_{BrC}$ by -31±12% and -26±7%, respectively. Cc-Cc featured oktas of 3.56±0.14, and were responsible for a -60±8% and -54±4% decrease of the $HR_{BC}$ and $HR_{BrC}$. Their impact was comparable to that of Ac (4.11±0.18 oktas): -60±6% and -46±4% decrease of the $HR_{BC}$ and $HR_{BrC}$. Sc (4.68±0.10 oktas) had a higher impact, decreasing $HR_{BC}$ and $HR_{BrC}$ of -63±6% and -58±4%. The highest impact was given by As (6.57±0.15 oktas; -78±5% and -73±4% of $HR_{BC}$ and $HR_{BrC}$) and by St (oktas: 7.19±0.04) suppressing the $HR_{BC}$ and $HR_{BrC}$ by -85±5% and -83±3%, respectively.

Overall, the derived linear regressions indicate a decrease of ~12% per oktas for both $HR_{BC}$ and $HR_{BrC}$ (with high $R^2$: 0.958 and 0.963, respectively). In details, the respective decreases of $HR_{BC}$ and $HR_{BrC}$ were -11.8±1.2% and -12.6±1.4% per okta, these values not being statistically different. We show that, while BC and BrC have different optical properties and wavelength dependence of absorption, their HR normalized to absorption, changed without any statistical difference as a function of cloudiness and cloud type. This simplifies the models and reduces the number of details needed to be considered: once $HR_{BC}$ and $HR_{BrC}$ are determined in clear sky conditions, their dependence on the cloudiness can be determined from the simple reduction of the HR normalized to the absorption coefficient (about 12% for both species, once dominant cloud type is known).

However, it noteworthy that normalized $HR_{BrC}$ values in Figure 16 were always greater or equal to the corresponding ones of BC (even if 95% confidence interval bands overlapped). A possible explanation can be the synergic effect between the different spectral absorption of BC and BrC and the influence of clouds on the energy of the impinging radiation; this is detailed in the Supplement (section: The role of average photon energy on the HR of BC and BrC). This feature needs further investigation in other seasons and elsewhere the world where the prevailing clouds type and the light absorption by BrC might be different.

### Summary and conclusions

The heating rates (HR) associated to the two major LAA species, i.e., Black Carbon (BC) and Brown Carbon (BrC) ($HR_{BC}$ and $HR_{BrC}$) were experimentally determined based on high time resolution radiation and aerosol measurements in the Po Valley. We determined the impact of cloud-aerosol-radiation interactions on the atmospheric heating by examining the total HR in different sky conditions. Results showed a constant decrease of LAA HR with increasing cloudiness of the atmosphere (~12%). Our real-atmosphere, all-sky, measurement-based results suggest that using a simplified assumption of clear sky in radiative transfer calculations might overestimate the HR by over 400%. The effect of different cloud types on the HR was also investigated. While cirrus were characterized by a modest impact, cumulus, cirrocumulus-cirrostratus and altocumulus suppressed the HR of both BC and BrC by a factor of ~2. Stratocumulus, altostratus stratus suppressed the $HR_{BC}$ and $HR_{BrC}$ up to 80%. The cloudiness also changed the diurnal pattern of HR with possible feedbacks on planetary boundary layer dynamics and/or regional circulation systems.

Total HR, $HR_{BC}$ and $HR_{BrC}$ are affected by both cloudiness and cloud type so that inaccurate $HR_{BC}$ and $HR_{BrC}$ estimations can be derived from simulations if presence of clouds is ignored and cloud type is not taken into account. Most important, the coupling between the cloud impact on the solar radiation spectrum (and its direct, diffuse and reflected components) and the spectral absorption properties of BC and BrC showed that the absolute

HR$_{BC}$ and HR$_{BrC}$ vary differently with cloudiness (especially the diffuse component), but feature a very similar normalized (to the absorption coefficient) dependence on the cloudiness. This simplifies the models and reduces the number of details that need to be considered: once HR$_{BC}$ and HR$_{BrC}$ are determined in clear sky conditions, their dependence on the cloudiness can be determined from the simple reduction of the HR normalized to the absorption coefficient (about 12% for both species). These data acquire importance when discussed in the context of the counterintuitive semi-direct effect proposed by Perlwitz and Miller (2010): the atmospheric heating induced by tropospheric absorbing aerosol could lead to a cloud cover increase stressing the needs for a proper determination and simulation of sky conditions during radiative transfer calculations.

**Acknowledgements**

This paper is an output of the GEMMA center in the framework of the MIUR project "Dipartimenti di Eccellenza 2018-2022". The work was in part funded by the Slovenian Research Agency program P1-0385 "Remote sensing of atmospheric properties". The authors want to acknowledge the COST action COLOSSAL CA16109, Aerosol d.o.o and LSI-Lastem for the cooperation during the campaign.

**Data Availability**

Data are available upon request.

**Author contribution**

Conceptualization, L. Ferrero, E. Bolzacchini, Griša Močnik, Martin Rigler and A. Gregorič; Methodology, L. Ferrero, A. Gregorič, S. Cogliati, N. Losi, F. Barnaba, L. Di Liberto, G.P. Gobbi; Data Investigation, L. Ferrero, A. Gregorič, S. Cogliati, F. Barnaba, G. Močnik, M. Rigler; Resources, E. Bolzacchini, M. Rigler; Original Draft Preparation, L. Ferrero; Writing-Review & Editing, L. Ferrero, A. Gregorič, G. Močnik and F. Barnaba; Supervision, G. Močnik, Martin Rigler, F. Barnaba, E. Bolzacchini.

**Conflict of interest**

The authors declare no conflict of interest. The funders had no role in the design of the study; in the collection, analyses, or interpretation of data; in the writing of the manuscript, and in the decision to publish the results.

**Appendix A: Nomenclature**

**Nomenclature**

*Aerosol Acronyms*

| | |
|---|---|
| AAE | Absorption Angstrom Exponent |
| AAE$_{BC}$ | Absorption Angstrom Exponent of Black Carbon |
| AAE$_{BrC}$ | Absorption Angstrom Exponent of Brown Carbon |
| ADRE | Absorptive Direct Raditive Effect |
| $b_{abs(\lambda)}$ | wavelength dependent aerosol absorption coefficient |
| BC | Black Carbon |
| BrC | Brown Carbon |

| | |
|---|---|
| eBC | equivalent Black Carbon concentration |
| LAA | Light Absorbing Aerosol |
| HR | Heating Rate |
| $HR_{BC}$ | Heating Rate of Black Carbon |
| $HR_{BrC}$ | Heating Rate of Brown Carbon |

*Cloud/Sky Acronyms*

| | |
|---|---|
| As | Altostratus |
| Ac | AltoCumulus |
| Ci | Cirrus |
| Cc-Cs | Cirrocumulus-Cirrostratus |
| Cu | Cumulus |
| CS | Clear Sky |
| St | Stratus |
| Sc | Stratocumulus |
| CBH | Cloud Base Height (km) |
| N | classes of sky conditions in oktas (from 0, i.e. clear sky, to 8, i.e. complete overcast) |
| R | ratio (R) between observed global irradiance (Fglo) and the modelled clear sky irradiance (GHI) |
| SD | standard deviation of the measured $F_{glo}$ in 20 minute time intervals |

*Other Symbols/Acronyms*

| | |
|---|---|
| $\phi$ | Azimuth angle |
| $\Phi_\lambda$ | photon flux density at wavelength $\lambda$ |
| $\lambda$ | Wavelength |
| $\rho$ | Air Density |
| $\theta$ | Zenith angle |
| $\theta_z$ | Solar zenith angle |
| a | empirical coefficient from Ehnberg and Bollen (2005); Table S1 |
| $a_0$ | empirical coefficient from Ehnberg and Bollen (2005); Table S1 |
| $a_1$ | empirical coefficient from Ehnberg and Bollen (2005); Table S1 |
| $a_3$ | empirical coefficient from Ehnberg and Bollen (2005); Table S1 |
| $AF(\lambda)$ | Actinic flux for wavelength $\lambda$ |
| APE | Average Photon Energy |
| $APE_{dif}$ | Average Photon Energy for diffuse radiation |
| $APE_{dir}$ | Average Photon Energy for direct radiation |
| $APE_{ref}$ | Average Photon Energy for reflected radiation |
| $c$ | speed of light (m s$^{-1}$) |
| $C_p$ | Isobaric specific heat of dry air (1005 J kg$^{-1}$ K$^{-1}$) |
| dif | diffuse |

| | |
|---|---|
| dir | direct |
| $F_{glo}$ | Global Irradiance; $F_{glo}= F_{dir} + F_{dif}$ |
| $F_{dif}$ | Diffuse Irradiance |
| $F_{dir}$ | Direct Irradiance |
| $F_{ref}$ | Reflected Irradiance |
| $F_{dir,dif,ref}(\lambda)$ | Spectral irradiance in function of $\lambda$ |
| $h$ | Plank constant (J s) |
| ref | reflected |
| L | empirical coefficient from Ehnberg and Bollen (2005); Table S1 |
| q | Electron charge |
| $R(\lambda,\theta,\phi)$ | Radiance at wavelength $\lambda$ from zenith and azimuth angles $\theta$ and $\phi$ |

**Appendix B: Cloud type validation**

The validation was conducted in two subsequent steps. In the first step the automatized cloud classification (based on Duchon and O'Malley, 1999 including lidar cloud base height) was compared to the visual cloud classification based on sky images collected during 1 month of field campaign.

The second validation step involved the recently published method discussed by Ylivinkka et al. (2020) which is based on the same methodological approach used in this study: the application of Duchon and O'Malley (1999) classification improved by the knowledge of the CBH. Thus, the aim of the second step was to determine the degree of consistency between the two approaches that were developed simultaneously and independently in two different regions of the globe.

Both the two validations were evaluated by means of a confusion matrix, a special kind of contingency table, with two dimensions and identical sets of "classes" in both of them. From the confusion matrix the balanced accuracy was computed as follows:

$$Balanced\ Accuracy = \frac{Sensitivity+Specificity}{2} \tag{B1}$$

where the *Sensitivity* describes the true positive rate (the number of correct positive predictions divided by the total number of positives) and the *Specificity* describes the true negative rate (the number of correct negative predictions divided by the total number of negatives). The balanced accuracy is especially useful when the investigated classes are imbalanced, i.e. one of the classes appears a lot more often than the other, a condition useful for cloud classification (García et al., 2009).

*Appendix B1: visual cloud classification*

Sky images were collected during 1 month (13 February – 9 March 2017) using a sky view camera (GoPro Hero4 Session installed on the U9 roof) characterized by a field of view of 95x123°; the camera was oriented south each day manually with the same declination of the shadow band applied to DPA154 global radiometer for diffuse broadband irradiance measurements (section 2.1.2); sky images were taken with 1 minute time resolution. Visual classification of sky images, based on the  principles of cloud classification published in Cloud Atlas (WMO). Figure B1 reports an example of SD-R diagram (section 2.3.2) with CBH for each sky/cloud conditions with the corresponding image.

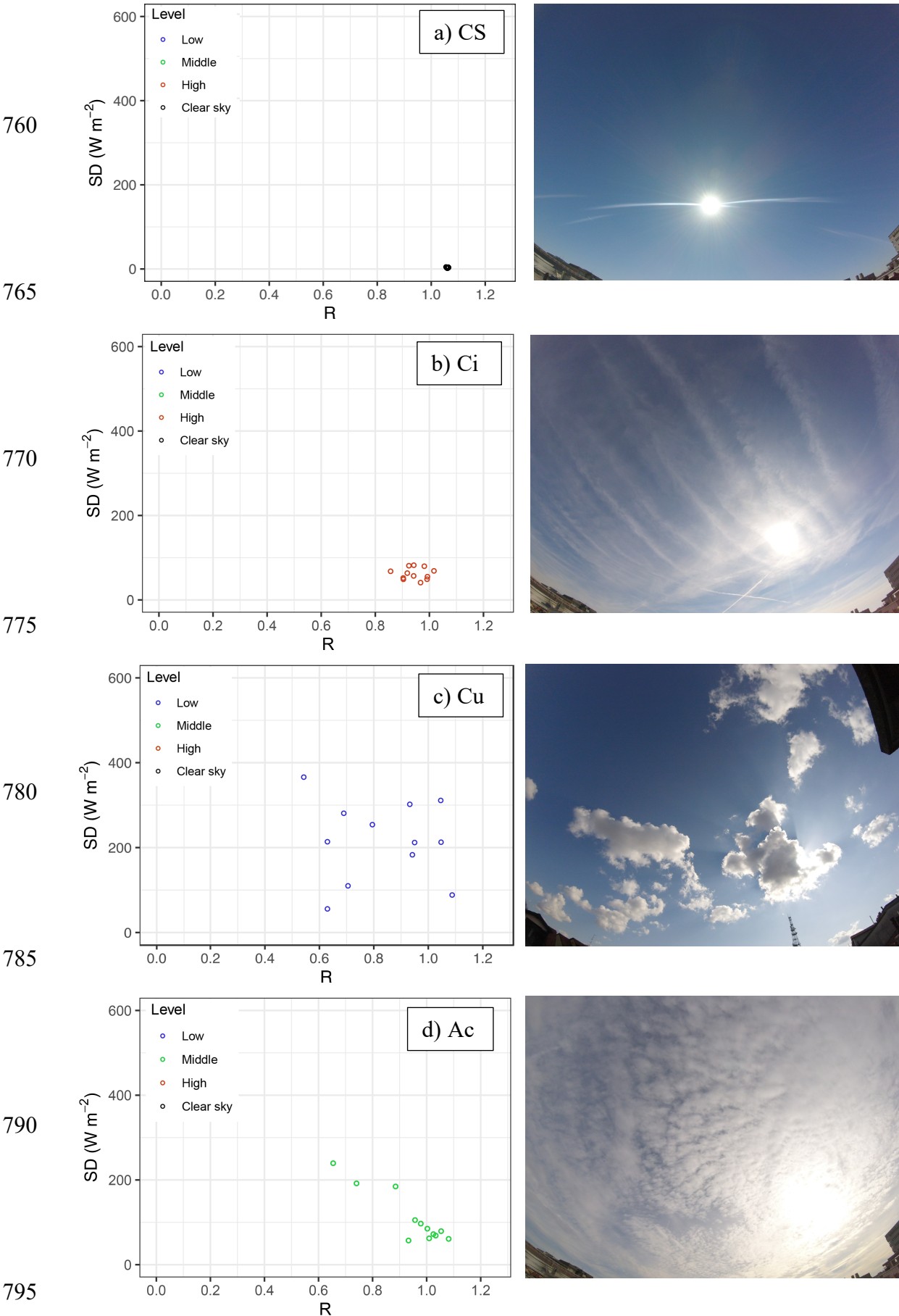

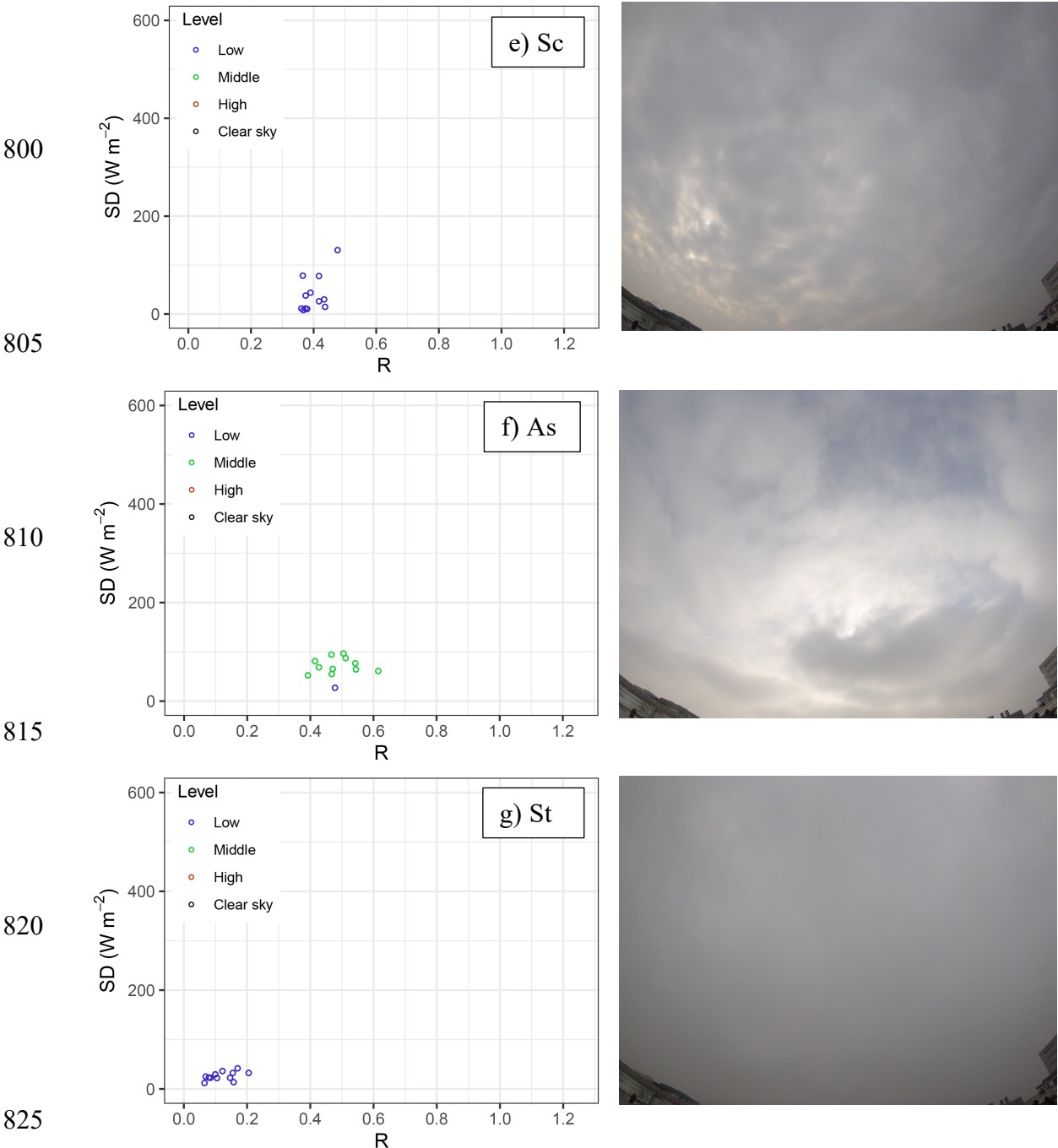

Figure B1. SD-R diagram (left panel) and the corresponding sky images for the February-March 2017 field campaign: a) CS case, b) Ci clouds case, c) Cu clouds case, d) Ac clouds case, e) Sc clouds case, f) As clouds case and g) St clouds case.

To test the performance, 869 sky images were analyzed, and the cloud type was determined through visual inspection. From the visual classification and the automatized one (Table 1) the following confusion matrix (Table B1) was created. The highest balanced accuracy was found for St data (95%) while the lowest (50%) for mixed cloud types (Cc-Cs) whose absolute number of cases, however, was ~0.6% of the total, probably biasing the obtained accuracy; the same happened for Cu and Ac. Overall, five classes over eight were above 68% of balanced

accuracy while the overall balanced accuracy was 80%, underlying the reliability of the classification algorithm allowing to study the impact of clouds on LAA HR with a sufficient grade of certainty.

| | Cloud type | Visual classification (reference) | | | | | | | | Balanced accuracy |
| --- | --- | --- | --- | --- | --- | --- | --- | --- | --- | --- |
| | | Cu | St | Sc | Ac | As | Ci | Cc-Cs | CS | |
| Cloud classification algorithm | Cu | **6** | 2 | 7 | 1 | | 2 | | 9 | 59% |
| | St | 1 | **259** | 25 | | 10 | | | | 95% |
| | Sc | 7 | 9 | **61** | 1 | | | | 15 | 81% |
| | Ac | | | | **1** | 4 | | | | 62% |
| | As | | 3 | | | **23** | | | | 81% |
| | Ci | | | | | | **45** | 4 | 10 | 70% |
| | Cc-Cs | | | | | | 3 | **0** | | 50% |
| | CS | 16 | | | 1 | | 56 | 1 | **287** | 89% |

Table B1. Confusion matrix and balanced accuracy for each cloud type classified visually and following the algorithm reported in Table 1 within the present work.

*Appendix B2: intercomparison with Ylivinkka et al. (2020)*

The second validation step involved the recently published method discussed by Ylivinkka et al. (2020), which is based on the same logical approach followed in our work: the application of Duchon and O'Malley (1999) classification improved by the knowledge of the CBH. At this purpose, the classification scheme of Ylivinkka et al. (2020) is resumed in Table B2 following the nomenclature used in the present work. It is necessary to underline that the cloud classes determined in the work Ylivinkka et al. (2020) differ from those reported in the present work. Particularly, while both approaches enabled the Cu, St, Sc classification, some of the cloud classes were merged in the Ylivinkka et al. (2020) study: CS and Ci (CS+Ci), Ac and As (Ac+As) and mixed situation composed by Ci, Cc, Cs (Ci+Cc+Cs). In addition they introduced the classes Cu+GRE and Ci+GRE to account for global radiation enhancement (GRE) due to this cloud types; a possible explanation for such difference with respect to present work could be hidden in the different latitude at which the two algorithms were developed, a parameter able to affect the solar zenith angle and the sun light interaction with clouds. A detailed investigation of this difference is beyond the aim of the present work. However, it is necessary to account for the classification differences in order to properly merge cloud classes with similar features to finally perform a comparison between the two methods. The cloud classes homogenization is summarized in Table B3 while the final intercomparison is reported in Table B4. The confusion matrix (Table B4) revealed a global balanced accuracy of 90% making the two methods comparable, despite the aforementioned differences. The highest accuracy (100%) was obtained for CS followed by Ac+As (99%); Cu, St and Sc reached values of 94, 93 and 86%, respectively. The lowest performance was reached for Ns whose presence cannot be detected in the present study generating a false positive signal in the Ac+As class; however, due to the very low number of Ns cases (1.8%), its impact on the cloud classification can be neglected. Overall, also the second validation step pointed out the reliability of the results obtained in the present work.

| Cloud type | CBH (m) | R | SD (W/m2) | N of cloud layers |
|---|---|---|---|---|
| Cu | < 2000 | 0.6 – 0.85 & Rmax > 1 | >= 200 | 1 |
| | < 2000 | > 0.85 & Rmax > 1 | 0 – 200 | 1 |
| St | < 2000 | < 0.6 | < 100 | 1 |
| Sc | < 2000 | 0.1 – 0.6 | >= 100 | 1 |
| Ns | 2000 - 3000 | < 0.3 | < 100 | 1 |
| Ac+As | 2000 - 5000 | >=0.3 | < 500 | 1 |
| Ci+Cc+Cs | >= 4000 | 0.85 – 1.1 | 50 - 400 | 1 |
| | >= 4000 | 0.5 – 0.85 | < 400 | 1 |
| CS+Ci | NaN | 0.85 – 1.05 | < 50 | 1 |
| Cu+GRE | < 2000 | > 1 & Rmax > 1 | >= 200 | 1 |
| Ci+GRE | >=4000 | > 1 | < 400 | 1 |

Table B2. Final criteria adopted for cloud classification in Ylivinkka et al. (2020). Ns here represents Nimbostratus while GRE global enhancement radiation.

| This study | Cu | St | Sc | / | Ac, As | Ci Cc-Cs | CS |
|---|---|---|---|---|---|---|---|
| Ylivinkka et al., 2020 | Cu, Cu+GRE | St | Sc | Ns | Ac+As | Ci+Cc+Cs Ci+GRE | CS+Ci |
| Merged Cloud type | Cu | St | Sc | Ns | Ac+As | Ci+Cc+Cs | CS+Ci |

Table B3. Cloud classes homogenization adopted for comparison purposes between the present study cloud classification and the one reported in Ylivinkka et al. (2020).

| Cloud type classification | | Ylivinkka et al. (2020) | | | | | | | Balanced accuracy |
|---|---|---|---|---|---|---|---|---|---|
| | | Cu | St | Sc | Ns | Ac+As | Ci+Cc+Cs | CS+Ci | |
| This study | Cu | 80 | | | | | | | 94% |
| | St | | 3853 | 58 | | 1 | | | 93% |
| | Sc | 11 | 596 | 231 | | | | | 86% |
| | Ns | | | | 0 | | | | 50% |
| | Ac+As | | | | 153 | 383 | 51 | | 99% |
| | Ci+Cc+Cs | | | | | | 846 | | 97% |
| | CS+Ci | | | | | | | 2142 | 100% |

Table B4. Confusion matrix and balanced accuracy for each cloud type classified using the algorithm reported in the present study and the one reported in Ylivinkka et al. (2020).

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

| Level | Cloud type | SD | R | cloud layer |
|---|---|---|---|---|
| | Stratus (St) | <120 | 0.0-0.4 | 1 |
| Low (<2 km) | Cumulus (Cu) | / | 0.8-1.1 | 1 |
| | Stratocumulus (Sc) | / | 0.4-0.8 | 1 |
| Middle (2-7 km) | Altostratus (As) | <120 | 0.0-0.4 | 1 |
| | Altocumulus (Ac) | >120 | 0.4-0.8 | 1 |
| High (>7 km) | Cirrus (Ci) | / | 0.8-1.1 | 1 |
| | Cirrocumulus-Cirrostratus (Cc-Cs) | / | 0.0-0.8 | 1 |
| | Clear Sky (CS) | / | / | 0 |

Table 1. Final criteria adopted for cloud classification. SD represents the standard deviation of the measured global
irradiance with respect to the theoretical behavior in clear sky conditions; R represents the ratio between observed
global irradiance ($F_{glo}$) and the modelled irradiance ($F_{glo\_CS}$) in clear sky conditions; cloud layer: is the number of
cloud layers detected by the lidar.

| Month | Metric | T | P | eBC* | $b_{abs}$* | ADRE | $ADRE_{dir}$ | $ADRE_{dif}$ | $ADRE_{ref}$ | HR | $HR_{dir}$ | $HR_{dif}$ | $HR_{ref}$ | $F_{glo}$ | $F_{dir}$ | $F_{dif}$ | $F_{ref}$ |
|---|---|---|---|---|---|---|---|---|---|---|---|---|---|---|---|---|---|
| | | °C | hPa | ng m$^{-3}$ | Mm$^{-1}$ | mW m$^{-3}$ | mW m$^{-3}$ | mW m$^{-3}$ | mW m$^{-3}$ | K day$^{-1}$ | K day$^{-1}$ | K day$^{-1}$ | K day$^{-1}$ | W m$^{-2}$ | W m$^{-2}$ | W m$^{-2}$ | W m$^{-2}$ |
| Nov-15 | mean | 12.8 | 1003.8 | 4288 | 21.2 | 18.42 | 10.17 | 5.62 | 2.64 | 1.30 | 0.72 | 0.40 | 0.19 | 200 | 131 | 69 | 51 |
| | CI 95% | 0.2 | 0.3 | 96 | 0.5 | 0.61 | 0.44 | 0.18 | 0.08 | 0.04 | 0.03 | 0.01 | 0.01 | 5 | 1 | 5 | 1 |
| Dec-15 | mean | 8.4 | 1012.8 | 6289 | 31.1 | 20.70 | 9.29 | 8.64 | 2.77 | 1.43 | 0.64 | 0.59 | 0.19 | 141 | 66 | 75 | 34 |
| | CI 95% | 0.1 | 0.1 | 97 | 0.5 | 0.68 | 0.48 | 0.24 | 0.08 | 0.05 | 0.03 | 0.02 | 0.01 | 4 | 2 | 3 | 1 |
| Jan-16 | mean | 7.2 | 997.4 | 4198 | 20.8 | 12.57 | 5.53 | 5.26 | 1.79 | 0.87 | 0.38 | 0.36 | 0.12 | 150 | 85 | 65 | 36 |
| | CI 95% | 0.2 | 0.4 | 106 | 0.5 | 0.55 | 0.36 | 0.23 | 0.07 | 0.04 | 0.02 | 0.02 | 0.01 | 5 | 2 | 5 | 1 |
| Feb-16 | mean | 9.2 | 995.5 | 2851 | 14.1 | 8.62 | 3.50 | 3.81 | 1.31 | 0.61 | 0.25 | 0.27 | 0.09 | 191 | 104 | 87 | 46 |
| | CI 95% | 0.1 | 0.3 | 74 | 0.4 | 0.35 | 0.23 | 0.14 | 0.05 | 0.02 | 0.02 | 0.01 | 0.00 | 6 | 3 | 6 | 2 |
| Mar-16 | mean | 12.6 | 996.2 | 1535 | 7.6 | 7.58 | 2.96 | 3.28 | 1.34 | 0.54 | 0.21 | 0.23 | 0.10 | 310 | 174 | 136 | 77 |
| | CI 95% | 0.1 | 0.2 | 36 | 0.2 | 0.22 | 0.14 | 0.08 | 0.04 | 0.02 | 0.01 | 0.01 | 0.00 | 7 | 3 | 7 | 2 |

Table 2. Monthly averaged data and confidence interval at 95% of temperature (T), pressure (P), equivalent black
carbon (eBC), absorption coefficient ($b_{abs}$), absorptive direct radiative effect (ADRE) together with the heating
rate (HR) divided into their direct (dir), diffuse (dif) and reflected (ref) components and, finally, global ($F_{glo}$),
direct ($F_{dir}$), diffuse ($F_{dif}$) and reflected ($F_{ref}$) irradiances. * denotes Aethalometer data referred to λ=880 nm.

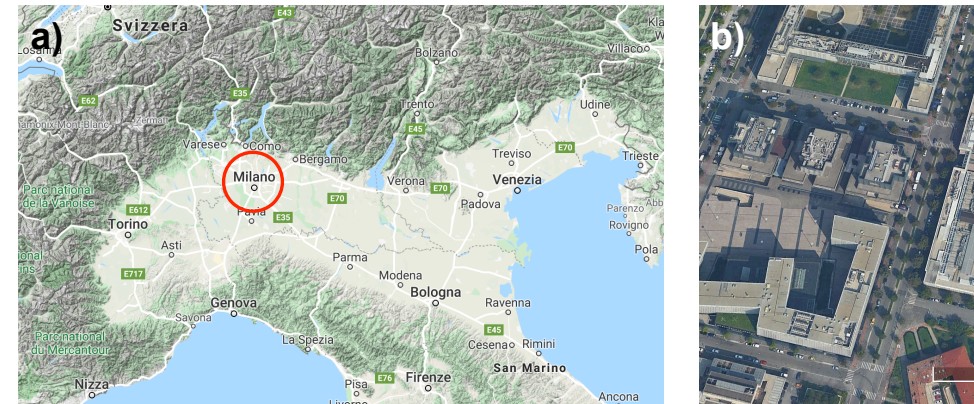

Figure 1. a) location of the Milano sampling site in the Po Valley, Italy; b) the U9 sampling site on the rooftop (10 m agl) of the University of Milano-Bicocca. The copyright holder of Figure 1 is GoogleMaps (©GoogleMaps).

Dati cartografici ©2020 Google,GeoBasis-DE/BKG (©2009),Inst. Geogr. Nacional      50 km

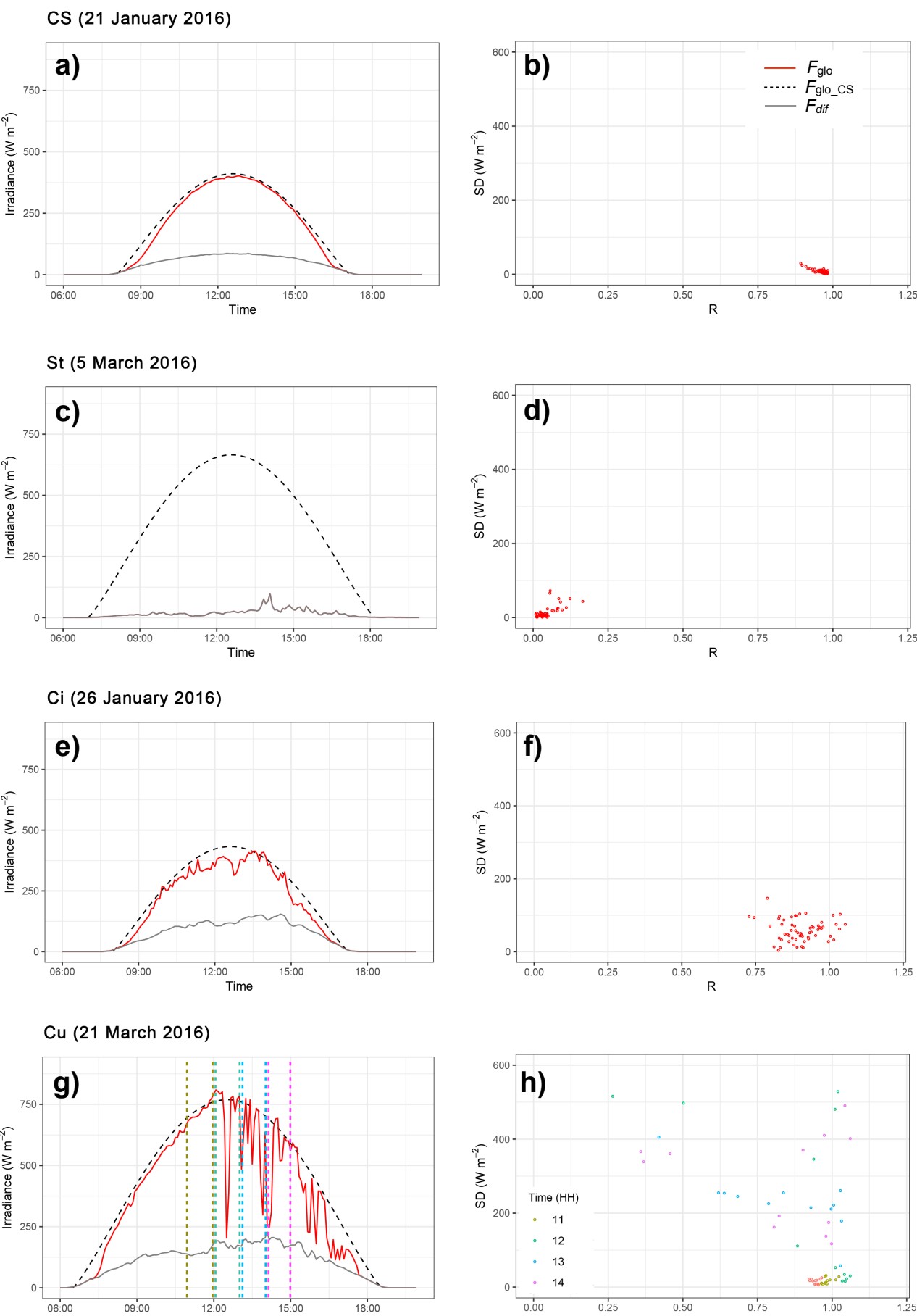

Figure 2. Cloud classification based on broadband solar radiation following Duchon & O'Malley (1999). Each row represents a different clout type in a specific day as a case study. The left columns represent the time series of global and diffuse measured solar irradiance ($F_{glo}$ and $F_{dif}$) and modelled clear sky irradiance ($F_{glo\_CS}$), while the right column the scatter SD-R plot of the observed standard deviation of irradiance (SD) vs. the fraction of modelled clear sky irradiance (R). In the panel (h) different colors are related to different time (hours) of the day as reported in the legend.

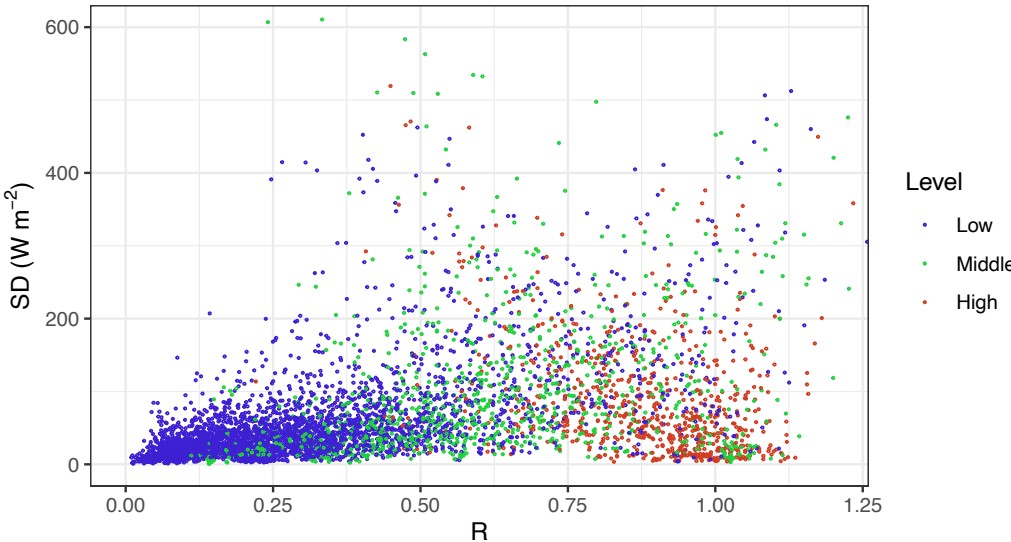

Figure 3. SD-R plot of the whole dataset concerning the cloud base altitude grouped into three levels, namely Low level clouds (<2 km), Middle altitude clouds (2-7 km) and High-altitude clouds (>7 km). The rectangular boxes (following the works of Duchon & O'Malley (1999) and of Harrison et al. (2008)) refers qualitatively to the areas dominated by stratiform clouds (blue rectangle), stratocumulus clouds (red rectangle), clear sky and cirrus clouds (yellow rectangle) and cumulus clouds (black rectangle).

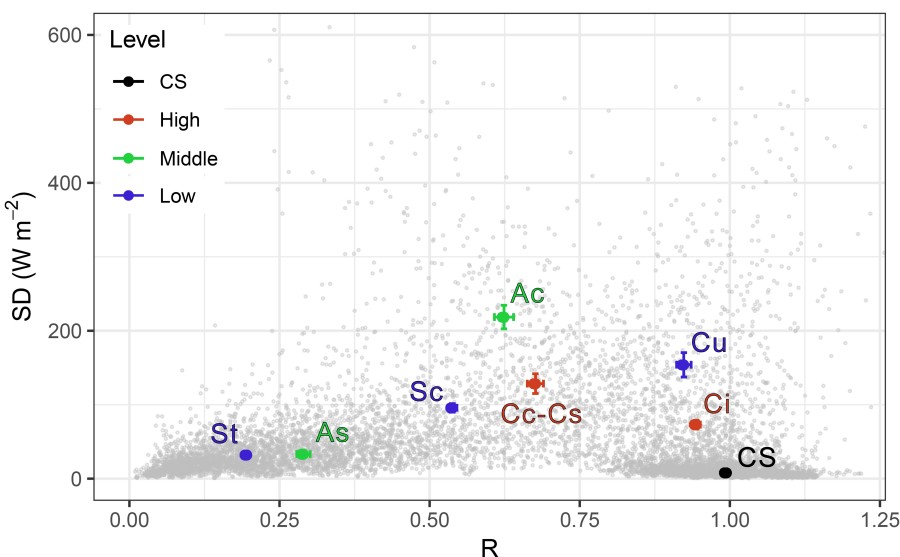

Figure 4. Cloud classification based on the improved broadband solar radiation following Duchon & O'Malley (1999) and Harrison et al. (2008) coupled with lidar data of cloud base height. From left to right: Stratus (St), Altostratus (As), Stratocumulus (Sc), Altocumulus (Ac), Cirruscumulus and Cirrusstratus (Cc-Cs), Cumulus (Cu), Cirrus (Ci), and finally clear-sky (CS). The SD-R plot reports in grey the single data of the whole dataset, while centroids and 99% confidence bound of each cloud type are plotted in a color scale related to the cloud base level.

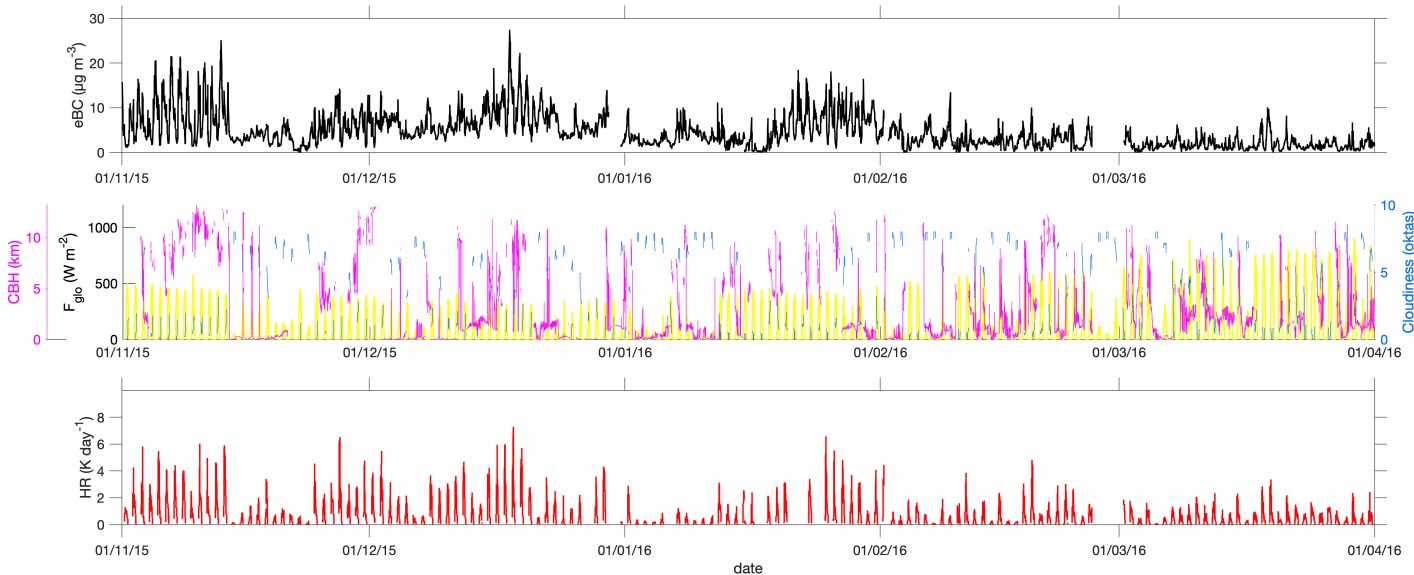

Figure 5. High time resolution data (5-min) for eBC, global irradiance ($F_{glo}$, yellow line) cloud base height (CBH), cloudiness (oktas) and the related heating rate (HR) from 1 November 2015 to 1 April 2016.

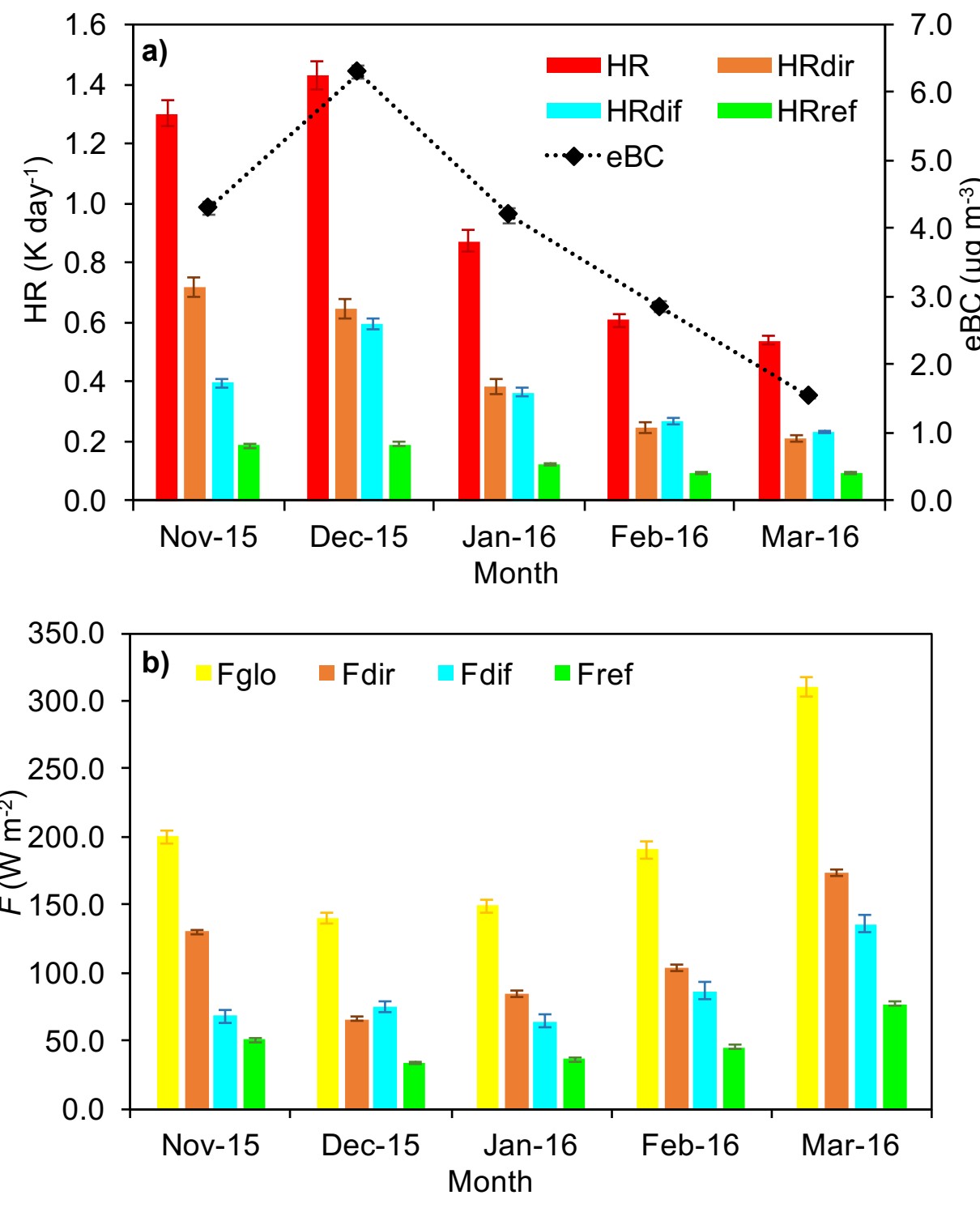

Figure 6. Monthly averaged values of: a) eBC, HR values and their direct, diffuse and reflected components (HR$_{dir}$, HR$_{dif}$ and HR$_{ref}$); b) global radiation values ($F_{glo}$) and their direct, diffuse and reflected components ($F_{dir}$, $F_{dif}$ and $F_{ref}$).

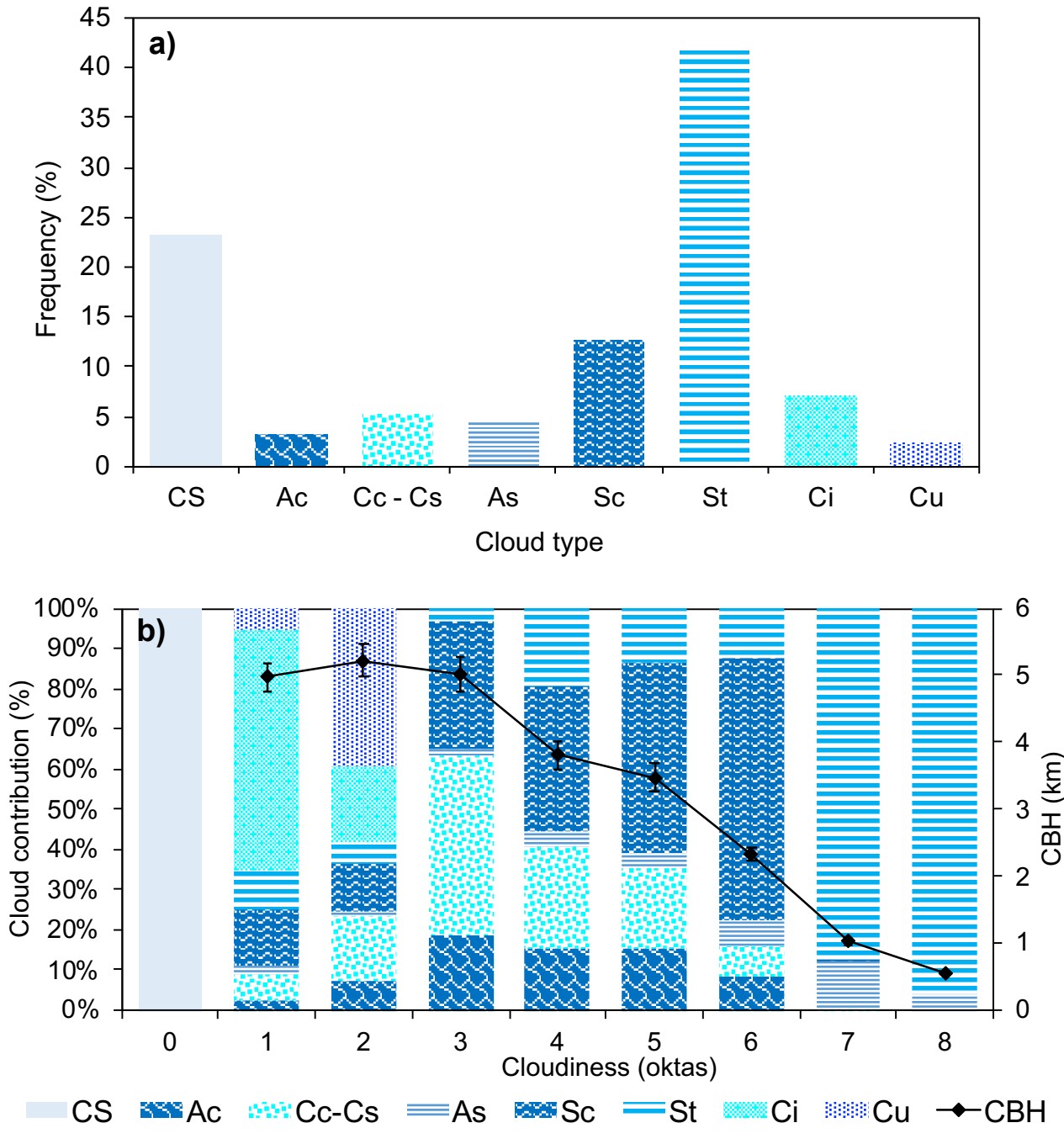

Figure 7. a) Time frequency (%) of the cloud type classified over the U9 site (CS means clear sky); b) contribution (%) of each cloud type to the oktas measured over the U9 site.

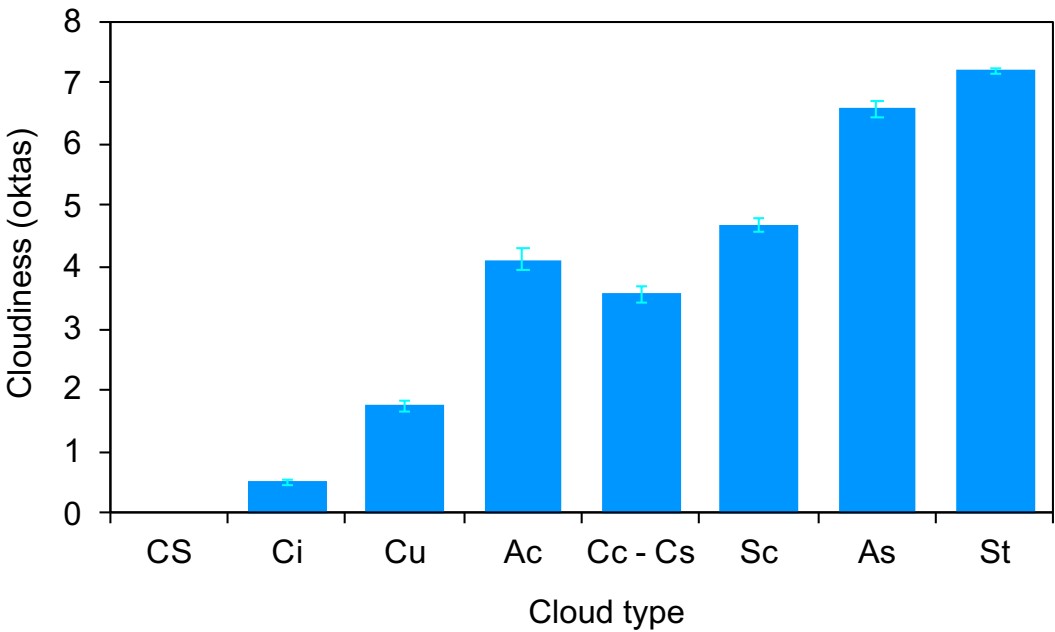

Figure 8. Cloudiness associated to each cloud type.

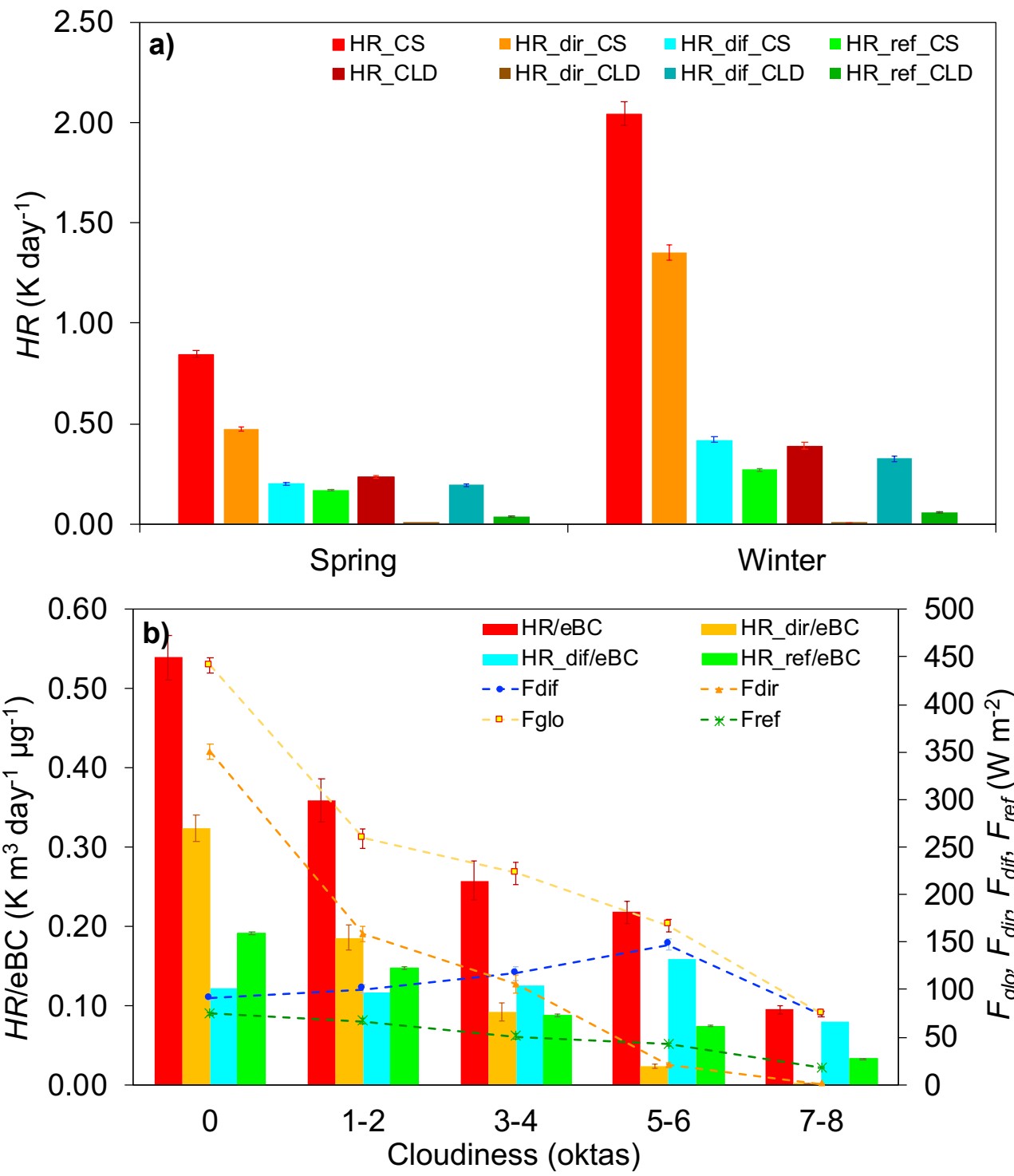

Figure 9. Monthly averaged values of: a) HR values and their direct, diffuse and reflected components (HR$_{dir}$, HR$_{dif}$ and HR$_{ref}$) during winter and spring both in clear sky (CS; oktas=0) and cloudy (CLD; oktas=7-8) conditions; b) HR/eBC values together with their direct, diffuse and reflected components (HR$_{dir}$/eBC, HR$_{dif}$/eBC and HR$_{ref}$/eBC), the direct, diffuse and reflected irradiance ($F_{dir}$, $F_{dir}$ and $F_{dif}$) and the global one ($F_{glo}$).

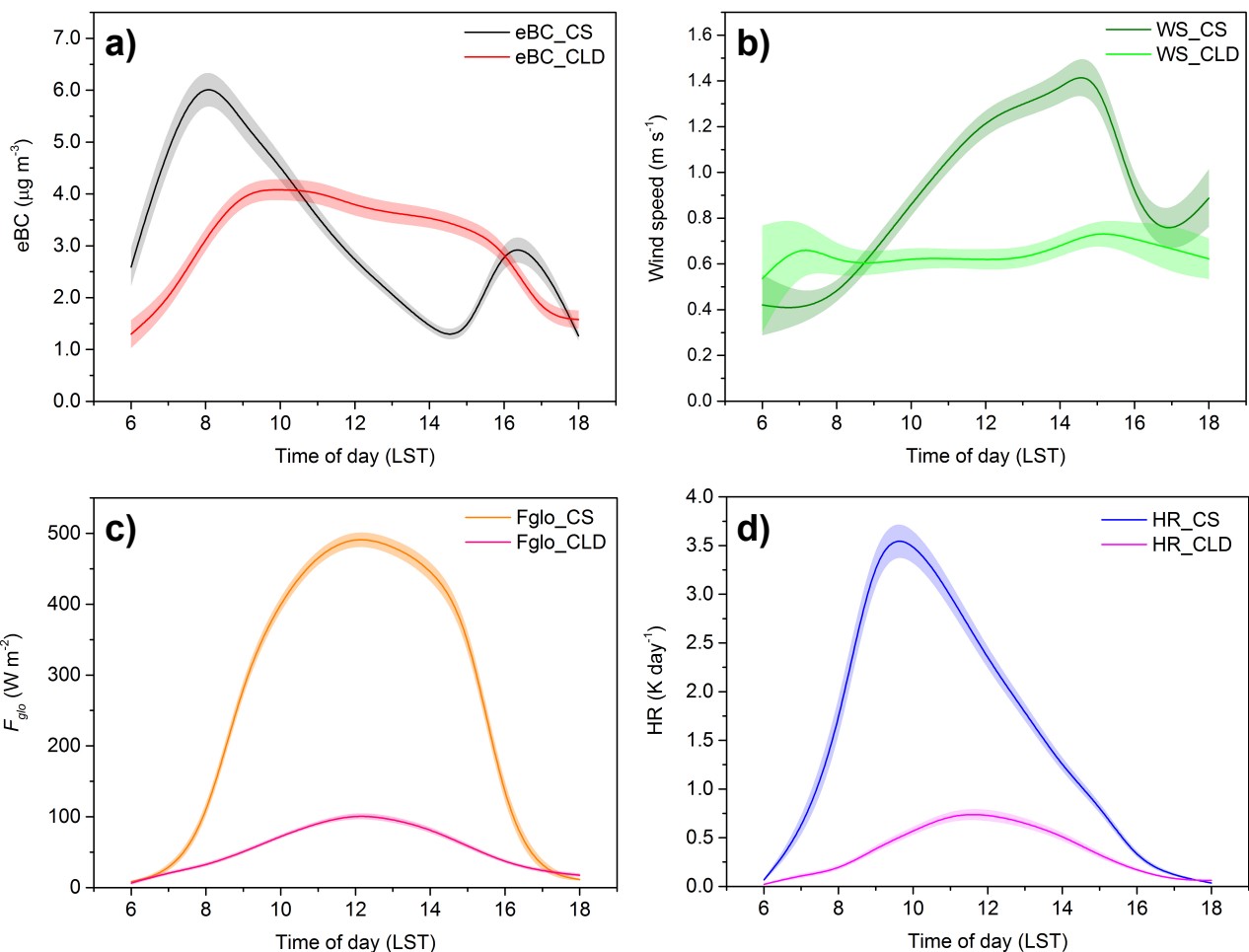

Figure 10. Diurnal pattern of eBC (a), wind speed (b), global irradiance ($F_{glo}$) (c) and HR (d). Data are averaged for clear sky conditions (CS, oktas=0) and cloudy conditions (CLD, oktas=7-8).

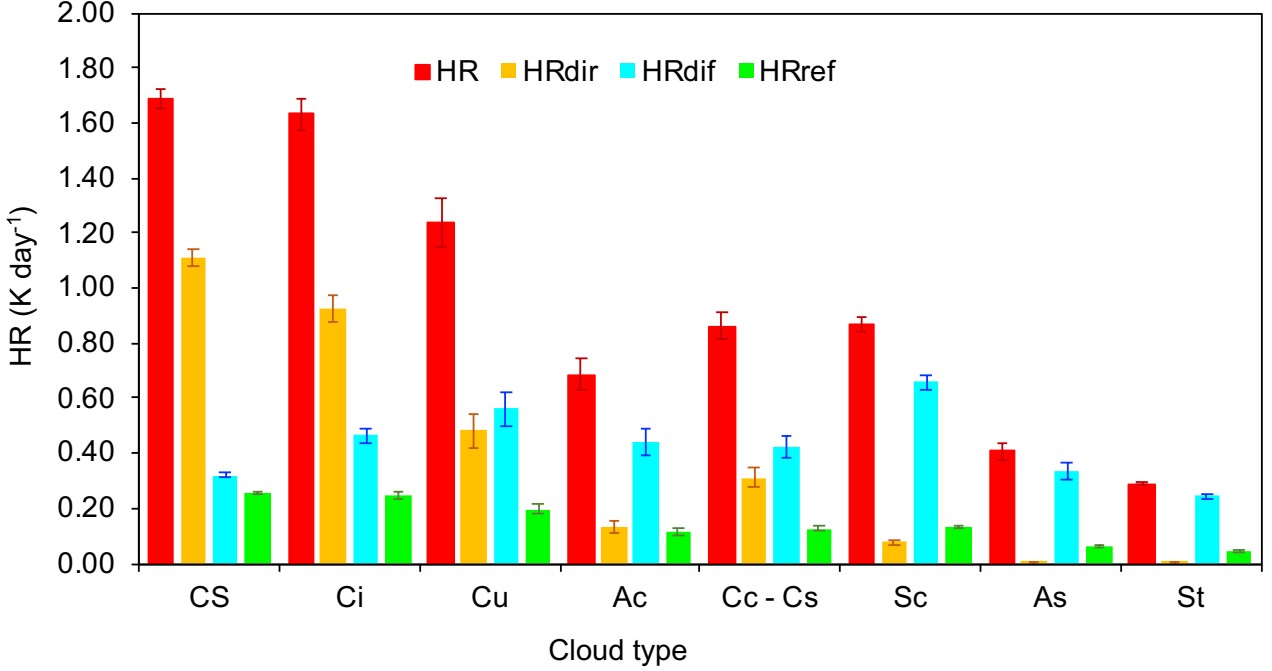

Figure 11. Impact of each cloud type on heating rate normalized to black carbon concentration: HR/eBC and $F_{glo}$ (a), $HR_{dir}$/eBC and $F_{dir}$ (b), $HR_{dif}$/eBC and $F_{dif}$ (c), $HR_{ref}$/eBC and $F_{ref}$.

Figure 12. Average values of total HR, $HR_{dir}$, $HR_{dif}$ and $HR_{ref}$ in function of the cloud type.

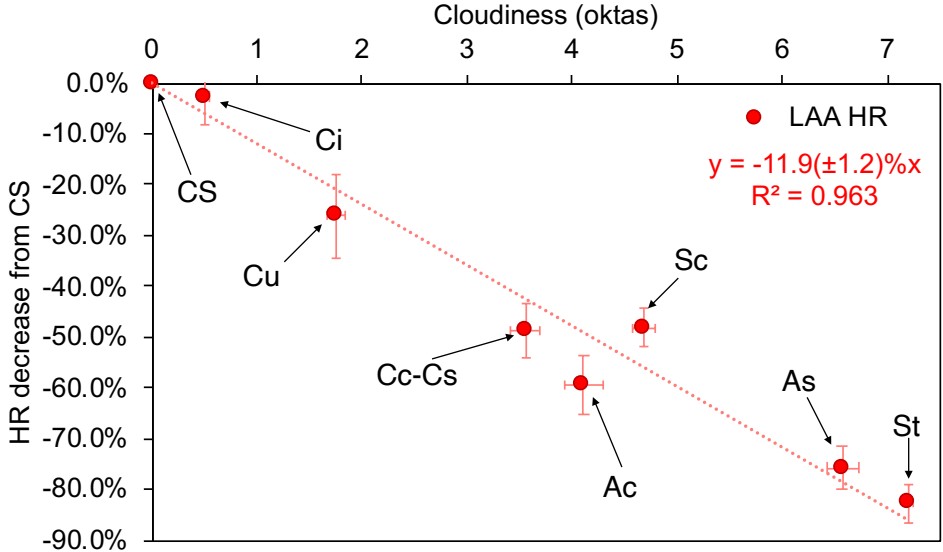

Figure 13. Percentage decrease of HR with respect to clear sky conditions in function of the oktas averaged for each cloud type.

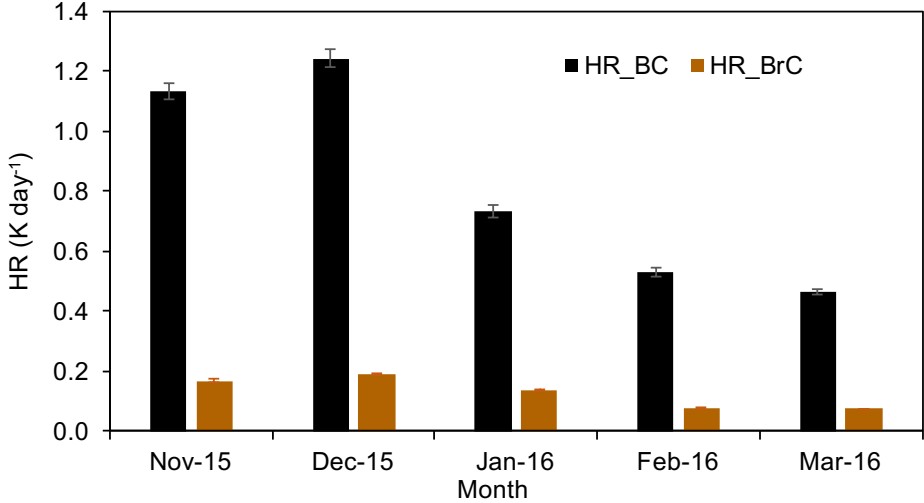

Figure 14. Monthly averaged data for the HR of both BC and BrC.

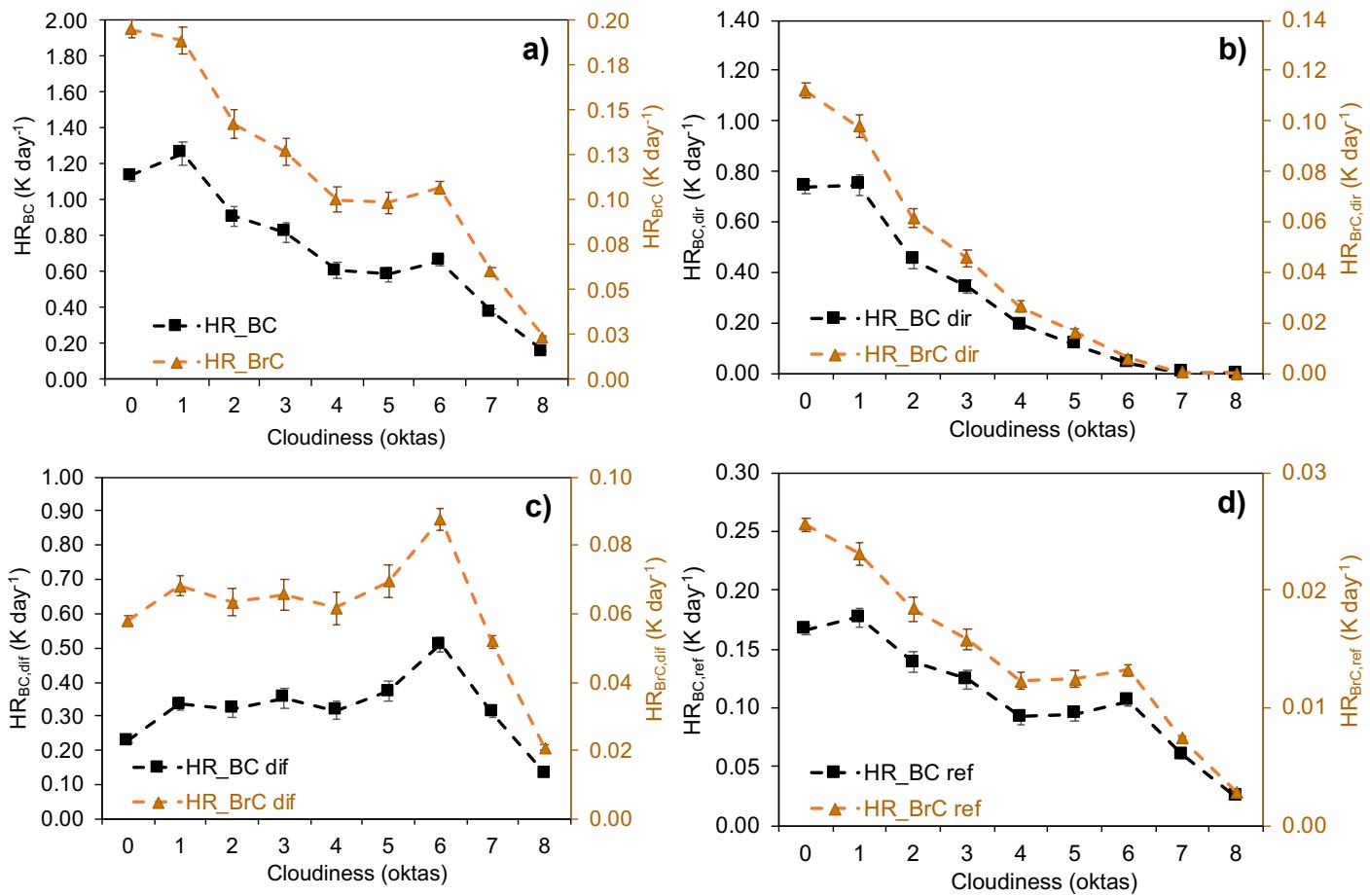

Figure 15. HR of BC and BrC in function of the oktas: a) total $HR_{BC}$ and $HR_{BrC}$, b) direct component of both $HR_{BC}$ and $HR_{BrC}$ ($HR_{BC,dir}$ and $HR_{BrC,dir}$), c) diffuse component of both $HR_{BC}$ and $HR_{BrC}$ ($HR_{BC,dif}$ and $HR_{BrC,dif}$) and d) reflected component of both $HR_{BC}$ and $HR_{BrC}$ ($HR_{BC,ref}$ and $HR_{BrC,ref}$). Note that, due to the different magnitude of $HR_{BC}$ and $HR_{BrC}$, the y-axis of $HR_{BrC}$ in the four panels was chosen as 1/10 of that of $HR_{BC}$.

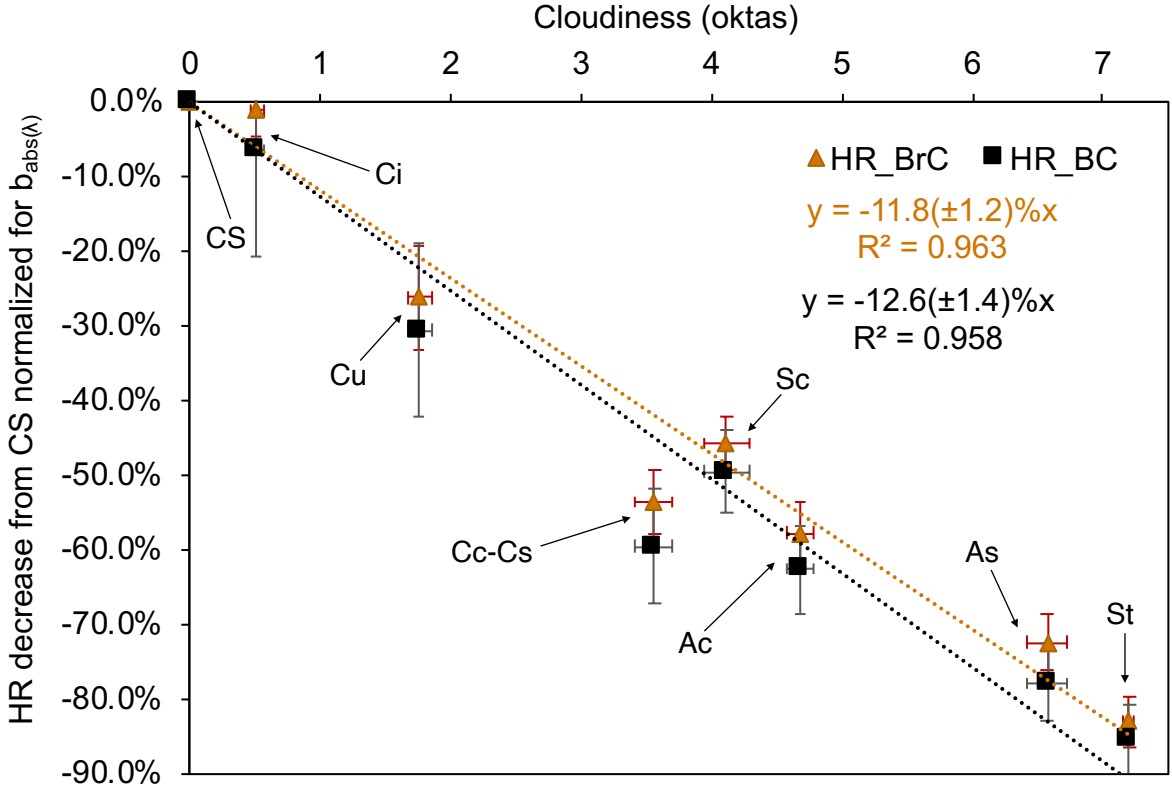

Figure 16. Percentage decrease of HR$_{BC}$ and HR$_{BrC}$ with respect to clear sky conditions in function of cloudiness (oktas) averaged for each cloud type.