# Peer review of "The impact of cloudiness and cloud type on the atmospheric heating rate of black and brown carbon in the Po Valley"

_Atmospheric Chemistry and Physics, 2020_

## Referee Comment (RC1) · Anonymous Referee #2 · 28 Jul 2020

This study explores the effect of clouds on heating rates driven by absorbing aerosols. They do so using observations and measurements sorted per different cloud types and coverage, separating the effects of black vs. brown carbon.

The data is collected in U9 sampling site in Milan which is a superstation that contains instruments to measure radiation, filter collecting aerosols that are analysed for their optical properties, meteorological station and a Lidar.

The topic of the paper is important. Exploring heating rates for different aerosol types under different cloud conditions will provide a very important information for aerosol effect on climate, and clouds. As the authors pointed out direct measurements of heating rates in different cloud conditions are quite uncommon.

[Figure]

The basic cloud classification makes sense in particularly as they added Lidar information for the clouds base. The results clearly show how cloudiness can affect heating rates and the bland between the radiation types.

One drawback of the paper is that it is very technical and not always easy to follow. Even if one understands the radiative transfer concepts, the physical assumptions and results are buried in the technicalities. It contains many technical terms that may appeal only to the instrumentation experts. Being familiar with radiation transfer concepts, I'm sure that there is a better way to describe the measurements and analyses such that a non-expert in the instrumentation could better enjoy it. The concertation of acronyms is high. It is hard to remember all of them and some that appear again later in the text force the reader to look back for their meaning and it disturbs the reading. On the other hand, some basic concepts that are key in this study are not well explained. The authors send the reader to read many other references for the basic methods and the equations. I believe that such study could be more of a standalone in which the basic physics is explained in a better way using less technical jargon.

I list here two basic comments on the methods and assumptions that should be clarified:

1) The aerosols that are collected at the station level serve as the only aerosol measurement and the basic assumption is that the filters collected at the station represent the whole boundary layer and therefore the heating rate is uniform for the layer below the clouds. I wonder how general this assumption is? This is always a key question of any work that try to link measurements near the surface to the atmospheric column. Is it always well mixed? Can the authors show that there is no dependency on the time of the day or the winds or the meteorology in general? Is it true for all seasons? For all cloud types? Moreover, if they have Lidar there can't they validate this assumption using the Lidar information. It would be nice to see uniform backscatter below the clouds to strengthen this basic assumption.

2) The radiation measurements are collected in the station and are product of electromagnetic radiation interaction with the whole atmospheric column. What about the contribution of aerosols above the boundary layer. Is it assumed to be canceled by the proposed method? Or is it assumed to be negligible? If not, how such aerosols can affect the results?

---

## Referee Comment (RC2) · Anonymous Referee #1 · 4 Aug 2020

The manuscript by Ferrero et al. acp-2020-264 titled "The impact of cloudiness and cloud type on the atmospheric heating rate of black and brown carbon" presents heating rate measurements of the atmosphere over the Po Valley, Italy.

The measurements are valuable as they are relatively rare in the community. The work is incremental on Ferrero et al. 2018, with the main incremental improvement being the automated separation of clouds into cloud types using radiometer measurements combined with Lidar-Ceilometer measurements. The introduction of lidar information into the automated cloud classification is novel and may be valuable to other work, yet was not thoroughly validated. **I recommend that the authors describe this cloud classification algorithm in detail in a separate paper and include more detailed validation work. If the authors do not follow this recommendation, they must**

**provide a clear argument for why in the review responses and in the manuscript.**

The actual presentation of the results in this manuscript is incredibly poor. Here I present 200 lines of comments which I had to make simply in order to understand the results. The discussion is long, dense, and disorganized. Most of these comments are on presentation and organization, at the level which is normally given to an author's first draft of a first manuscript. After I finally understood what was done and what the results were, I see a valuable data set. However, the scientific interpretation is on a similar level to the writing.

I fear that my scientific feedback has been drowned by the poor writing, manuscript organization, and figure presentation in this work. **To emphasize my main scientific comments I have used boldface text in the following.** The authors should streamline their manuscript by referring to Ferrero et al. 2018 whenever possible, by separating their cloud analysis from their light-absorbing aerosol analysis, and by clearly demonstrating whether or not there is any value to the different levels of information available here. Those levels are: 1) heating rate resolved in time, 2) heating rate resolved by time and cloud height, 3) heating rate resolved by time, cloud height, and cloud type.

My recommendation to the authors is to completely rewrite this manuscript and reinterpret the results. Since this work is incremental to earlier, well-presented work (Ferrero et al., 2018), and since the results are well supported if poorly presented and interpreted, I do not recommend rejection but major revisions to the Editor.

(NB: I have not numbered my feedback below. When the authors respond, please refer to my comments as "C1P2" for page C1, paragraph 2, etc. Please also copy and paste the comment before responding.)

**1 General comments**

**The Introduction provides a strong motivation for the importance of HR and cloudiness, but the final 2 paragraphs are poorly structured. Please emphasize more the importance of cloudiness, including common levels to be expected (i.e. expand the discussion around Crock et al.) and feedbacks (i.e. expand the discussion around Perlwitz and Miller 2010, which is central to this work's motivation)**

In many places in this manuscript the authors say that "$F_\lambda$ is the radiation" when they seem to mean "spectral irradiance" (W/m2/nm). The word "radiation" is not accurate. On line 351 they suddenly use "irradiance". Be consistent. Avoid confusing your readers.

It appears that all reported quantities are strongly corelated for direct and reflected irradiance. If this is correct then please sum these two quantities in all figures. The presentation is too confusing.

The preceding work (Ferrero et al. 2018) included diurnal trends in irradiance, which would be valuable here. Why were they not included?

**2 Unclear text or discussion**

**I would require that the authors add a glossary table (defining abbreviations) before publication due to the large number of symbols in the figures. Moreover, please improve the legends (e.g. 6b)**

I will suggest that the authors change Fig 8, 9 axis labels from "Ci" to "cirrus" etc for all cloud types.

Figure 10's axis label should include HR.

79 Higher than clear sky conditions in certain localized regions only, or?

110 'Conversely' is not appropriate here

110 Please start a new paragraph at "This study". End this paragraph instead with "This study aims to fill this gap" or similar.

Fig S2. Please add an "uncorrected" panel to give readers an idea of the magnitude of the correction (which is related to uncertainties).

Line 168 What is the physical meaning of the C value being close to the GAW value? (e.g.: The particle size and single scattering albedo were typical of atmospheric monitoring sites. Collaud Coen et al., 2010)

171 Was the MRI built by U Milano-Bicocca? Please add manufacturer, even if it is homebuilt.

200 What is the uncertainty or accuracy of the Nimbus 15k? In other words, please mention the limitations as well as the strengths of this system.

238 "This is due to the negligible ..." This sentence is not accurate. The authors may instead consider the simple harmonic oscillator (Moosmuller et al. 2011, doi:10.5194/acp-11-1217-2011 ) or energy gap (Sun 2007 doi:10.1029/2007GL029797) models here.

243 change 'successfully' to 'previously'

251 First use of LAA on this line was not defined. Please introduce the concept in the introduction. It is a nice and useful abbreviation.

350 Give limits of the integral in the equation.

**3 Opportunities to shorten and clarify the text**

135 This information is redundant with lines 110-112, please shorten 110-112. 171-177 Please cite Cogliati immediately after introducing the MRI.

263 Change "N represents one of the possible 9 classes" to "N = 1,2,3, ..., 9, representing 9 classes of cloud fractions"

**3.1 Heating rate measurements – Equation 1,2, and 3**

There is no need for all 3 equations here. Delete Eq 1. Start with Eq 2 to introduce and define ADRE first, then the reader will understand Equation 3 (new Eq 2) naturally. Remove $\mu$ and replace it with $\cos\theta$. There is no need to introduce $\mu$ because it is only used twice in the manuscript, and anyway $\cos\theta$ is more easily understood.

Also, use the integral sign to specify "integral over the whole $2\pi$" (line 209) instead of writing it only in words. Same for $\theta$.

Equations 4 and 5 use subscripts dir, dif, ref to specify direct, diffuse and reflected radiation. Equations 1, 2, 3 use "nth type of F" to do the same. Choose one and stick to it. The text subscript is better, and the authors obviously agree as they used it later in the manuscript.

The term $F_{glo} = F_{dir} + F_{dif} + F_{ref}$ must be introduced already in the first equation of Section 2.2. Prepare the reader for Equation 6. I have to assume that this is the definition, the authors never gave it.

**The only real difference between HR and ADRE in Figure 5 is the air density $\rho$. So, the authors should plot $\rho$ in the figure and emphasize this in the caption to avoid confusing readers who are not familiar with HR or ADRE (in other words, most readers).**

**Since the main contribution of this manuscript is to discuss heating rates, why discuss ADRE at all? Leave that to the SI. Or, of the authors disagree, then discuss only ADRE. HR appears more valuable as $\rho$ is a meteorological variable.**

**3.2 Cloud classification – Section 2.3.2, and Figures 2 and 3**

This section is a mess.

Do not mix discussion and results in S 2.3.2. Review the literature first, then present your results. Present "failed analysis" in the SI, not in the main text. Use only 1 or 2 sentences in the main text for failed analysis.

This reviewer spent several minutes studying Figure 2 and writing the following comments before learning that it is a "this did not work" figure. The writing should make this clear immediately. Restructure S2.3.2 to fix this.

The section here concludes that the 2 literature methods discussed (Duchon 1999 and Harrison et al 2008) were not adequate, based on the conclusions of Harrison's work. So the authors introduce a new method, but with no validation of it. How can the reader trust this? I believe the author's work is valuable but the discussion needs to include validation.

Line 308 how many cases (%) were analyzed after this limitation?? The authors should not discard cases of multiple cloud layers. Simply include a category "Multiple layers" or "Complex cloud layers" or similar.

If I am to believe this section then the authors have contributed a numerical algorithm to the topic of automated cloud type analysis. Only 2 papers have been published on this topic, and most cloud type identification remains manual. This is the 3rd paper to contribute to this topic in 30 years, yet the authors did not include a solid analysis of the algorithm.

**Either the authors have used an unvalidated algorithm in their work, or the authors should write an entire manuscript describing their validation of what seems to be a valuable contribution. Separating the cloud-algorithm work from the radiative heating work would mean removing Figures 2, 3, 4, 7, 14, and some SI figures from this to another manuscript. This would avoid breaking up the "BC+BrC" story.**

I note that the Harrison et al. 2008 work was missing from the reference list.

Other comments:

Make Fig S3 axis labels consistent with the language in S 2.3.2. Put R on the x axis.

**The use of a 20 minute interval in calculating SD for the Duchon and O'Malley method means that wind speeds are included in the measurement of cloudiness fluctuations. This must be discussed. How do wind speeds compare with this 20 minute interval?**

Lines 281-314 break up this huge paragraph.

Figure 2 comments: Fig 2's legend is inaccurate, there is no dashed line in the legend. In Fig 2g, colour the red line in the same way that the points in Fig 2h are coloured. Move the entire figure to the SI. Consider adding photographs to this figure.

**4 Results and conclusions**

The results and discussion are too long, relative to the information content of the manuscript. The information is valuable but does not require extensive discussion.

As I mentioned above, the discussion is broken up by switching between the cloud analysis and the BC+BrC analysis. Start from the top and go down. Focus on the cloud effects before attributing RH to LAA afterwards and then to BC+BrC.

**The authors introduce direct, diffuse, and reflected irradiance yet do not present the data consistently. Some figures separate all 3. Some figures present direct, diffuse, and total (Fig 14). Some figures (Fig 13) present sums of 2, in various combinations. Some figures combine all 3 as "global irradiance" others combine all 3 as "total irradiance". Please, assess your data, choose one message, and present it clearly to your audience. Follow Harrison et al. 2008 in presenting the diffuse fraction unless your data support an alternative. Figure 9 is the only figure that suggests a difference between direct and reflected, but the impact on heating rate is unclear because Figure 13 changed the presentation strategy.**

The conclusions are similarly confused. Why are different cloud types discussed in detail when Figure 10 and 15 clearly show that the key predictor is oktas and not cloud type? Only high clouds (cirrus, cirrocumulus, and cirrostratus) do not follow this trend, presumably because they are well above the aerosol layers.

**I do not see any support for the final conclusion that the cloud impact affected HR of BC more than of BrC. The absolute value of the BC HR was higher initially, so it would naturally change more. My interpretation of the authors' results is that there is no need to attribute cloud types in future work, and that cloud height data combined with diffuse fraction (Harrison et al. 2008) may be sufficient. If this work is to be extended to other monitoring sites the authors must address this point explicitly. Simpler measurements are more likely to be adopted by others.**

---

## Author Comment (AC1) · 15 Nov 2020

We thank the reviewer for his or her helpful comments and insight. They allowed us to improve the scientific quality and presentation of the work done. We respond to the general and to the specific points below. All the comments are addressed in the revised manuscript.

**GENERAL COMMENTS**

**General Comment 1 (C1P1): The manuscript by Ferrero et al. acp-2020-264 titled "The impact of cloudiness and cloud type on the atmospheric heating rate of black and brown carbon" presents heating rate measurements of the atmosphere over the Po Valley, Italy.**
**The measurements are valuable as they are relatively rare in the community. The work is incremental on Ferrero et al. 2018, with the main incremental improvement being the automated separation of clouds into cloud types using radiometer measurements combined with Lidar-Ceilometer measurements. The introduction of lidar information into the automated cloud classification is novel and may be valuable to other work, yet was not thoroughly validated. I recommend that the authors describe this cloud classification algorithm in detail in a separate paper and include more detailed validation work. If the authors do not follow this recommendation, they must provide a clear argument for why in the review responses and in the manuscript.**

Answer to General Comment 1 (C1P1): We thank the reviewer for the comment on the experimental results, their relevance and implications reported in this work. Indeed, as underlined in the review, the present work represents an important incremental step of Ferrero et al. (2018).
We carefully considered the suggestion to split the paper in two. However, the cloud classification is only one of the incremental improvements. The main goal of our study is to experimentally unravel the relative and synergic role of cloudiness and of different cloud types on the heating rate (HR) of light absorbing aerosol (LAA) in general and that of BC and BrC in particular. As we state at the end of the introduction (revised version of the manuscript) we aim to:

1. describe the interaction between cloudiness and light-absorbing aerosol, to describe the resulting aerosol HR as a function of cloudiness, and in turn to estimate the systematic bias introduced by incorrectly assuming clear-sky conditions in radiative transfer models;
2. introduce an original cloud type classification to investigate the impact of both cloudiness and cloud types on the total HR;
3. separate the contributions of BC and BrC carbonaceous fractions to HR and investigate their relative impact on the total HR as a function of sky conditions.

The results we present in this study add an important piece of information to the influence of the two most important LAA species (i.e. BC and BrC) in different sky conditions. Therefore, the manuscript was planned from the beginning as a whole, with the main focus on the environmental influence of LAA on the climate.
Immediately after our submission (20 Mar 2020), Ylivinkka et al. submitted to Atmospheric Measurement Technique (03 Apr 2020) a paper titled "Clouds over Hyytiälä, Finland: an algorithm to classify clouds based on solar radiation and cloud base height measurements"( https://doi.org/10.5194/amt-13-5595-2020) which was accepted and published (22 Oct 2020). The

paper discusses a cloud classification technique very similar to ours. This is a clear coincidence of an interesting scientific development.

Taking into account the reasons above, due to the fact that the concerns were mostly related to one section (2.3.2, cloud classification section), and due to Reviewer#2 asking for a technical simplification of the paper, we have decided (previously asking the opinion of the handling editor) to not split the paper into two, but rather to improve the present manuscript. We have rewritten large part of the manuscript main body, and moved material to the Appendix and the now modified Supplemental material. We have additionally taken into account the publication of Ylivinkka et al. (2020) and included this and other references related to the lidar-ceilometer capabilities at detecting cloud base and cloud classification. To answer this reviewer comment, a validation of the classification scheme was carried out in in two steps.

The first validation step was carried out comparing the automatized cloud classification (based on Duchon and O'Malley, 1999, and additionally lidar cloud base height) with a visual cloud classification based on sky images collected during 1 month of the field campaign. The second validation step involved the recent published method discussed by Ylivinkka et al. (2020). Their method is based on the same logical approach followed in our work: the application of Duchon and O'Malley (1999) classification improved by the knowledge of the cloud base height. The aim of the second step was to determine the degree of consistency between the two approaches which were developed simultaneously and independently in two completely different European regions.

The complete validation is reported in Appendix B ("Cloud type validation"). This was performed not to interrupt the flow of the manuscript, as requested in the Specific Comment 25 (C6P1-C7P1). The overall balanced accuracy was 80% for the visual validation and 90% for the intercomparison with Ylivinkka et al. (2020) (please see answer to your specific question 23, C6P1, for further details). This shows the reliability of the classification algorithm, allowing us to study the impact of clouds on LAA HR with a sufficient degree of certainty.

**General comment 2 (C2P1). The actual presentation of the results in this manuscript is incredibly poor. Here I present 200 lines of comments which I had to make simply in order to understand the results. The discussion is long, dense, and disorganized. Most of these comments are on presentation and organization, at the level which is normally given to an author's first draft of a first manuscript. After I finally understood what was done and what the results were, I see a valuable data set. However, the scientific interpretation is on a similar level to the writing.**

**I fear that my scientific feedback has been drowned by the poor writing, manuscript organization, and figure presentation in this work. To emphasize my main scientific comments I have used boldface text in the following. The authors should streamline their manuscript by referring to Ferrero et al. 2018 whenever possible, by separating their cloud analysis from their light-absorbing aerosol analysis, and by clearly demonstrating whether or not there is any value to the different levels of information available here. Those levels are: 1) heating rate resolved in time, 2) heating rate resolved by time and cloud height, 3) heating rate resolved by time, cloud height, and cloud type.**

**My recommendation to the authors is to completely rewrite this manuscript and reinterpret the results. Since this work is incremental to earlier, well-presented work (Ferrero et al., 2018), and since the results are well supported if poorly presented and interpreted, I do not recommend rejection but major revisions to the Editor.**

**(NB: I have not numbered my feedback below. When the authors respond, please refer to my comments as "C1P2" for page C1, paragraph 2, etc. Please also copy and paste the comment before responding.)**

Answer to General Comment 2 (C2P1): We have considerably rewritten the manuscript as suggested in the comment.

We started with the suggestion "to separate the cloud analysis from the light-absorbing aerosol analysis". As reported in the answer to General Comment 1, we cannot split the manuscript in two manuscripts, as a similar cloud classification scheme was just published. The strength and the innovation of the present paper is the synergy between the automatic classification of cloudiness and quantifying the effect of the light-absorbing aerosols on the climate. Thus, we fully re-organized the Results and discussion section following the suggestion in order to improve the full manuscript, to clarify the logic behind the methodology, and to more specifically discuss the different aspects (levels) of the results. Now, following the suggestion on the three different levels of information, the Results and discussion section features the following arrangement of the subsections:

- Section 3.1 introduces the environmental context of the measurement campaign and the magnitude of the observed parameters (eBC, irradiance, HR and cloud data). We have incorporated here the suggestion "to separate the cloud analysis from the light-absorbing aerosol analysis". All cloud analysis is presented here. The validation of the cloud classification was moved to the Appendix B.
- Old sections 3.2 and 3.3 were re-written in line with the changes performed in section 3.1 and merged in a new section 3.2 with two sub-sections discussing the influence of clouds (cloudiness and cloud type; sub-sections 3.2.1 and 3.2.2) on the total HR.
- The old section 3.4 (now section 3.3) was completely re-written merging and shortening the two original sub-sections 3.4.1 and 3.4.2 discussing the influence of cloudiness, cloud type and average photon energy on the HR apportioned to BC and BrC.

This gradual approach streamlines the manuscript, making it easier to read. We improved the Results and Discussions outline at the beginning of section 3 describing this approach. Moreover, all the manuscript was revised simplifying all the sections and making them more concise and easier to follow. We did not use the acronyms for the concepts which did not appear too often and also added an Appendix explaining all the remaining acronyms and symbols present in the paper.

To address the suggestions about the data analysis and the most relevant results, we moved the Figure S5 (time resolved heating rate) to the main body of the manuscript (now Figure 5 in the manuscript; here below as Figure A1) adding a proper description. We first improved the new Figure 5 adding both the cloudiness (expressed in oktas) and the cloud base height. The same was also done for Figure S6.

Then we focused on the reviewer's suggestions concerning the relationship between 1) the heating rate and cloud height and 2) the heating rate, cloud height and cloud type. This helped us to enrich the explanation of the interaction between the clouds and light-absorbing aerosols. We prepared Figures A2a-c, A3 and A4a-d which are discussed here below.

[Figure]

Figure A1 (Figure 5 in the revised version of the paper). High time resolution data (5-min) for eBC, global irradiance ($F_{glo}$), cloud base height (CBH), coldness (oktas), and the related heating rate (HR) from 1 November 2015 to 1 April 2016.

[Figure]

Figure A2. Relationship between a) cloud base height (CBH) and cloudiness (oktas), b) HR/eBC and CBH and c) HR/eBC and cloudiness (oktas).

Figure A2a shows a clear relationship between the cloud base height (averaged for each okta) and cloudiness. This is not surprising, considering the contribution of each cloud type to the cloudiness reported in the manuscript (section 3.3, page 12, lines 440-445): "the contribution (expressed in oktas) of the cloud type is reported in Figure 7b. This clearly shows that, while Stratus clouds were mostly

responsible of overcast situations (oktas=7-8, frequency: 87 and 96%), Stratocumulus clouds dominated the intermediate cloudiness conditions (oktas=5-6, frequency: 47 and 66%); moderate cloudiness (oktas=3-4) were mostly due to a transition from Cirrocumulus and Cirrostratus to Stratocumulus while low cloudiness (oktas=1-2) were mostly dominated by Cirrus and Cumulus (frequency: 59 and 40%, respectively)."

In section 2.3.2 and Figure 4 we reported that low level clouds (<2 km) include Stratus (St), Cumulus (Cu) and Stratocumulus (Sc), mid-altitude clouds (2-7 km) include Altostratus (As), and Altocumulus (Ac) and high-altitude clouds (>7 km) include Cirrus (Ci), Cirrocumulus and Cirrostratus (Cc-Cs). Combining these cloud altitudes with the overcast situation statistics above, it appears that the higher cloudiness (higher oktas) was due to a higher frequency in low-mid altitude clouds. We described this point better in the manuscript text (section 3.1): we included the whole Figure A2 to the Supplemental material and modified Figure 7b by adding the average cloud base height for each okta (below as Figure A3). Moreover, as cloud base height was related with cloudiness, a good linear relationship can be derived between the HR/eBC and cloud base height (Figure A2b; $R^2$=0.857), but this relationship is weaker than that between HR/eBC and cloudiness (Figure A2c; $R^2$=0.935). The cloudiness, describing the fraction of sky covered by clouds, is a better predictor of the capability to suppress the incoming radiation and thus lower the HR of BC and BrC, because the relationship between cloudiness and cloud base height – shown on Figure A2a, is weaker at higher cloud base heights. Addressing also the specific question 33(C8P1), we can add that the relationship shown in Figure A2a between cloud base height and cloudiness should be also investigated in other monitoring sites around the world to see whether the cloud base height can be used as a promising prognostic variable for the HR of light absorbing aerosols (see also answer to specific comment 33).

[Figure]

Figure A3 (Figure 7b, revised). Contribution (%) of each cloud type to the oktas measured over the U9 site, and averaged (±95% confidence interval) cloud base height (CBH) for each okta. CS=clear sky,; Ac=AltoCumulus; Cc-Cst=CirroCumulus-CirroStratus; As=AltoStratus; Sc=StratoCumulus; St=Stratus; Ci=Cirrus; Cu=Cumulus.

Figure A4 (heating rate, cloud height and cloud type) shows the absence of any relationship between HR/eBC (and each of its components) under different cloud types and the cloud base height (averaged

in this case for each cloud type). This is reasonable, since, as reported in the manuscript (section 2.3.3, Figures 3 and 4), the cloud base height is a prognostic variable needed for the cloud classification and does not account for the amount of clouds present in the sky. We stress again that in Figure A4 the cloud base height is averaged for each cloud type thus reflecting mostly a property of the different cloud types above the measuring site.

[Figure]

Figure A4. Impact of each cloud type on: HR/eBC (a), $HR_{dir}$/eBC (b), $HR_{dif}$/eBC (c), $HR_{ref}$/eBC. The cloud base height (CBH) is reported in each panel. CBH is not present in clear sky conditions (CS).

**General comment 3 (C3P1). The Introduction provides a strong motivation for the importance of HR and cloudiness, but the final 2 paragraphs are poorly structured. Please emphasize more the importance of cloudiness, including common levels to be expected (i.e. expand the discussion around Crock et al.) and feedbacks (i.e. expand the discussion around Perlwitz and Miller 2010, which is central to this work's motivation)**

Answer to General Comment 3 (C3P1): We thank the reviewer for these comments. The work of Perlwitz and Miller (2010) was introduced in the original manuscript at lines 118-120 as they reported a counterintuitive feedback linking the atmospheric heating induced by tropospheric absorbing aerosol to a cloud cover increase. Particularly, they observed this change for low level clouds as a response to relative humidity due to opposite changes in specific humidity and temperature. Furthermore, Perlwitz and Miller (2010) concluded that higher levels of absorption by aerosols were responsible for two counter-acting processes: a larger diabatic heating warming of the atmospheric column (decreasing relative humidity), and a corresponding increase in the specific humidity, counteracting the drop on the relative humidity and resulting in more low cloud cover with increasing aerosol absorption.

This was an important result, given the fact that the traditional semi-direct effect relates the atmospheric heating induced by absorbing aerosol to a decreasing relative humidity and less cloud cover. This can further increase the amount of the incoming solar radiation that reaches Earth's surface and is absorbed, leading to positive feedback characterized by additional warming and a decrease in the cloud amount (e.g. Koren et al., 2004). Thus, the aim of introducing the work of Perlwitz and Miller (2010) was to point the readers' attention to the fact, that measuring the atmospheric heating rate in cloudy conditions is needed as constrain and/or input for more comprehensive climate model, to shed light on the sign and magnitude on the related feedbacks on cloud dynamics.

For the reason above we extended the introduction section by adding these considerations. At the same time, results were discussed with the above cited studies in mind, recapping this also in the conclusions.

**General comment 4 (C3P1). In many places in this manuscript the authors say that "Fλ is the radiation" when they seem to mean "spectral irradiance" (W/m2/nm). The word "radiation" is not accurate. On line 351 they suddenly use "irradiance". Be consistent. Avoid confusing your readers.**

**It appears that all reported quantities are strongly corelated for direct and reflected irradiance. If this is correct then please sum these two quantities in all figures. The presentation is too confusing.**

**The preceding work (Ferrero et al. 2018) included diurnal trends in irradiance, which would be valuable here. Why were they not included?**

Answer to General Comment 4 (C3P1): We thank the reviewer for addressing the terminology question. In many parts of the manuscript the term "radiation" was used as a general synonymous of different more specific terms under the assumption that it was easier to identify the specific object of the sentence (e.g. spectral irradiance) from its context. We understand from the comment that this can lead to some confusion, therefore we changed the generic word "radiation" with the appropriate term everywhere in the manuscript. We left the term "radiation" when a generic reference to it was needed (e.g. introduction).

The observation that "all reported quantities are strongly correlated for direct and reflected irradiance" is right. However, a sum of the two quantities is wrong from a methodological point of view. Also, we need to first show that they are correlated. In fact, the relationship that appears in the present work is due to the simple coincidence that measurements were collected upon Milan were the surface albedo is quite stable in time. For any other application (e.g. in the Arctic and Antarctic or other regions featuring extreme changes between snow cover and bare ground, over a steppes or other grassed regions, measurements from ships, drones) the reflected spectral irradiance can change with sky conditions, leading to a nonlinear relationship with the direct spectral irradiance. For these reasons we decided to maintain them separated, and, in agreement with the specific comment 31 (C8P1), we rigorously reported direct, diffuse and reflected spectral irradiance contribution to the HR in every Figure of the manuscript.

Finally, the diurnal variation in irradiance was not reported in this work, since they were presented earlier (Ferrero et al. 2018). However, we agree with the suggestion that they would be valuable in the present manuscript, especially when reporting the diurnal pattern of the HR averaged for clear sky conditions and cloudy conditions – old Figure 11 (now Figure 10 in the revised version of the

manuscript). We added to Figure 11 (now Figure 10a-b) the global irradiance and eBC, together with wind speed. The new Figure 10a-b is reported here below as Figure A5.

[Figure]

Figure A5 (Figure 10 in the revised version of the manuscript). Diurnal variation of: a) eBC, global irradiance ($F_{glo}$), and wind speed (WS) and b) HR. All three parameters are averaged for clear sky conditions (CS, oktas=0) and cloudy conditions (CLD, oktas=7-8).

**SPECIFIC COMMENTS**

**Specific Comment 1 (C3P1): I would require that the authors add a glossary table (defining abbreviations) before publication due to the large number of symbols in the figures. Moreover, please improve the legends (e.g. 6b)**
**I will suggest that the authors change Fig 8, 9 axis labels from "Ci" to "cirrus" etc for all cloud types.**

Answer to Specific Comment 1 (C3P1): We fully agree with you that a glossary table is needed and we added it in the manuscript as a new section "Appendix A".
Figure 6b (now Figure 9b in the revised version of the paper) required a legend improving in order to better separate the clear sky case with respect to cloudy ones. The same is also valid for Figure 11 (now Figure 10 in the revised version of the paper). At this purpose we used the term "CLD" in each legend for cloudy conditions (oktas=7-8) and "CS" for clear sky. After considering the comment about the cloud acronyms, we decided to maintain the acronyms and improve them using the nomenclature in the international abbreviations for cloud genera and species of the World Meteorological Organization (WMO, https://cloudatlas.wmo.int/en/home.html) for brevity, clarity and comparability with Figures 10 and 15 (now Figures 13 and 16 in the revised version of the paper). We are now using the nomenclature in the international abbreviations for cloud genera and species of the World Meteorological Organization. They were included in in the new section "Appendix A".

**Specific Comment 2 (C4P1): Figure 10's axis label should include HR.**

Answer to Specific Comment 2 (C4P1): We thank the reviewer for the suggestion. We changed Figure 10 y-axis (now Figure 13 in the revised version of the paper) accordingly and we did the same also with Figure 15 (now Figure 16 in the revised version of the paper). The x-axis was also improved in both Figures with a more rigorous label "Cloudiness (oktas)".

**Specific Comment 3 (C4P1): 79 Higher than clear sky conditions in certain localized regions only, or?**

Answer to Specific Comment 3 (C4P1): Line 79 summarizes results from the works of Mims and Frederick (1994) and Feister et al. (2015). Mims and Frederick (1994) determined that scattering from the sides of cumulus clouds can enhance the total (global) UV-B solar irradiance by 20% or more over the maximum solar noon value when cumulus clouds were just near the Sun (the cloud not blocking the solar disk). In a similar way, Feister et al. (2015) concluded that the scattering of solar radiation by clouds can enhance UV irradiance at the surface; for example, Cumulonimbus clouds with top heights close to the tropical tropopause layer have the potential to significantly enhance diffuse UV-B radiance over its clear sky value. We reported these findings as UV represents an important region for BrC absorption and future studies should investigate specific cases of UV enhancement (which is actually beyond the aims of the present work) with respect to the impact of BrC on the climate. We extended and improved this part of the Introduction.

**Specific Comment 4 (C4P1): 110 'Conversely' is not appropriate here**

Answer to Specific Comment 4 (C4P1): We agree and the sentence was rewritten as follows: "To our knowledge, there has been no experimental investigation on the impact of aerosol layers below the clouds, where most of the aerosol pollution typically resides".

**Specific Comment 5 (C4P1): 110 Please start a new paragraph at "This study". End this paragraph instead with "This study aims to fill this gap" or similar.**

Answer to Specific Comment 5 (C4P1): We thank the reviewer for the suggestion. We changed the text accordingly. Please note that we added about 15 lines as answer to your general comment 3 before this sentence, and that the end of the Introduction was reordered for clarity.

**Specific Comment 6 (C4P1): Fig S2. Please add an "uncorrected" panel to give readers an idea of the magnitude of the correction (which is related to uncertainties).**

Answer to Specific Comment 6 (C4P1): This figure (Figure A6) is attached below. Care needs to be taken when interpreting the meaning of this plot as it is not related to the Aethalometer measurement uncertainty.

First, we need to recall that in the Aethalometer AE31, the aerosol sample is continuously deposited on the filter tape. Seven light sources with different wavelengths ($\lambda$) illuminate the tape. Attenuation (ATN) of light is measured under the sample for each of the 7 wavelengths relative to an illuminated sample-free part of the tape acting as a reference. ATN is calculated as:

$$ATN = 100 * \ln (I_0/I) \tag{A1}$$

where $I_0$ and $I$ are the intensity of light transmitted through the reference and aerosol blank spot of the filter respectively. The attenuation coefficient of the aerosol particles collected on the filter tape, $b_{ATN(\lambda)}$, is then defined as follows (Weingartner, et al., 2003):

$$b_{ATN(\lambda)} = \frac{A}{Q} \frac{\Delta ATN(\lambda)}{\Delta t} \tag{A2}$$

where A is the filter spot area, Q the flow rate and $\Delta$ATN is the change in attenuation during the time interval $\Delta$t.

It is noteworthy that $b_{ATN}$ differs from the aerosol absorption coefficient of airborne particles because it is determined from the attenuation of light passing through a particle-laden filter. The filter is responsible for measurement artifacts. These artifacts can be corrected with different procedures to account for the so-called loading effect and multiple scattering inside the filter matrix (Weingartner, et al. 2003, Arnott, et al., 2005, Schmid, et al. 2006, Collaud Coen, et al. 2010; Drinovec et al., 2015). We used a known correction scheme (Weingartner et al., 2003). Parameters *C* and *R(ATN,λ)*, are introduced to convert Aethalometer attenuation measurements to absorption coefficients ($b_{abs(\lambda)}$):

$$b_{abs(\lambda)} = \frac{b_{ATN(\lambda)}}{C \cdot R(ATN,\lambda)} \tag{A3}$$

where *C* and *R(ATN,λ)* are the filter multiple scattering enhancement parameter and the wavelength-dependent loading effect correction parameter, respectively. The parameter *R(ATN,λ)* corrects for the loading effect due to the reduction in the optical path due to an increase of the sample collected on the filter over time (Weingartner, et al., 2003). *R(ATN,λ)* was dynamically determined following the Sandradewi et al. (2008b) algorithm. This approach was recognised to be one of the best approaches as correction does not affect data in terms of the absorption Ångström exponent (AAE) (Collaud Coen et al., 2010), the parameter describing the dependence of the absorption coefficient on the

wavelength. This scheme was previously applied to data collected at the investigated site (Ferrero et al., 2018), because the experimental assessment of HR must avoid any artificial perturbation of the AAE.

The parameter $C$ corrects for the enhanced optical path through the filter caused by multiple scattering of light by the filter fibers and by the particles embedded in it. The multiple scattering coefficient $C$ is determined by comparing the attenuation coefficient, that needs to be previously corrected for the loading effect ($b_{ATN}/R$; see equation A3), with the absorption coefficient measured simultaneously at the same wavelength with a reference instruments ($b_{abs\_ref}$) (in our case, MAAP) as follows:

$$C = \frac{b_{ATN}/R}{b_{abs\_ref}}$$
(A4)

Thus, applying a non-corrected attenuation coefficient (raw data, not corrected for the loading effect; e.g. Figure A6) represents an erroneous application of the Aethalometer data, as it features a systematic error – the loading effect. Correcting for the loading effect increases the value of the parameter C. For the reason above we did not included Figure A6 (here below) to the Supplemental material.

[Figure]

Figure A6. Linear correlation between the Aethalometer AE31 attenuation coefficient at 660 nm, not corrected for the loading effect ($b_{ATN,660nm}$ raw data), and the MAAP absorption coefficient at 637 nm.

However, the reviewer question posed the important issue of uncertainty. We describe it here below and we added the description to the method section 2.1 (Instruments), where the uncertainties of all the other instruments were already reported.

As mentioned above, absorption coefficient measurements are based on measurements of light transmission through the sample-laden filter, which needs to be compensated for different artifacts, like the multiple scattering effect and loading effect (Liousse et al., 1993; Petzold et al., 1997; Bond et al., 1999). In this respect, Collaud-Coen et al. (2010) tested different correction schemes on data from different sites and showed linear regression between the Aethalometer data corrected with the Weingartner et al. (2003) procedure and reference MAAP data, with slopes close to one and relative standard deviations on average of 23%. This is an estimation of the global uncertainty of Weingartner et al. (2003) procedure applied in the present work. Moreover, Drinovec et al. (2015) showed a good agreement between Aethalometer AE31 data (corrected using Weingartner et al., 2003) and that of the new Aethalometer AE33 with a slope close to one and $R^2>0.90$. We referred to the Collaud-Coen et al. (2010) uncertainty estimation in our work.

**Specific Comment 7 (C4P1): Line 168 What is the physical meaning of the C value being close to the GAW value? (e.g.: The particle size and single scattering albedo were typical of atmospheric monitoring sites. Collaud Coen et al., 2010).**

Answer to Specific Comment 7 (C4P1): We thank the reviewer for this question as it enabled us to improve the manuscript. The physical meaning of the similarity between the obtained C value (3.24) and the GAW ones implies that Milan (in the middle of the Po Valley) features aerosol with continental characteristics (e.g. Carbone et al., 2010) not far from the global ones. However, the question that emerges is the physical reliability of the C value given the findings reported in Collaud Coen et al. (2010). Collaud Coen et al. (2010) defined the reference value of C ($C_{ref}$) for the AE31 tape in the pristine atmosphere of Jungfraujoch and Hohenpeissenberg, where aerosol has a single scattering albedo of ~1; $C_{ref}$ was equal to 2.81±0.11.

At the same time, Collaud Coen et al. (2010) took into account the cross-sensitivity to scattering of the filter measurements and its influence on the parameter C, starting from $C_{ref}$ as follows:

$$C = C_{ref} + \alpha \frac{\omega_0}{1-\omega_0} \tag{A5}$$

where $\alpha$ is the parameter for the Arnott (2005) scattering correction (0.0713 at 660 nm) and $\omega_0$ the single scattering albedo which, in wintertime in Milan, within the mixing layer, was found to be 0.846±0.011 at 675 nm by Ferrero et al. (2014). With respect to C interpretation, we need to underline first that the nominal AE31 660 nm channel is provided by a Kingbright light-emitting diode (APT 1608SRC PRV 1.6 x 0.8 mm SMD Chip LED Lamp; King bright, 2018) which is characterized by a 20 nm spectral full bandwidth at half maximum under 20mA of supplied current (information from manufacturer). This is in agreement with the absorption photometer intercomparison, reported by Muller et al. (2011), in which the nominal AEs red channel was found to have a 23 nm spectral full bandwidth at half maximum. Thus, for practical purposes, the single scattering albedo (0.846±0.011 at 675 nm) reported in Milan at a wavelength slightly different from the one featured in the AE31 by Ferrero et al. (2014) was applied to eq. A5.

Considering the variability for both $C_{ref}$ (±0.11) and $\omega_0$ (±0.011) the obtained C for Milan was 3.20±0.15. This lies within the experimental range obtained from the comparison of the AE31 with the MAAP: 3.24±0.03. Calculating in the opposite direction, the retrieval of $\omega_0$ using the experimental C and $C_{ref}$, led to a value of 0.858±0.043 which is very close to the value reported by Ferrero et al. (2014), underling the reliability of the obtained results.

We added the aforementioned considerations in section 2.1 where the experimental C is presented. Moreover section 2.1 was divided in two subsections (2.1.1 Light absorbing aerosol measurements and 2.2.2 Radiative and meteorological measurements) due to the requirements of the Specific Comment 8 below.

**Specific Comment 8 (C4P1): 171 Was the MRI built by U Milano-Bicocca? Please add manufacturer, even if it is homebuilt.**

Answer to Specific Comment 8 (C4P1): We thank the reviewer for the suggestion which improves the instrument description. The MRI was developed at the University of Milano-Bicocca by PhD Sergio Cogliati using commercial-grade optoelectronics devices. The instrument uses an optical switch (MPM-2000-2x8-VIS, Ocean Optics Inc., USA) to sequentially select between different input

fibers fixed to the upwards- and the downwards-looking entrance fore-optics. The configuration used in the present work connects each spectrometer to 3 input ports: 1) The CC-3 cosine-corrected irradiance probes to collect the down-welling irradiance; 2) the bare fiber optics with a 25° Field-of-View to measure the up-welling radiance from the terrestrial surface; 3) the blind port that is used to record the instrument dark-current. A 5 m long optical fiber with a bundle core with a 1 mm diameter is used to connect the entrance fore-optics to the multiplexer input, while the connection between the multiplexer output ports and the spectrometers is obtained with a 0.3 meters long optical fibers. The set-up allows to sequentially measure the dark-current and both up- and down-welling spectra simultaneously with the two spectrometers – High Resolution HR4000 holographic grating spectrometers (Ocean Optics Inc., USA). Finally, the MRI is equipped with a 3648-element linear CCD-array detector (Toshiba TCD1304AP, Japan) with a 14-bit A/D resolution.

We added this description to section 2.1, which was also divided in two subsections (2.1.1 Light absorbing aerosol measurements and 2.1.2 Radiative and meteorological measurements) due to the deepening also required by your previous Specific Comment 7.

**Specific Comment 9 (C4P1): 200 What is the uncertainty or accuracy of the Nimbus 15k? In other words, please mention the limitations as well as the strengths of this system.**

Answer to Specific Comment 9 (C4P1): The Lufft Nimbus CHM-15K is a high-performance lidar-ceilometer system operating at 1064 nm and capable of providing vertical profiles of aerosols and clouds in the bottom 15 km of the atmosphere with a temporal resolution of 30 seconds and a vertical resolution of 15 m.

In order to improve the signal to noise ratio of the backscatter signal, the signal is processed with temporal averages of 2 minutes. The full overlap is obtained at altitude of some hundred meters from the observation site and overlap correction functions are applied in the first layers. More technical information are provided by: Wiegner, M. and Geiß, A.: Aerosol profiling with the Jenoptik ceilometer CHM15kx, Atmos. Meas. Tech., 5, 1953–1964, doi:10.5194/amt-5-1953-2012, 2012, and Madonna, F., Amato, F., Vande Hey, J., and Pappalardo, G.: Ceilometer aerosol profiling versus Raman lidar in the frame of the INTERACT campaign of ACTRIS, Atmos. Meas. Tech., 8, 2207–2223, 2015.

We added the Madonna et al. (2015) reference to the manuscript, while the Wiegner and Geiß (2012) reference was already included.

We added the following sentence to section 2.1.2 Radiative and meteorological measurements: "Given the vertical resolution of the instrument, expected accuracy on the cloud base height derived by the lidar-ceilometer is < ±30 m".

**Specific Comment 10 (C4P1): 238 "This is due to the negligible ..." This sentence is not accurate. The authors may instead consider the simple harmonic oscillator (Moosmuller et al. 2011, doi:10.5194/acp-11-1217-2011 ) or energy gap (Sun 2007 doi:10.1029/2007GL029797) models here.**

Answer to Specific Comment 10 (C4P1): We thank the reviewer for this comment. The sentence aimed simply at recalling the intrinsic property of BrC: it features an absorption spectrum that smoothly increases from the VIS to UV wavelengths, as recently described by Laskin et al. (2015; DOI: 10.1021/cr5006167 Chem. Rev. 2015, 115, 4335−4382) who pointed out that "light absorption

by BrC at 440 nm is~40% of the light absorption by BC at this wavelength, while BrC contributes only 10% to the light absorption at 675 nm". Similarly, Moosmueller et al. (2011) shows in their Fig. 7 that there are ~1.5 orders of magnitude between the mass absorption efficiencies for relevantly sized particles. However, we understood from your question that this sentence was poorly connected with the previous one: "Conversely, BrC absorption is spectrally more variable, with an AAE from 3 to 10 (Ferrero et al., 2018; Shamjad et al., 2015; Massabò et al., 2015; Bikkina et al., 2013; Yang et al., 2009; Kirchstetter et al., 2004)." leading to misinterpretations.

We fully agree with you that the lower absorption coefficient of BrC in the IR region (compared to UV) is a consequence of the large wavelength difference (IR) with respect to the resonance in the UV, as can be described by the simple harmonic oscillator reported in Moosmuller et al. (2011). The band-gap model with the Urbach tail (Sun et al., 2007; and referenced in Moosmuller et al., 2011), where the key factor is the difference between the highest occupied and lowest unoccupied energy state of the molecules in the BrC ensemble, gives similar results.

We reworded the sentence as required adding this explanation.

**Specific Comment 11 (C4P1): 243 change 'successfully' to 'previously'**

Answer to Specific Comment 11 (C4P1): Done.

**Specific Comment 12 (C4P1): 251 First use of LAA on this line was not defined. Please introduce the concept in the introduction. It is a nice and useful abbreviation**.

Answer to Specific Comment 12 (C4P1): We thank the reviewer for addressing this. We modified the introduction accordingly.

**Specific Comment 13 (C4P1): 350 Give limits of the integral in the equation.**

Answer to Specific Comment 13 (C4P1): There is no equation in line 350 . We interpreted your question as relating to Eq. 7. We added the limits to the integral.

**Specific Comment 14 (C5P1): 135 This information is redundant with lines 110-112, please shorten 110-112.**

Answer to Specific Comment 14 (C5P1): We thank the reviewer for addressing this. We shortened lines 110-112 as required.

**Specific Comment 15 (C5P1): 171-177 Please cite Cogliati immediately after introducing the MRI.**

Answer to Specific Comment 15 (C5P1): We agree. Changed.

**Specific Comment 16 (C5P1): 263 Change "N represents one of the possible 9 classes" to "N = 1,2,3, ..., 9, representing 9 classes of cloud fractions".**

Answer to Specific Comment 16 (C5P1): We thank the reviewer for the question from which we understand that the sentence was poorly written. In the original version, we stated "where $N$ represents one of the possible 9 classes of cloud fraction". The term "cloud fraction" was improperly used as the correct sentence would be "where $N$ represents one of the possible 9 classes of sky conditions expressed in oktas (from 0, clear sky, to 8, complete overcast)". We rephrased the sentence as above.

**Specific Comment 17 (C5P1): 3.1 Heating rate measurements – Equation 1,2, and 3 There is no need for all 3 equations here. Delete Eq 1. Start with Eq 2 to introduce and define ADRE first, then the reader will understand Equation 3 (new Eq 2) naturally. Remove μ and replace it with cos θ. There is no need to introduce μ because it is only used twice in the manuscript, and anyway cos θ is more easily understood.**
**Also, use the integral sign to specify "integral over the whole 2π" (line 209) instead of writing it only in words. Same for θ.**

Answer to Specific Comment 17 (C5P1): Thanks for the suggestion. We modified the text according to the suggestion above.

**Specific Comment 18 (C5P1): Equations 4 and 5 use subscripts dir, dif, ref to specify direct, diffuse and reflected radiation. Equations 1, 2, 3 use "nth type of F" to do the same. Choose one and stick to it. The text subscript is better, and the authors obviously agree as they used it later in the manuscript.**

Answer to Specific Comment 18 (C5P1): We agree indeed. We changed the manuscript accordingly by using the subscripts dir, dif, ref to specify direct, diffuse and reflected radiation to avoid unnecessary new symbols and make the work more readable. This also goes in the direction asked by reviewer#2 which requires the use of less technical jargon.

**Specific Comment 19 (C5P3.1): The term Fglo = Fdir + Fdif + Fref must be introduced already in the first equation of Section 2.2. Prepare the reader for Equation 6. I have to assume that this is the definition, the authors never gave it.**

Answer to Specific Comment 19 (C5P3.1): We thank the reviewer for addressing this point. There is a misunderstanding that we have to clarify. This improved the methodology section. We need to start from the radiometric definition of the global downwelling irradiance which is as follows:

$$F_{glo} = F_{dir} + F_{dif} \tag{A6}$$

We added this definition in Appendix A. Equations 1-4 (now 2-3 in the revised version of the manuscript) could have caused the misinterpretation, as the calculation of the ADRE and the HR requires the sum of the total amount of radiative energy interacting with light-absorbing aerosol, also including the reflected irradiance in addition to the direct and diffuse components from the sun and sky. In fact, an alternative writing of the ADRE is:

$$ADRE = \int_{\lambda} AF_{(\lambda)} b_{abs(\lambda)} d\lambda \tag{A7}$$

where $AF_{(\lambda)}$ represents the actinic flux, that is the total spectral flux of photons per unit area and wavelength interval available to molecules/aerosol at a particular point in the atmosphere. The

radiative flux from all directions onto a volume of air is called the actinic flux (Seinfeld and Pandis, 2006). We added this information in section 2.1.2.

The actinic flux consists of three components: direct solar radiation, diffuse radiation originating from scattering in the atmosphere, and diffuse radiation originating from reflection from the Earth's surface.

Thus, it is only for the AF that the following sum is valid:

$$AF_{tot} = AF_{dir} + AF_{dif} + AF_{ref} \tag{A8}$$

The actinic flux at a particular point in the atmosphere is calculated by integrating the spectral radiance over all directions of space. The actinic flux must be distinguished from spectral irradiance, which is the hemispherically integrated radiance weighted by the cosine of the angle of incidence, and represents the photon flux per unit area through a plane surface. Under the isotropic and Lambertian assumptions, the diffuse and reflected irradiances are related with the corresponding radiances by a factor $\pi$; the direct irradiance is related to the radiance as a function of the solar zenith angle.

From a physical point, given a generic monochromatic radiance $R(\lambda, \theta, \phi)$, the corresponding $AF(\lambda)$ and irradiance $F(\lambda)$ (Seinfeld and Pandis, 2006; Liu, 2007) are given by:

$$AF(\lambda) = \int_{\phi=0}^{\phi=2\pi} \int_{\theta=0}^{\theta=\pi/2} R(\lambda, \theta, \phi) \sin(\theta) \, d\theta d\phi \tag{A9}$$

$$F(\lambda) = \int_{\phi=0}^{\phi=2\pi} \int_{\theta=0}^{\theta=\pi/2} R(\lambda, \theta, \phi) \cos(\theta) \sin(\theta) \, d\theta d\phi \tag{A10}$$

For the direct component, the radiance comes only from the sun direction (the solar zenith angle, SZA), it can be assumed to be a collimated beam, essentially parallel, and originates from a very small solid angle and thus:

$$AF_{dir}(\lambda) = R_{dir}(\lambda) = F_{dir}(\lambda)/\cos(SZA) \tag{A11}$$

For the diffuse and reflected component (under the isotropic and Lambertian assumptions, respectively) the radiance comes homogeneously from each direction and thus:

$$AF_{dif,ref}(\lambda) = 2\pi R_{dif,ref}(\lambda) \tag{A12}$$

$$F_{dif,ref}(\lambda) = \pi R_{dif,ref}(\lambda) \tag{A13}$$

implying:

$$AF_{dif,ref}(\lambda) = 2F_{dif,ref}(\lambda) \tag{A14}$$

Now, as in section 2.2 we gave the following definition:

$$ADRE = ADRE_{dir} + ADRE_{dif} + ADRE_{ref} \tag{A15}$$

we can finally rewrite it (given eq. A7 and A8) as follows:

$$ADRE = \frac{1}{\cos(SZA)} \int_{\lambda} F_{dir}(\lambda) \, b_{abs}(\lambda) \, d\lambda + 2 \int_{\lambda} F_{dif}(\lambda) \, b_{abs}(\lambda) \, d\lambda + 2 \int_{\lambda} F_{ref}(\lambda) \, b_{abs}(\lambda) \, d\lambda \tag{A16}$$

With the heating rate being:

$$HR = \frac{1}{\rho C_p} \cdot ADRE \tag{A17}$$

We included this description in the supplemental material and changed the use of the word "radiation" in the manuscript text as already described in the answer to the general comment 4.

**Specific Comment 20 (C5P1): The only real difference between HR and ADRE in Figure 5 is the air density ρ. So, the authors should plot ρ in the figure and emphasize this in the caption to avoid confusing readers who are not familiar with HR or ADRE (in other words, most readers).**

**Since the main contribution of this manuscript is to discuss heating rates, why discuss ADRE at all? Leave that to the SI. Or, of the authors disagree, then discuss only ADRE. HR appears more valuable as ρ is a meteorological variable.**

Answer to Specific Comment 20 (C5P1): Thanks for this comment. The paper focuses on the HR which is the valuable parameter. ADRE is introduced in section 2.2 (Heating rate measurements) for methodological purposes (see answer to Specific Comment 17, where we followed the instruction to first introduce ADRE). For these reasons and in keeping with the suggestion, we removed both ADRE values from section 3 (results and discussions) and its plot from Figure S5 (now Figure 5 in the revised version of the manuscript); this avoids confusing readers and keeps them focused on the main target of the work (i.e. HR, cloudiness and cloud type).

**Specific Comment 21 (C6P1): This section is a mess. Do not mix discussion and results in S 2.3.2. Review the literature first, then present your results. Present "failed analysis" in the SI, not in the main text. Use only 1 or 2 sentences in the main text for failed analysis.**

Answer to Specific Comment 21 (C6P1): Indeed section 2.3.2 was differently structured in the first draft of the manuscript, underling first the literature methods and presenting the methodology. A shortening of the paper before submission probably resulted in the confusing section. We apologize for that. We completely restructured it following this comment.

**Specific Comment 22 (C6P1): This reviewer spent several minutes studying Figure 2 and writing the following comments before learning that it is a "this did not work" figure. The writing should make this clear immediately. Restructure S2.3.2 to fix this.**

Answer to Specific Comment 22 (C6P1): Thank for addressing it as we were able to fix a couple of inaccuracies. We improved the legend and we also fixed panels (g-h) by indicating with coloured dashed lines the time periods to which the dots reported in the SD-R plot of panel h refers to. The new Figure 2 is reported here below as Figure A7.

[Figure]

Figure A7 (new Figure 2 in the manuscript). Cloud classification based on broadband solar radiation following Duchon & O'Malley (1999). Each row represents a different clout type in a specific day as a case study. The left columns represent the time series of global and diffuse measured solar irradiance ($F_{glo}$ and $F_{dif}$) and modelled clear sky irradiance ($F_{glo\_CS}$), while the right column the scatter plot of the observed standard deviation of irradiance (SD) vs. the fraction of modelled clear sky irradiance (R). In the panel (h) different colors are related to different time (hours) of the day as reported in the legend.

**Specific Comment 23 (C6P1):** **The section here concludes that the 2 literature methods discussed (Duchon 1999 and Harrison et al 2008) were not adequate, based on the conclusions of Harrison's work. So the authors introduce a new method, but with no validation of it. How can the reader trust this? I believe the author's work is valuable but the discussion needs to include validation.**

Answer to Specific Comment 23 (C6P1): Thank you for remarking on the need for a validation. As reported in the answer to the general comment 1 (C1P1) a thorough validation was carried out and described at length in Appendix B ("Cloud type validation") resumed here below.

**A resume of Appendix B: Cloud type validation**
The validation was conducted in two subsequent steps.
The first validation step was carried out by comparing the automatized cloud classification (based on Duchon and O'Malley, 1999, and additionally lidar cloud base height) with a visual cloud classification based on sky images collected during 1 month of field campaign. We describe this in Appendix B1.
The second validation step involved the recently published method (Ylivinkka et al., 2020), based on the same approach followed in our work: the application of Duchon and O'Malley (1999) classification improved by the knowledge of the cloud base height. Thus, the aim of the second step was to determine the degree of consistency between the two approaches that were developed simultaneously and independently in two completely different European regions. We describe this in Appendix B2.

Both validations were evaluated by means of a confusion matrix, a special kind of contingency table, with two dimensions and identical sets of "classes" in both of them. From the confusion matrix, the balanced accuracy was computed as follows:

$$Balanced\ Accuracy = \frac{Sensitivity + Specificity}{2} \qquad (B1)$$

[revised manuscript text omitted]

**Specific Comment 24 (C6P1): Line 308 how many cases (%) were analyzed after this limitation?? The authors should not discard cases of multiple cloud layers. Simply include a category "Multiple layers" or "Complex cloud layers" or similar.**

Answer to Specific Comment 24 (C6P1): Thanks. We analysed 8405 one single layer cases, 61% of the total. The single layer choice is related to the aim of the paper: "to experimentally measure for the first time the impact of different cloudiness and cloud types on the HR exerted by light-absorbing aerosol" as stated in the introduction section. In this respect it was also clarified in section 2.3.2 "Finally, to avoid misclassification cases due to the presence of multiple cloud layers, we limited the

analysis to those cases where only one cloud layer was detected by ceilometer. This choice was also done given the main goal of this work: to quantify the effects of different cloudiness and cloud types on light-absorbing aerosol HR. ''
We added the number of cases and their percentage to section 2.3.2.

**Specific Comment 25 (C6P1-C7P1): If I am to believe this section then the authors have contributed a numerical algorithm to the topic of automated cloud type analysis. Only 2 papers have been published on this topic, and most cloud type identification remains manual. This is the 3rd paper to contribute to this topic in 30 years, yet the authors did not include a solid analysis of the algorithm. Either the authors have used an unvalidated algorithm in their work, or the authors should write an entire manuscript describing their validation of what seems to be a valuable contribution. Separating the cloud algorithm work from the radiative heating work would mean removing Figures 2, 3, 4, 7, 14, and some SI figures from this to another manuscript. This would avoid breaking up the "BC+BrC" story.**
**I note that the Harrison et al. 2008 work was missing from the reference list.**

Answer to Specific Comment 25 (C6P1-C7P1): The cloud classification literature reports a huge quantity of papers and reviews aimed at classifying clouds (to avoid the limits of a simple manual human inspection) by means of different techniques and their integration. Some examples are reported in Singh and Glennen (2005), Ricciardelli et al. (2008), Calbó and Sabburg (2008), Tapakis and Charalambides (2013). Whith respect to the Po Valley, the Duchon and O'Malley (1999) was previously successfully applied by Galli et al. (2004). Moreover, the introduction of ceilometer data for cloud classification and cloud study purposes does not represent an absolute novelty in literature as demonstrated by Huertas-Tato et al. (2017) and Costa-Surós et al. (2013).
The novelty of the actual study is the combination of the Duchon and O'Malley (1999) with ceilometer cloud base height data. However, as reported in the answer to your General Comment 1, we have to underline that just after our submission of the present paper (20 Mar 2020), Ylivinkka et al. submitted to Atmospheric Measurement Technique (03 Apr 2020) a paper titled "Clouds over Hyytiälä, Finland: an algorithm to classify clouds based on solar radiation and cloud base height measurements"(https://doi.org/10.5194/amt-13-5595-2020) which was recently published. The paper discusses a cloud classification technique very similar to ours. This is a clear coincidence of an interesting scientific development. We therefore maintained the present paper as whole and improved it following reviewer suggestions by appropriately balancing the main body of the manuscript, Appendix, and the Supplemental material. For the validation we refer to the answer to the Specific Comment 23.
Finally, many thanks for finding that the Harrison et al. (2008) reference was missing. We added it the reference list together with the other aforementioned ones.

**Specific Comment 26 (C7P1): Make Fig S3 axis labels consistent with the language in S 2.3.2. Put R on the x axis.**

Answer to Specific Comment 26 (C7P1): We thank the reviewer for addressing it. Done.

**Specific Comment 27 (C7P1):** **The use of a 20 minute interval in calculating SD for the Duchon and O'Malley method means that wind speeds are included in the measurement of cloudiness fluctuations. This must be discussed. How do wind speeds compare with this 20 minute interval?**

Answer to Specific Comment 27 (C7P1): The 20-minute interval reported in the Duchon and O'Malley (1999) considers the variability induced by cloud movement and evolution (e.g. cloud microphysical processes) on the global irradiance. The standard deviation changes in the global irradiance can therefore be due to the wind influence on the cloud dynamics. However, the wind influencing these processes is the wind at the cloud altitude, not the one at ground where we carried out measurements. It would be great to have a Doppler lidar able to measure wind speeds at these altitudes. Nevertheless, we investigated the ground wind behavior for the cloud type classified in the present work together with the SD parameter. Results are reported in Figure A7 below. As expected, there is no strong correlation between the two parameters, as the wind speed was measured at ground level and reflects the stagnant conditions typical of the Po Valley. The average wind speed during each cloud type and clear sky conditions was below 1 m s$^{-1}$. Despite this, it is clearly visible that low-level clouds (e.g. stratus) are present in the lowest wind speed conditions. We added Figure A7 in the Supplemental material and added this discussion in the main body of the manuscript.

[Figure]

Figure A7. Wind speed (at ground) and SD for each cloud type.

**Specific Comment 28 (C7P1):** **Lines 281-314 break up this huge paragraph.**

Answer to Specific Comment 28 (C7P1): The section was rewritten (see answer to Specific Comments 21-25).

**Specific Comment 29 (C7P1):** **Figure 2 comments: Fig 2's legend is inaccurate, there is no dashed line in the legend. In Fig 2g, colour the red line in the same way that the points in Fig 2h are coloured. Move the entire figure to the SI. Consider adding photographs to this figure.**

Answer to Specific Comment 29 (C7P1): As reported in the answer to your Specific Comment 22, Figure 2 was improved by changing the legend: the nomenclature and assigning the coloured and

dashed symbols to the proper lines. We also fixed panels (g) by indicating with coloured dashed lines the time periods to which the dots reported in the SD-R plot of panel (h) refer to. In Appendix B, a new Figure B1 shows the SD-R plot with the corresponding cloud pictures. Figure 2 instead has the aim to introduce the reader to Duchon and O'Malley (1999) method as required by the Specific comment 21.

**Specific Comment 30 (C7P1): The results and discussion are too long, relative to the information content of the manuscript. The information is valuable but does not require extensive discussion.**
**As I mentioned above, the discussion is broken up by switching between the cloud analysis and the BC+BrC analysis. Start from the top and go down. Focus on the cloud effects before attributing HR to LAA afterwards and then to BC+BrC.**

Answer to Specific Comment 30 (C7P1): We agree that the paper needs a shortening in the results and discussion with more concise and precise sentences. We also took care of the organization of the sections. We refer to the answer to the General Comment 2, which details all the changes reported in the manuscript.
We thank the reviewer for these comments which enabled us to improve the paper.

**Specific Comment 31 (C8P1): The authors introduce direct, diffuse, and reflected irradiance yet do not present the data consistently. Some figures separate all 3. Some figures present direct, diffuse, and total (Fig 14). Some figures (Fig 13) present sums of 2, in various combinations. Some figures combine all 3 as "global irradiance" others combine all 3 as "total irradiance". Please, assess your data, choose one message, and present it clearly to your audience. Follow Harrison et al. 2008 in presenting the diffuse fraction unless your data support an alternative. Figure 9 is the only figure that suggests a difference between direct and reflected, but the impact on heating rate is unclear because Figure 13 changed the presentation strategy.**

Answer to Specific Comment 31 (C8P1): We carefully read this comment. The rationale was to present the total HR behaviour and that of each of its components: $HR_{dir}$, $HR_{dif}$, $HR_{ref}$.
Thus, under this presentation strategy we have made the following changes:
- Figure 5a (now Figure 6a in the revised version of the manuscript) presents both the total HR (due to the contribution of direct, diffuse, and reflected irradiance to the actinic flux, see answer to your Specific Comment 19) and its components $HR_{dir}$, $HR_{dif}$, $HR_{ref}$. Figure 5b (now Figure 6b in the revised version of the manuscript) reports the standard irradiance measurements $F_{glo}$, $F_{dir}$, $F_{dif}$, $F_{ref}$. Please note that the global irradiance is related to the reflected one just via the surficial albedo effect. Please see also our reply to Specific Comment 19.
- Figure 6 (now Figure 9 in the revised version of the manuscript) was improved by adding to old Figure 6a the $HR_{ref}/eBC$ and $F_{ref}$. Conversely, we added the total HR to Figure 6b. The figure is reported below as Figure A8.
- Figure 8a-d (now Figure 11a-d in the revised version of the manuscript) complies with the rationale of presenting the total HR/eBC behaviour and the each of its components: $HR_{dir}/eBC$, $HR_{dif}/eBC$, $HR_{ref}/eBC$. These are reported in panels a, b, c and d, respectively.

- Figure 9 (now Figure 12 in the revised version of the manuscript) presented only HR$_{dir}$, HR$_{dif}$, HR$_{ref}$. We also added the total HR. It is reported below as Figure A9.
- Figure 13a-d (now Figure 15a-d in the revised version of the manuscript) complies with the rationale of presenting the total HR behaviour and the each of its components: HR$_{dir}$, HR$_{dif}$, HR$_{ref}$ for both BC and BrC. They are reported in panels a, b, c and d, respectively.

The diffuse fraction was introduced in Harrinson et al. (2008) mostly for cloud applications. In the present work, the splitting of the total HR in the each of its components (HR$_{dir}$, HR$_{dif}$, HR$_{ref}$) reflects not only a radiative behavior, but a synergy with the light-absorbing aerosol features. The contribution of the HR$_{dif}$ to the total is discussed in the results and discussion sections.

[Figure]

Figure A8 (Figure 9 in the revised version of the manuscript). Monthly averaged values of: a) HR/eBC values together with their direct and diffuse components (HRdir/eBC and HRdif/eBC) and the direct and diffuse components of global radiation (*Fdir* and *Fdif*); b) HR values and their direct, diffuse and reflected components (HRdir, HRdif and HRref) during winter and spring both in clear sky (CS; oktas=0) and cloudy (CLD; oktas=7-8) conditions.

[Figure]

Figure A9 (Figure 12 in the revised version of the manuscript). Average values of total HR, HR*dir*, HR*dif* and HR*ref* for different cloud types.

**Specific Comment 32 (C8P1): The conclusions are similarly confused. Why are different cloud types discussed in detail when Figure 10 and 15 clearly show that the key predictor is oktas and not cloud type? Only high clouds (cirrus, cirrocumulus, and cirrostratus) do not follow this trend, presumably because they are well above the aerosol layers.**

Answer to Specific Comment 32 (C8P1): Figure 10 and 15 relate the HR variation (compared to CS values) with respect to the oktas induced by a different cloud types. Figure A2c (in the answer to the General Comment 2) reports a linear relationship between HR and the cloudiness (expressed in oktas) without accounting for the cloud types responsible for such sky coverage. The linear relationship was very good ($R^2$=0.935) and only slightly different from the similar relationship reported in Figure 10 ($R^2$=0.963; now Figure 13 in the revised version of the paper). This strengthens the synergy between the fraction of sky covered by clouds and cloud type influencing the transmission of shortwave radiation. In addition to this, the cloudiness (oktas) is a non-linear function of the cloud type, as cloud types are related to the meteorological pattern: e.g. highly persistent stratiform clouds generate cloudy weather in conditions with lower wind. Figure A7 (answer to the Specific Comment 27) reports an average ground wind speed of 0.64±0.02 m s$^{-1}$ in stratus conditions, lower than the 0.92±0.04-1.04±0.03 m s$^{-1}$ in cirrus-clear sky conditions. As a result (Figure A10 added below and appearing as Figure 8 in the main body of the manuscript) the cloudiness associated with different cloud types starts at cirrus clouds (0.51±0.05 oktas) and increases to stratus clouds (7.20±0.04 oktas). This is in agreement with the recent work of Bartoszek et al. (2020) who associated higher cloudiness level with the presence of stratiform clouds. Moreover, they observed an increase in sunshine duration with a decrease in the incidence of low-level clouds, including mainly stratiform clouds underling the connection between cloudiness-cloud type and shortwave radiation.

We added this discussion to section 3.1 in the manuscript.

Finally, Figures 10 and 15 do not show any deviation of cirrus, cirrocumulus, and cirrostratus from the linear trend.

[Figure]

Figure A10. Cloudiness associated to each cloud type.

**Specific Comment 33 (C8P1): I do not see any support for the final conclusion that the cloud impact affected HR of BC more than of BrC. The absolute value of the BC HR was higher initially, so it would naturally change more. My interpretation of the authors' results is that there is no need to attribute cloud types in future work, and that cloud height data combined with diffuse fraction (Harrison et al. 2008) may be sufficient. If this work is to be extended to other monitoring sites the authors must address this point explicitly. Simpler measurements are more likely to be adopted by others.**

Answer to Specific Comment 33 (C8P1): We thank the reviewer for this comment which give us the opportunity to extend our previous answer. As answered to your Specific Comment 32, a relationship between cloud-type and cloudiness is present, moreover figure A2c showed that cloudiness alone is a good predictor of light-absorbing aerosol HR behaviour, but the association is closer when using cloud type (Figure 10, now Figure 13). In fact, as detailed in Tapakis and Charalambides (2013), in order to model the incoming irradiance, not only the effect of the cloudiness (in oktas) has to be taken into account, but also the cloud type, as not all clouds have the same effect on irradiance. There are different types of clouds with different dimensions, opacity and properties affecting the incoming irradiance differently. We added these considerations to section 3.2.2.

However, we agree that the cloudiness is a simpler parameter that can be likely adopted by others; at the same time, data suggests that the role of the cloud-type on the spectral irradiance and HR of both BC and BrC shouldn't be neglected and needs to be further investigated elsewhere. We explain this at the end of section 3.3.

Finally, it is necessary to point out that Figure 15 (now Figure 16) reports not the absolute decrease in the HR (which is obviously higher for BC as you reported) but the relative decrease in % normalized by the HR value in clear sky conditions for BC and BrC. Results are therefore independent of the magnitude of the absolute variation.

---

## Author Comment (AC2) · 15 Nov 2020

**Response to Reviewer#2**

We thank the reviewer for his or her helpful comments and insight. We respond to the general and specific points below. All the comments are addressed in the revised manuscript.

**GENERAL COMMENTS**

General Comment 1: This study explores the effect of clouds on heating rates driven by absorbing aerosols. They do so using observations and measurements sorted per different cloud types and coverage, separating the effects of black vs. brown carbon.

The data is collected in U9 sampling site in Milan which is a superstation that contains instruments to measure radiation, filter collecting aerosols that are analysed for their optical properties, meteorological station and a Lidar.

The topic of the paper is important. Exploring heating rates for different aerosol types under different cloud conditions will provide a very important information for aerosol effect on climate, and clouds. As the authors pointed out direct measurements of heating rates in different cloud conditions are quite uncommon.

The basic cloud classification makes sense in particularly as they added Lidar information for the clouds base. The results clearly show how cloudiness can affect heating rates and the bland between the radiation types.

Answer to General Comment 1: We thank the reviewer for the comment which remarks on the big effort put in this work, and the quality of both the methodological approach and of the obtained experimental results.

General Comment 2: One drawback of the paper is that it is very technical and not always easy to follow. Even if one understands the radiative transfer concepts, the physical assumptions and results are buried in the technicalities. It contains many technical terms that may appeal only to the instrumentation experts. Being familiar with radiation transfer concepts, I'm sure that there is a better way to describe the measurements and analyses such that a non-expert in the instrumentation could better enjoy it. The concertation of acronyms is high. It is hard to remember all of them and some that appear again later in the text force the reader to look back for their meaning and it disturbs the reading. On the other hand, some basic concepts that are key in this study are not well explained. The authors send the reader to read many other references for the basic methods and the equations. I believe that such study could be more of a standalone in which the basic physics is explained in a better way using less technical jargon.

Answer to General Comment 2: Thank you very much for this comment which enabled us to improve the scientific quality and presentation of the paper. Reviewer#1 asked for a shortening of the results and discussion as well as an improvement of the logical steps in the methods sections. Thus, the paper was substantially changed and improved to make it more readable and easier to follow. We have rearranged the sections to improve readability, especially the results and discussion section in the manuscript:

- In section 3.1 we introduce the environmental context of the measurement campaign and the magnitude of the observed parameters ("eBC, irradiance, HR and cloud data"). We separated

the cloud analysis from the light-absorbing aerosol analysis. All the cloud analysis is presented to the reader in this section.

- The old sections 3.2 and 3.3 were re-written in agreement with the changes performed in section 3.1, and merged in a new section 3.2 with two sub-sections. The first discusses the influence of clouds in term of cloudiness while the second the influence of cloud type sub- on the total HR only (sections 3.2.1 and 3.2.2, respectively).
- Finally, the old section 3.4 (now section 3.3) was completely re-written merging and shortening the two original sub-sections 3.4.1 and 3.4.2. We discuss the role of cloudiness, cloud type and their effect on the HR apportioned with respect to BC and BrC fractions.

We have reduced the number of equations in the text. Instead of Eqs. 1-3 we now present only 2 equations in the main body of the manuscript, as follows:

"The radiative power absorbed by the aerosol in a unit volume of air (W m-3; ADRE: absorptive direct radiative effect) describes the interaction between the radiation (either direct from the Sun, diffuse by atmosphere and clouds, or reflected from the ground) and the LAA (BC and BrC, Ferrero et al., 2018) and is determined as follows:

$$ADRE = \sum_{dir,dif,ref} \int_{\theta} \int_{\lambda} \frac{F_{dir,dif,ref(\lambda,\theta)}}{\cos(\theta)} b_{abs(\lambda)} d\lambda d\theta$$
(A1)

where the subscript *dir*, *dif* and *ref* refers to the direct, diffuse and reflected component of the spectral irradiance  $F(\lambda, \theta)$  of wavelength  $\lambda$  that strikes (from any azimuth) with an angle  $\theta$  (from the zenith) the aerosol layer and  $b_{abs(\lambda)}$  is the wavelength dependent aerosol absorption coefficient. Please see Supplemental Material for further details.

To obtain the heating rate of the light-absorbing aerosol HR we divide ADRE by the air density ( $\rho$ , kg m-3) and the isobaric specific heat of dry air ( $C_p$ , 1005 J kg-1 K-1):

$$HR = \frac{1}{\rho c_p} \cdot ADRE \tag{A2}$$

We introduce the indices *dir*, *dif*, *ref* to avoid the readers' confusion about the original "n" symbol which refered to each of the different kinds of impinging radiation.

In keeping with your suggestion, we removed many acronyms and technical terms whenever possible. In agreement with a suggestion from reviewer#1, we prepared a list of acronyms and symbols (used in the manuscript) which was added in the new section Appendix A at the end of the paper.

In line with the suggestion to reference more papers, we added in the Supplemental Material an alternative notation of equations 1 as follows:

$$ADRE = \int_{\lambda} AF(\lambda)b_{abs(\lambda)}d\lambda \tag{A3}$$

where  $AF(\lambda)$  represents the actinic flux, that is the total spectral flux of photons per unit area and wavelength interval illuminating the molecules/aerosol at a particular point in the atmosphere where the term *actinic* refers to radiation capable of causing photochemical reactions or capable of being absorbed. The actinic flux (actually a flux density) consists of three components: direct solar radiation, diffuse radiation originating from scattering in the atmosphere, and diffuse radiation originating from reflection at the earth's ground surface. Accordingly, the actinic flux at a particular point in the atmosphere is calculated by integrating the spectral radiance over all directions in space. The actinic flux must be distinguished from the spectral irradiance, which is the hemispherically integrated radiance weighted by the cosine of the angle of incidence, and represents the photon flux per unit area through a plane surface. A more exhaustive description can be found in Liou (2007), Tian et al. (2020) and Gao et al. (2008). We added these references to the method section 2.2 of the manuscript and deepened the topic in the Supplemental material.

**SPECIFIC COMMENTS**

Specific Comment 1 (SC1): The aerosols that are collected at the station level serve as the only aerosol measurement and the basic assumption is that the filters collected at the station represent the whole boundary layer and therefore the heating rate is uniform for the layer below the clouds. I wonder how general this assumption is? This is always a key question of any work that try to link measurements near the surface to the atmospheric column. Is it always well mixed? Can the authors show that there is no dependency on the time of the day or the winds or the meteorology in general? Is it true for all seasons? For all cloud types? Moreover, if they have Lidar there can't they validate this assumption using the Lidar information. It would be nice to see uniform backscatter below the clouds to strengthen this basic assumption. The radiation measurements are collected in the station and are product of electromagnetic radiation interaction with the whole atmospheric column. What about the contribution of aerosols above the boundary layer. Is it assumed to be canceled by the proposed method? Or is it assumed to be negligible? If not, how such aerosols can affect the results?

Answer to Specific Comment 1 (SC1): Thank for all your questions. They are related to the methodology. In order to properly answer them it is necessary to address the following points: 1) the advantages and limitations of the applied methodology (relating to the measurements and derivation of the heating rate HR) and 2) the environmental context of the measuring site in the Po Valley (addressing the representativeness).

**Methodology advantages and limitation**

The most important advantages/limitations of the new method are resumed here. The first consideration is that the ADRE (and thus the HR) is the vertical derivative of the aerosol direct radiative effect (ADRE=dDRE/dz; see Ferrero et al. (2018)); we provide a detailed analysis at the end of the answer (*Methodology details and demonstration*). Thus, both the ADRE and the HR become independent from the thickness ( $\Delta z$ ) of the investigated atmospheric layer as happens for routine atmospheric pollution measurements (i.e. BC, PM and particle number concentrations). The most important *advantages* in terms of HR measurements are:

- no radiative transfer assumptions are needed (i.e. clear sky situation), the input parameters into equations A1 and A2 are all measured,
- measurements of the spectral irradiance and the absorption coefficient are carried out at high time resolution, allowing to follow the HR dynamic with same temporal resolution,

- measurements of the spectral irradiance, the absorption coefficient and thus the HR are carried out in any sky conditions, enabling to investigate the impact by the cloud layers on the near-surface HR.

The most important *limitation* is the following:

- as both the ADRE and the HR are independent of the thickness ( $\Delta z$ ) of the investigated atmospheric layer, they refer to the vertical location of the atmospheric layer in which both the ADRE and the HR are experimentally determined. In the present work, they are determined in the near-surface atmospheric layer.

It is noteworthy to consider the advantages that the new method allows to obtain: experimental measurement (not estimations) of ADRE and HR continuous in time with a high time resolution as a function of sources, species of light absorbing aerosol, and cloud cover. The use of the vertical derivative of the Direct Radiative Effect allows us to obtain a temporal continuity of ADRE and HR but "paying" it with the loss of vertical information.

Due to your question, we first clarified these points in the methodological section 2.2 expanding the sentence (lines 245-247 in the submitted version of the manuscript):

"As already pointed out in Ferrero et al. (2018), it is worth recalling that in the present method (equation 1), both the ADRE and the HR are independent of the investigated atmospheric aerosol layer thickness."

and at lines 250-254:

"The main advantage of the new method to quantify the impact of clouds on the light-absorbing aerosol HR is that it allows to obtain experimental measurement (not estimations) of ADRE and HR, which are continuous in time with a high time resolution, and resolved in terms of sources, species of light-absorbing aerosol, cloud cover, and cloud types."

**Environmental context of HR measurements**

In this section we address the representativeness of the HR determination at ground and answer the Reviewer's questions. As reported in the submitted version of the manuscript at lines 247-250: "BC and HR vertical profiles data previously collected both at the same site and in other basin valley sited (Ferrero et al., 2014) revealed that ADRE and HR were constant inside the mixing layer. The methodology is therefore believed to be valid for applications in atmospheric layers below clouds, assuming that near-surface measurements are representative of the whole mixing layer." This assumption is the core of your question.

The aim of the paper is the investigation of the impact of cloudiness and cloud-type on the HR induced by light absorbing aerosol. Ground-based highly time-resolved HR data are suitable to reach this goal – we need to introduce the representativeness shown in Ferrero et al. (2014) over Milan.

We performed combined in-situ and remote vertical profile measurements in Milan with tethered balloons and a lidar (in cooperation with the ISAC-CNR of Rome) since 2005. The collected data shows a homogeneous distribution of aerosol concentration within the mixing layer. Figure A1

reports averaged wintertime balloon profiles (PM concentrations and extinction coefficient) and lidar range corrected signal for Milan (Ferrero et al., 2019).

Figure A1. Milan averaged wintertime a) balloon profiles of  $PM_1$ ,  $PM_{2.5}$  and  $PM_{10}$  and extinction coefficient b) lidar range corrected signal. Data for the present figure are from Ferrero et al. (2019).

The same condition was verified by the lidar-ceilometer data collected during the present campaign (Figure A2, here below).

Figure A2. Milan averaged wintertime lidar range corrected signal  $(SxR^2)$  during the campaign presented in the manuscript.

Vertical profiles data reported in Figure A1 and A2 experimentally verify the assumption "that nearsurface measurements are representative of the whole mixing layer" in wintertime in Milan. Figures A1b and A2 show a typical mixing layer height diurnal behavior in wintertime conditions, with the mixing layer height not exceeding 500 m above ground. The same was previously retrieved from the vertical gradient of tethered balloon aerosol profiles (Ferrero et al., 2010; Figure A3). Within the mixing layer, aerosol concentrations were uniform (as reported in Figure A1) along each time of the day.

---

## Author Response (AR1)

**Author's response**

Dear Editor, Prof. Urs Baltensperger,

Thank you and thank to the staff of ACP for the deadline extension given to our final revision paper. We are glad to notice that both referees appreciated the experimental efforts and the potential high relevance of the results presented in our paper.

Furthermore, both referees focused their attention on several issues, asking for elucidation of a number of technical points, which we are glad to focus on in the following responses to the referees. All the raised criticisms and relative answers have been addressed in the revised manuscript. Accordingly to the referees' concerns, a list (not exhaustive) of the most important changes made to the manuscript is the following:

- the paper was substantially changed and improved to make it more readable and easier to follow. We rearranged the sections to improve readability,
- the method section was improved both simplifying the heating rate measurements methodology (moving the demonstration of the radiative transfer concepts to the supplemental material) and clarifying the cloud classification algorithm; the non-core part of "average photon energy" was also moved to the supplemental material,
- a thorough validation of cloud classification was carried out and described at length in Appendix B,
- we fully re-organized the Results and discussion section following the referees' suggestions in order to improve the full manuscript and to clarify the logic behind the results presentation,
- Figures were improved accordingly to referees' comments and to make the data presentation more effective,
- a comprehensive list of acronyms was added in Appendix A

Finally, the whole text was proofread and edited to emendate the typos and to improve the language. We are pleased that this discussion based on the constructive criticisms of both referees has helped us to improve the scientific quality of the work done. A tracked version of the manuscript changes is present at the bottom of the answers.

With our best regards,

Yours sincerely,

Dr. Luca Ferrero

**Response to Reviewer#1**

We thank the reviewer for his or her helpful comments and insight. They allowed us to improve the scientific quality and presentation of the work done. We respond to the general and to the specific points below. All the comments are addressed in the revised manuscript. A tracked version of the manuscript changes is present at the bottom of the answers.

**GENERAL COMMENTS**

General Comment 1 (C1P1): The manuscript by Ferrero et al. acp-2020-264 titled "The impact of cloudiness and cloud type on the atmospheric heating rate of black and brown carbon" presents heating rate measurements of the atmosphere over the Po Valley, Italy.

The measurements are valuable as they are relatively rare in the community. The work is incremental on Ferrero et al. 2018, with the main incremental improvement being the automated separation of clouds into cloud types using radiometer measurements combined with Lidar-Ceilometer measurements. The introduction of lidar information into the automated cloud classification is novel and may be valuable to other work, yet was not thoroughly validated. I recommend that the authors describe this cloud classification algorithm in detail in a separate paper and include more detailed validation work. If the authors do not follow this recommendation, they must provide a clear argument for why in the review responses and in the manuscript.

Answer to General Comment 1 (C1P1): We thank the reviewer for the comment on the experimental results, their relevance and implications reported in this work. Indeed, as underlined in the review, the present work represents an important incremental step of Ferrero et al. (2018).

We carefully considered the suggestion to split the paper in two. However, the cloud classification is only one of the incremental improvements. The main goal of our study is to experimentally unravel the relative and synergic role of cloudiness and of different cloud types on the heating rate (HR) of light absorbing aerosol (LAA) in general and that of BC and BrC in particular. As we state at the end of the introduction (revised version of the manuscript) we aim to:

- 1. describe the interaction between cloudiness and light-absorbing aerosol, to aerosol HR as a function of cloudiness, and in turn to estimate the systematic bias introduced by incorrectly assuming clear-sky conditions in radiative transfer models;
- 2. introduce an original cloud type classification to investigate the impact of both cloudiness and cloud types on the total HR;
- 3. separate the contributions of BC and BrC carbonaceous fractions to HR and investigate their relative impact on the total HR as a function of sky conditions.

The results we present in this study add an important piece of information to the influence of the two most important LAA species (i.e. BC and BrC) in different sky conditions. Therefore, the manuscript was planned from the beginning as a whole, with the main focus on the environmental influence of LAA on the climate.

Immediately after our submission (20 Mar 2020), Ylivinkka et al. submitted to Atmospheric Measurement Technique (03 Apr 2020) a paper titled "Clouds over Hyytiälä, Finland: an algorithm to classify clouds based on solar radiation and cloud base height measurements"(

https://doi.org/10.5194/amt-13-5595-2020) which was accepted and published (22 Oct 2020). The paper discusses a cloud classification technique very similar to ours. This is a clear coincidence of an interesting scientific development.

Taking into account the reasons above, due to the fact that the concerns were mostly related to one section (2.3.2, cloud classification section), and due to Reviewer#2, asking for a technical simplification of the paper, we have decided (previously asking the opinion of the handling editor) to not split the paper into two, but rather to improve the present manuscript. We have rewritten large part of the manuscript main body, and moved material to the Appendix and the now modified Supplemental material. We have additionally taken into account the publication of Ylivinkka et al. (2020) and included this and other references related to the lidar-ceilometer capabilities at detecting cloud base and cloud classification. To answer this reviewer comment, a validation of the classification scheme was carried out in in two steps.

The first validation step was carried out comparing the automatized cloud classification (based on Duchon and O'Malley, 1999, and additionally lidar cloud base height) with a visual cloud classification based on sky images collected during 1 month of the field campaign. The second validation step involved the recent published method discussed by Ylivinkka et al. (2020). Their method is based on the same logical approach followed in our work: the application of Duchon and O'Malley (1999) classification improved by the knowledge of the cloud base height. The aim of the second step was to determine the degree of consistency between the two approaches which were developed simultaneously and independently in two completely different European regions.

The complete validation is reported in Appendix B ("Cloud type validation"). This was performed not to interrupt the flow of the manuscript, as requested in the Specific Comment 25 (C6P1-C7P1). The overall balanced accuracy was 80% for the visual validation and 90% for the intercomparison with Ylivinkka et al. (2020) (please see answer to your specific question 23, C6P1, for further details). This shows the reliability of the classification algorithm, allowing us to study the impact of clouds on LAA HR with a sufficient degree of certainty.

General comment 2 (C2P1). The actual presentation of the results in this manuscript is incredibly poor. Here I present 200 lines of comments which I had to make simply in order to understand the results. The discussion is long, dense, and disorganized. Most of these comments are on presentation and organization, at the level which is normally given to an author's first draft of a first manuscript. After I finally understood what was done and what the results were, I see a valuable data set. However, the scientific interpretation is on a similar level to the writing.

I fear that my scientific feedback has been drowned by the poor writing, manuscript organization, and figure presentation in this work. To emphasize my main scientific comments I have used boldface text in the following. The authors should streamline their manuscript by referring to Ferrero et al. 2018 whenever possible, by separating their cloud analysis from their light-absorbing aerosol analysis, and by clearly demonstrating whether or not there is any value to the different levels of information available here. Those levels are: 1) heating rate resolved in time, 2) heating rate resolved by time and cloud height, 3) heating rate resolved by time, cloud height, and cloud type.

My recommendation to the authors is to completely rewrite this manuscript and reinterpret the results. Since this work is incremental to earlier, well-presented work (Ferrero et al., 2018), and

since the results are well supported if poorly presented and interpreted, I do not recommend rejection but major revisions to the Editor.

(NB: I have not numbered my feedback below. When the authors respond, please refer to my comments as "C1P2" for page C1, paragraph 2, etc. Please also copy and paste the comment before responding.)

Answer to General Comment 2 (C2P1): We have considerably rewritten the manuscript as suggested in the comment.

We started with the suggestion "to separate the cloud analysis from the light-absorbing aerosol analysis". As reported in the answer to General Comment 1, we cannot split the manuscript in two manuscripts, as a similar cloud classification scheme was just published. The strength and the innovation of the present paper is the synergy between the automatic classification of cloudiness and quantifying the effect of the light-absorbing aerosols on the climate. Thus, we fully re-organized the Results and discussion section following the suggestion in order to improve the full manuscript, to clarify the logic behind the methodology, and to more specifically discuss the different aspects (levels) of the results. Now, following the suggestion on the three different levels of information, the Results and discussion section features the following arrangement of the subsections:

- Section 3.1 introduces the environmental context of the measurement campaign and the magnitude of the observed parameters (eBC, irradiance, HR and cloud data). We have incorporated here the suggestion "to separate the cloud analysis from the light-absorbing aerosol analysis". All cloud analysis is presented here. The validation of the cloud classification was moved to the Appendix B.
- Old sections 3.2 and 3.3 were re-written in line with the changes performed in section 3.1 and merged in a new section 3.2 with three sub-sections. The first discusses the influence of clouds in term of cloudiness, the second the influence of cloudiness on the diurnal pattern of the HR while the third the cloud type effect on the total HR only (sections 3.2.1, 3.2.2 and 3.2.3, respectively).
- The old section 3.4 (now section 3.3) was completely re-written merging and shortening the two original sub-sections 3.4.1 and 3.4.2. We discuss the role of cloudiness, cloud type and their effect on the HR apportioned with respect to BC and BrC fractions. The discussion concerning the role of average photon energy was moved to the supplemental material.

This gradual approach streamlines the manuscript, making it easier to read. We improved the Results and Discussions outline at the beginning of section 3 describing this approach. Moreover, all the manuscript was revised simplifying all the sections and making them more concise and easier to follow. We did not use the acronyms for the concepts which did not appear too often and also added an Appendix explaining all the remaining acronyms and symbols present in the paper.

To address the suggestions about the data analysis and the most relevant results, we moved the Figure S5 (time resolved heating rate) to the main body of the manuscript (now Figure 5 in the manuscript; here below as Figure A1) adding a proper description. We first improved the new Figure 5 adding both the cloudiness (expressed in oktas) and the cloud base height. The same was also done for Figure S6 (now Figure S4).

Then we focused on the reviewer's suggestions concerning the relationship between 1) the heating rate and cloud height and 2) the heating rate, cloud height and cloud type. This helped us to enrich the explanation of the interaction between the clouds and light-absorbing aerosols. We prepared Figures A2a-c, A3 and A4a-d which are discussed here below.

Figure A1 (Figure 5 in the revised version of the paper). High time resolution data (5-min) for eBC, global irradiance (Fglo), cloud base height (CBH), coldness (oktas), and the related heating rate (HR) from 1 November 2015 to 1 April 2016.

---

## Editor Decision (ED1)

**The impact of cloudiness and cloud type on the atmospheric heating rate of black and brown carbon in the Po Valley**

Luca Ferrero1,\*, Asta Gregorič2,3, Grisa Močnik3,4, Martin Rigler2, Sergio Cogliati1,5, Francesca

5 Barnaba6, Luca Di Liberto6, Gian Paolo Gobbi6, Niccolò Losi1 and Ezio Bolzacchini1
1GEMMA and POLARIS Research Centers, Department of Earth and Environmental Sciences, University of Milano-Bicocca, Piazza della Scienza 1, 20126, Milan, Italy.
2Aerosol d.o.o., Kamniška 39A, SI-1000 Ljubljana, Slovenia.
3Center for Atmospheric Research, University of Nova Gorica, Vipavska 11c, SI-5270 Ajdovščina, Slovenia.

4Department of Condensed Matter Physics, Jozef Stefan Institute, SI-1000 Ljubljana, Slovenia.
 5Remote Sensing of Environmental Dynamics Lab., DISAT, University of Milano-Bicocca, P.zza della Scienza 1, 20126, Milano, Italy

[revised manuscript text omitted]

**3.2 Cloud impact on the heating rate**

**3.2.1 The role of cloudiness**

- Figure 6a already provided the first indication of the important influence of clouds on the total HR. In fact, it shows the magnitude of the absolute (and relative) contribution of the diffuse component (HRdif) with respect to the total HR revealing that, on a monthly basis, the diffuse contribution accounted on average  $40\pm1\%$  (of the total HR). In most cases this was comparable or even higher than HRdir. The only exception was in November 2015 when the lowest HRdif (Figure 6a) and Fdif (Figure 6b) fractions in total HR and Fglo were measured ( $30.4\pm1.4\%$  and  $34.3\pm2.6\%$  of the total, respectively), this also being the month with the lowest average cloudiness ( $2.91\pm0.06$
- 520 oktas). The aforementioned data demonstrate the importance of the diffuse component of radiation. Therefore, the absolute values of the HR and its components were firstly investigated as a function of cloudiness (clear sky and complete overcast situations, seasonal averages, Figure 9a). In the wintertime clear sky, the direct component of the HR (HRdir) was higher than HRdif and HRref accounting for  $1.35\pm0.04$  K day-1 and explaining on average  $60\pm5\%$  of the total HR. Similarly, in the springtime clear sky HRdir was  $0.47\pm0.01$  K day-1 again higher than HRdif and

[revised manuscript text omitted]

- 600 absolute contribution of  $84.4\pm3.8$ ,  $83.0\pm10.7$  and  $76\pm4\%$  (HRdif:  $0.25\pm0.01$ ,  $0.34\pm0.03$  and  $0.66\pm0.02$  K day-1), respectively.

Given this impact of cloud type, the ability of cloudiness to be a good predictor for the HR (as detailed in section 3.2.1) and the relationship (over the investigated site) between cloudiness and cloud type (section 3.1, Figure 7b), the synergic impact of cloudiness and cloud type on HR was investigated and presented in Figure 13. In the figure,

- 605 we summarize the HR results in terms of percent difference from the clear sky (CS) case by averaging the cloudiness (in oktas) for each cloud type (as detected in section 3.3). Overall, the derived linear regression indicates a HR decrease of  $-11.9\pm1.2\%$  per okta. The regression R2 (0.963) was slightly higher than that reported in Figure S5b ( $R^2=0.935$ ; relationship with the cloudiness only) suggesting the need (for precise calculations) to account for the cloud types responsible for any sky coverage in agreement with a recent work of Bartoszek et al. (2020). Figure
- 610 13 also allowed us to associate the HR decrease to each specific cloud type over Milan. Particularly, Ci were found to produce a modest impact on cloudiness ( $0.50\pm0.05$  oktas) decreasing the HR by ~3%, while Cu ( $1.76\pm0.09$ oktas) decrease the HR by  $-26\pm8\%$ . Cc-Cs (oktas of  $3.56\pm0.14$ ) were responsible for a  $-49\pm6$  decrease of the HR. Their impact was comparable to that of Sc (4.68±0.10 oktas, -48±4% of HR). Ac (4.11±0.18 oktas) had a higher impact, decreasing the HR by -59±6%. The highest impact was due to As (6.57±0.15 oktas; -76±4% of HR) and
- 615 by St (oktas:  $7.19\pm0.04$ ) that suppressed the HR by a factor of  $-83\pm4\%$ .

**3.3** The impact of clouds on the BC and BrC heating rates**

In this last part of the work we focus on the HR of the two main absorbing aerosol species: BC and BrC (obtained as detailed in section 2.1.1). The monthly averaged values of HR of BC and BrC ( $HR_{BC}$  and  $HR_{BrC}$ ) are reported

620 in Figure 14. The highest  $HR_{BC}$  and  $HR_{BrC}$  values were recorded in December (1.24±0.03 K day-1 and 0.19±0.01 K day-1) while the lowest were recorded in March (0.46±0.01 K day-1 and 0.07±0.01 K day-1). Overall,  $HR_{BrC}$  accounted for 13.7±0.2% of the total HR.

The variability of total HRBC and HRBrC as a function of cloudiness is reported in Figure 15a, with panels b-d showing their direct (HRBC,dir and HRBrC,dir), diffuse (HRBC,dif and HRBrC,dif) and reflected (HRBC,ref and HRBrC,ref)

- 625 components. Figure 15a shows that both HRBC and HRBrC decreased with increasing cloudiness, going from the CS maxima (HRBC and HRBrC: 1.14±0.03 and 0.20±0.01 K day-1) to the completely overcast conditions (oktas=8) minima of 0.16±0.01 and 0.02±10-3 K day-1 (mainly due to St and As clouds; see Figure 7b). As shown in Figure 9b, the change of irradiance magnitude with cloudiness was different for direct, diffuse and reflected components affecting the corresponding direct, diffuse and reflected components of HRBC and of HRBrC (Figure 15b-d). HRBC,dir
- 630 and HRBrC,dir (Figure 15b) decreased as a function of cloudiness from 0.74±0.03 and 0.11±0.01 K day-1 (oktas=0) to negligible levels (HR<10-4 K day-1) in completely overcast conditions. HRBC,dif and HRBrC,dif (Figure 15c) increased with cloudiness, reaching their maximum in partially cloudy conditions (at oktas=6, 0.51±0.01 and 0.09±0.01 K day-1). Further increasing cloudiness reduced their values to minimum values (0.13±0.01 and 0.02±0.01 K day-1). HRBC,ref (Figure 15d) behave similarly to the total HRBC and HRBrC, since the
- 635 reflected irradiance is dominated by the global irradiance impinging on the ground (see Figure 9b for a comparison); HRBC,ref and HRBrC,ref decreased with increasing oktas from maximum values in clear sky (HRBC,ref and HRBrC,ref: 0.17±4\*10-3 and 0.03±1\*10-3 K day-1) down to overcast minimum (HRBC,ref and HRBrC,ref 0.02±10-3 and 3\*10-3±10-3 K day-1). Figure 15a-d also shows that HRBC was always greater (in absolute values) than HRBrC, as expected. The relative decrease of HRBrC from CS to complete overcast conditions was 12±6% larger with

[revised manuscript text omitted]

---

## Author Response (AR2)

Dear Editor,
Prof. Urs Baltensperger,

Thank you and thank to the staff of ACP for your work. We are glad to notice that both referees appreciated the effort done in the revision process.
As required the whole text was proofread and edited to emendate the typos and to improve the language (here below the complete list of changes).

With our best regards,
Yours sincerely,

Dr. Luca Ferrero

**Language corrections**
Line 1: "cloud" was modified in "clouds"
Lines 22-23: "in all sky condition" was changed: "in all sky conditions"
Line 29: "Average cloudiness" was changed in "The average cloudiness"
Line 31: "situations" was changed to "conditions"
Line 36: "in" was changed to "of"
Line 37: "from" was changed to "in"
Line 38: "and over" was changed to "up to"
Line 44; "respectively, while cirrocumulus-cirrostratus by -60±8 and -54±4%," changed to "respectively; cirrocumulus-cirrostratus decreased the $HR_{BC}$ and $HR_{BrC}$ by -60±8 and -54±4%,"
Lines 45-46: "A higher impact on $HR_{BC}$ and $HR_{BrC}$ was found for stratocumulus" changed to "A higher impact on $HR_{BC}$ and $HR_{BrC}$ suppression was found for stratocumulus"
Line 49: "by a factor" was removed
Lines 60-61: "Both direct and indirect radiative effects of anthropogenic and natural aerosols on climate are still the major sources of uncertainties" changed to "Both direct and indirect radiative effects of anthropogenic and natural aerosols are still the major sources of uncertainties on climate"
Line 62: "that the aerosol direct radiative effect, on a global scale, may switch from positive to negative" changed to "that the aerosol direct radiative effect (on a global scale) may switch from positive to negative"
Lines 64-65: "aerosols are a heterogeneous complex mixture of particles characterized by different size, chemistry, and shape" changed to "aerosol is a heterogeneous complex mixture of particles characterized by different size, chemistry, and shape"
Lines 69-71: "However, inaccuracies related to simplified model assumptions on chemistry, shape, and the mixing state of particles can affect the results (Nordmann et al., 2014; Koch et al., 2009), amplifying the uncertainties on the estimated global and regional aerosol effects on the climate (Andreae and Ramanathan, 2013)." changed to "However, inaccuracies related to simplified model assumptions on chemistry, shape, and the mixing state of particles can affect the results (Nordmann et al., 2014; Koch et al., 2009); this amplifies the uncertainties on the related global and regional aerosol effects on the climate (Andreae and Ramanathan, 2013).
Lines 72-74: "Although the clear sky approximation is useful when comparing measurements to radiative transfer modelling outcomes during experimental campaigns performed in fair weather conditions (e.g., Ferrero et al., 2014; Ramana et al., 2007), in general" changed to "The clear sky approximation is useful when comparing measurements to radiative transfer modelling outcomes

during experimental campaigns performed in fair weather conditions (e.g., Ferrero et al., 2014; Ramana et al., 2007); however, in general"

Line 77: a comma was added before "especially"

Line 80: "the" was added before "scattering"

Line 82: "determined the that scattering" was changed to "determined that the scattering"

Line 96: "and, for example for the south eastern Atlantic," changed to "and, for the south eastern Atlantic,"

Lines 100-101: "they estimated an all-sky direct radiative effect for total anthropogenic aerosols of -0.27 W m$^{-2}$ (range: −0.58 to −0.02 W m$^{-2}$)" changed to "they estimated an all-sky direct radiative effect of -0.27 W m$^{-2}$ (range: −0.58 to −0.02 W m$^{-2}$) for total anthropogenic aerosols"

Lines 104-105: "BC is the second most important positive anthropogenic climate-forcing agent" was changed to "BC is the second most important positive anthropogenic climate forcer"

Line 105: ", while BrC contributes" was changed to "; BrC contributes"

Line 110: "absorbing aerosol" was substituted by "LAA"

Lines 111-112: "thus affecting atmospheric stability, cloud distribution and even synoptic winds such as the monsoons" was changed to "thus affecting the atmospheric stability, the cloud distribution and even the synoptic winds such as the monsoons"

Lines 116-117: "Since, similarly to aerosols, cloudiness and cloud type change on short time scales, long-term, highly time-resolved measurements, covering different sky conditions," was changed to "Since cloudiness and cloud type change on short time scales similarly to aerosols, long-term, highly time-resolved measurements (covering different sky conditions)"

Lines 127-128: "reported a counterintuitive feedback linking the atmospheric heating induced by tropospheric absorbing aerosol to a cloud cover increase" was changed to "reported a counterintuitive feedback: the atmospheric heating induced by tropospheric absorbing aerosol could lead to a cloud cover increase.

Lines 137-140: "The study was performed in Milan (Italy), located in the middle of the Po Valley (section 2), which is an air pollution hot-spot in Europe with meteorological characteristics similar to those of a multitude of basin valleys surrounded by hills or mountains in which low wind speeds and stable atmospheric conditions promote the accumulation of aerosol" was improved in "The study was performed in Milan (Italy), located in the middle of the Po Valley (section 2), which is an air pollution hot-spot in Europe; its meteorological conditions are similar to those of a multitude of basin valleys (surrounded by hills or mountains) in which low wind speeds and stable atmospheric conditions promote the accumulation of aerosol"

Lines 141-143: "Cloud presence cannot be neglected over the investigated area considering that in the last 50 years annual mean cloudiness, expressed in oktas, is estimated to be ~5.5 over Europe" was changed to "Cloud presence cannot be neglected over the investigated area considering that, in the last 50 years, the annual average cloudiness, expressed in oktas, was estimated to be ~5.5 over Europe"

Line 144: "This feature is similar with" changed to "This feature is similar to"

Line 156: "Aerosol, cloud and spectral irradiance measurements were carried in an experimental measurement station located in Milan" was changed to "Aerosol, cloud and spectral irradiance were measured in Milan"

Lines 161-162: "A full description of the aerosol behavior in Milan at the University of Milano-Bicocca and of the aerosol properties" was changed to "A full description of the aerosol behavior in Milan, at the University of Milano-Bicocca, and of the related properties"

Line 172: "The aerosol, cloud and radiation instrumentations has been installed" was changed to "The aerosol, cloud and radiation instrumentations have been installed"

Lines 188-189: "negative values are not generated and results in good agreement with other filter photometers" was changed to "negative values are not generated and the results are in good agreement with other filter photometers"

Line 191: a comma was added after 3.24±0.03

Line 193: "Global Atmospheric Watch" was corrected into "Global Atmosphere Watch"

Line 197: "the" was added before "Collaud Coen et al. (2010) procedure"

Lines 204-205: "From eq. 1 it follows that the expected C in Milan is 3.20±0.35; within its range the experimental 3.24±0.03 value lies." was rewritten as "From equation 1, it follows that the computed C in Milan is 3.20±0.35, in keeping with the experimental one (3.24±0.03)."

Line 210: "sites" was improved with "locations"

Line 211: "applied in the present work" was put in brackets

Line 213: "that of" was deleted

Line 222: "explaining that" was changed with ":"

Lines 223-224: "yields a decreasing AAE values" was improved to "yields to decreasing AAE values"

Line 234: "(Toshiba" was corrected in "; Toshiba"

Line 263: "measurements" was added to the end of the sentence.

Line 272: "The full overlap is obtained at altitude of some hundred meters" was changed to "The full overlap is obtained at an altitude of some hundred meters"

Line 280: a comma was added after the word "data"

Line 284: "absorbing aerosol" was substituted with "LAA"

Line 298: "describing" was changed to "which describes"

Line 294: "(as used in Ferrero et al., 2018)  equation 2" was improve into "(as used in Ferrero et al., 2018), the equation 2"

Line 299: "Eqs." was changed to "Equations"

Line 300: "eq. 3" was changed to "the equation 3"

Line 304: "to derive HR" was changed to "to derive the LAA HR"

Line 307: "input to equations 3" was corrected to "input to equation 3"

Line 310: "possibility to derive HR in all sky conditions," was corrected in "possibility to derive the HR in all sky conditions,"

Lines 315-316: "the HR was determined to the near-surface atmospheric layer" was changed to "the HR was determined into the near-surface atmospheric layer"

Line 320: "were performed since 2005" was corrected with "are performed since 2005"

Line 333: a "the" was added.

Line 340: "eq." was changed to "equation"

Line 347: "Identification of cloud classes" was changed to "The identification of clouds classes"

Line 351: "literature reports a huge quantity of papers and reviews aimed at classify clouds" was changed to "literature reports a huge quantity of papers and reviews aimed at classifying clouds"

Line 368: "$R_t$ is the 20-minutes running average" was changed to "where $R_t$ is the 20 minutes moving average"

Line 401: "an hourly-resolvedcolor code" was corrected into "an hourly-resolved color code"

Line 462: "The lowest" was changed to "The lower"

Line 463: "their highest values" was change into "their higher values"

Line 469: "the lowest monthly irradiance" was changed into "the lower monthly irradiance"

Line 470: "while the highest" was change in "while the higher"

Lines 470-471: "The highest monthly average HR was recorded in December (1.43±0.05 K day$^{-1}$) while the lowest one in March" was changed in "The higher monthly average HR was recorded in December (1.43±0.05 K day$^{-1}$) while the lower one in March"

Lines 477 and 479: "eq. 3" was changed with "equation 3"

Lines 488-489: "during the whole campaign, the average cloudiness was 3.58±0.04 oktas with the highest monthly value in February (4.56±0.07 oktas) and the lowest in November (2.91±0.06 oktas)." was changed in "during the whole campaign, the average cloudiness was 3.58±0.04 oktas with the higher monthly value in February (4.56±0.07 oktas) and the lower in November (2.91±0.06 oktas)."

Lines 519 and 520: "lowest" was substituted with "lower"

Lines 524, 536, 540, 627, 632, 640: "overcast conditions" was changed in "overcast condition"

Line 569: "during afternoon" was changed in "during the afternoon"

Lines 611-612: "were found to produce" was improved in "produced"

Line 643: a "that" was added to the sentence in brackets

Line 667: "Overall, the derived linear regressions indicate a decrease of ~12% per oktas" was corrected in "Overall, the derived linear regressions indicate a decrease of ~12% per okta"

Line 668: "In details," was corrected in "In detail,"

Line 675: a "the" was added the sentence.

Line 680: "prevailing clouds type" was corrected in "prevailing clouds types"

Lines 684-685: "were experimentally determined based on high time resolution radiation and aerosol measurements in the Po Valley" was changed in "were experimentally determined based on radiation and aerosol measurements (at high time resolution) in the Po Valley"

Line 702: "about 12% for both species" was improved in "about 12% per okta for both species"